
# From complex to simple : hierarchical free-energy landscape renormalized in deep neural networks

**Hajime Yoshino**[1,2*]

**1** Cybermedia Center, Osaka University, Toyonaka, Osaka 560-0043, Japan
**2** Graduate School of Science, Osaka University, Toyonaka, Osaka 560-0043, Japan

⋆ yoshino@cmc.osaka-u.ac.jp

## Abstract

We develop a statistical mechanical approach based on the replica method to study the design space of deep and wide neural networks constrained to meet a large number of training data. Specifically, we analyze the configuration space of the synaptic weights and neurons in the hidden layers in a simple feed-forward perceptron network for two scenarios: a setting with random inputs/outputs and a teacher-student setting. By increasing the strength of constraints, i.e. increasing the number of training data, successive 2nd order glass transition (random inputs/outputs) or 2nd order crystalline transition (teacher-student setting) take place layer-by-layer starting next to the inputs/outputs boundaries going deeper into the bulk with the thickness of the solid phase growing logarithmically with the data size. This implies the typical storage capacity of the network grows exponentially fast with the depth. In a deep enough network, the central part remains in the liquid phase. We argue that in systems of finite width N, the weak bias field can remain in the center and plays the role of a symmetry-breaking field that connects the opposite sides of the system. The successive glass transitions bring about a hierarchical free-energy landscape with ultrametricity, which evolves in space: it is most complex close to the boundaries but becomes renormalized into progressively simpler ones in deeper layers. These observations provide clues to understand why deep neural networks operate efficiently. Finally, we present some numerical simulations of learning which reveal spatially heterogeneous glassy dynamics truncated by a finite width $N$ effect.

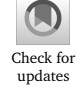
# 1 Introduction

Machine learning by deep neural networks (DNN) is successful in numerous applications [1]. However, it remains challenging to understand why DNNs actually work so well. Given the enormous parameter space, which is typically orders of magnitude larger than that of the data space, and the flexibility of non-linear functions used in DNNs, it is not very surprising that they can express complex data [2]. What is surprising is that such extreme machines can be put under control. On one hand, one would naturally fear that learning such a huge number of parameters would be extremely time-consuming because the fitness landscape is presumably quite complex with many local traps. Moreover, over-fitting or poor generalization ability seems unavoidable in such over-parametrized machines. We would not dare to fit a data set of 10 points by a 100 the order polynomial, which does not make sense usually. Quite unexpectedly, these issues seem to be somehow resolved in practice and such extreme machines turned out to be very useful. Thus it is a very interesting scientific problem to uncover what is going on in DNNs [3,4]. This is also important in practice because we wish to use DNNs not merely as mysterious black boxes but control/design them in rational ways. In the present paper we develop a statistical mechanical approach based on the replica method to obtain some insights into these issues.

In this paper, we investigate a class of simple machines made of feed-forward networks of layered perceptrons whose depth is $L$ and the width is $N$ (see Fig. 1). Such a machine is parametrized by a configuration of synaptic weights in the hidden layers. We consider the coupling between adjacent layers are *global* in the sense that all neurons in the $l$-th layer are connected to all neurons in $l+1$-th layer. For a given pair of inputs/outputs patterns imposed on the input and output layers, there can be different realizations of the synaptic weights that match the same constraints. We call each of them as a 'solution'. Following the work of Gardner [5,6] for the single perceptron, we consider statistical mechanics of the design space of the neural network which is compatible with a large number $M = \alpha N$ patterns of training data, in the large width $N \to \infty$ limit with fixed $\alpha$. For the choice of the training data, we consider two simple scenarios: 1) pairs of purely random inputs/outputs patterns 2) teacher-student setting - pairs of random input and the corresponding output of a teacher machine with random synaptic weights are handed over to a student machine.

From a broader perspective, the setting 1) can be viewed as a random constraint satisfaction problem (CSP) [7,8], which is deeply related to the physics of glass transitions and jamming [9–11]. In the context of neural networks, it is a standard setting to study the storage capacity [5,12]. If $\alpha$ is small so that the constraint is weak enough, it is natural to expect that the phase space looks like that of a liquid: there are so many realizations of machines compatible with a given set of constraints that essentially all solutions are continuously connected. Increasing $\alpha$ the system becomes more constrained so that the volume of the solution space shrinks and ultimately vanishes at some critical value $\alpha_j$. This is an SAT/UNSAT (jamming) transition and $\alpha_j$ defines the storage capacity. Interestingly, before reaching $\alpha_j$, the solution

space can become clustered into mutually disconnected islands. This is a glass transition and it accompanies some type of replica symmetry breaking (RSB) [13,14]. Recently non-trivial glass transitions accompanying continuous replica symmetry breaking, which imply the emergence of hierarchical free-energy landscape and ultrametricity [14–17], and common jamming critically as that of the hard-spheres [10] were found in a family of CSPs including a single perceptron problem [18,19] and a family of vectorial spin models [20]. Understanding the nature of such glass transitions and jamming is a fundamental problem in CSPs since it is intimately related to the efficiency of algorithms to solve CSPs. In the context of DNN, it is certainly important to understand the characteristics of the free-energy landscape to understand the efficiency of various learning algorithms for DNNs [21–23].

On the other hand the setting 2), is a statistical inference (SI) problem. While the constraint satisfaction problems are related to the physics of glass transitions and jamming, solving a statistical inference problem can be said to be equivalent to searching of a hidden (planted) crystalline state [8]. In the context of neural networks, it is a standard setting to study *learning* [6,12]. As $\alpha$ becomes sufficiently large, the synaptic weights of the student machine starts to become closer to those of the teacher machine. If this happens, the student machine starts to generalize: the probability that the student machine yields the same output as the teacher machine for a test data (not used during training), increases with $\alpha$. Although very simple, this setting will provide useful insights into the generalization ability of DNNs.

The present work is following the standard statistical mechanical approach to machine learning [12]. Extension of it to deeper neural networks has remained challenging. Our key strategy is to regard a DNN, not as a system of long-ranged interaction between the input and output through a highly convoluted non-linear mappings but rather as a system with short-ranged interactions between adjacent layers. This is enabled by the *internal representation* [24], in which one takes into account not only 'bonds' (synaptic weights) but also 'spins' (neurons) in the hidden layers as dynamical variables which are constrained to satisfy proper inputs/outputs relations at each perceptron embedded in the hidden layers. Representing the states of a neuron associated with $M$-patterns as $M$-component vectorial spins, the system can be represented as a network of dynamical variables with a large number of components with dense connections to each other.

The system is almost disorder-free in the sense that the 'quenched disorder' is present only on the boundaries so that one would fear that the usual replica theory for the single perceptron [5] cannot be easily extended for DNNs. However, the replica approach is not merely a trick to take the average over the quenched disorder. The recent progress on the exact replica theory in disorder-free systems like simple hard-spheres [10, 11, 25–27] in the large dimensional limit and disorder-free glassy spin systems [20] have proved that spontaneous replica symmetry breaking (RSB) exist in such systems which become manifest by considering infinitesimal symmetry breaking field explicitly as pointed out by Parisi and Virasoro [28]. One can even study spontaneous glass transitions of multiple degrees of freedoms such as translational and orientational degrees of freedom in aspherical particulate systems by the same approach [29].

The above observations motivate us to investigate the interior of the DNN through local glass or crystalline order parameters, for both the spins (neurons) and bonds (synaptic weights), which are allowed to vary over the space. We formulate a replica theory to analyze the design space of the deep perceptron network analyzing a free-energy expressed as a functional of the space-dependent, local order parameters. For simplicity, we limit ourselves within a tree-approximation which neglects effects of interaction loops along the $z$-axis. Thus our theory is inevitably a mean-field approximation of the original problem, which does not faithfully take into account 1 dimensional fluctuation along the $z$-axis (See Fig. 1). Nevertheless we believe our theory captures important aspects of the DNN. In a sense the present work may

be regarded as a Ginzburg-Landau type theory for the DNN. Consideration of loop-corrections would improve the quantitative accuracy of the microscopic mean-field theory but we leave it for future studies.

The main result of the present paper is that the solutions of the glass/crystalline order parameters of over-parametrized DNNs become quite heterogeneous in space (along $z$-axis). In both settings 1) and 2), the amplitude of the order parameters close to the inputs/outputs boundaries become finite and take higher values as the strength of the constraint $\alpha$ increases while the amplitudes decay down to 0 going deeper into the bulk. Moreover, in the case of setting 1) random inputs/outputs, even the pattern of the replica symmetry breaking (RSB) varies in space: it is most complex close to the boundaries with $k$(+continuous)-RSB, which becomes $k-1$(+continuous)-RSB in the next layer, ... down to a replica symmetric (0 RSB) state in the central part. The thickness $\xi$ of the region around the boundaries where the glass/crystalline order parameters become finite roughly scales as $\xi \propto \ln \alpha$. This implies the storage capacity of the network $\alpha_j(L)$ for *typical* instances grows exponentially fast with the depth $L$, while the worst case scenarios [30] would predict linear growth with $L$.

Thus if the network is deep enough $L > \xi$, the central part of a typical network remains in the liquid phase: there are so many possibilities left in the central part all of which meet the same constraints imposed at the boundaries. The heterogeneous profile of the order parameters should have important implications on how DNNs work.

The organization of the paper is the following. In sec. 2 we define the deep neural network model studied in the present paper and explain the two scenarios : 1) random inputs/outputs and 2) teacher-student setting. In sec. 3 we formulate a replica theory to perform statistical mechanical analyses of the design space of the deep neural network within a tree-approximation. In sec. 3.3 and sec. 3.5 we study the cases of 1) random inputs/outputs and 2) teacher-student settings respectively using the replica theory. In sec. 4, we present some results of numerical simulations to examine the theoretical predictions. Finally in sec. 5, we conclude the paper and present some outlook. In the appendices A, B and C we present some details of the theoretical formulation.

## 2 Model

### 2.1 Multi-layer feed-forward network

We consider a simple multi-layer neural network (See Fig. 1) which consists of an input layer ($l = 0$), output layer ($l = L$) and hidden layers ($l = 1, 2, \ldots, L-1$). Each layer consists of $i = 1, 2, \ldots, N$ neurons $\mathbf{S}_{l,i}$, each of which consists of $M$-component Ising spins $\mathbf{S}_{l,i} = (S_{l,i}^1, S_{l,i}^2, \ldots, S_{l,i}^M)$ with $S_{l,i}^\mu = \pm 1$. Here the label $\mu = 1, 2, \ldots, M$ is used to distinguish different firing patterns of the neurons (spins). The spins in the inputs/outputs layers represent 'data' provided by external sources. We follow the notation of [20] to represent a factor node, which is a perceptron here, as ∎. We consider a feed-forward network of $N_\blacksquare = NL$ perceptrons. A perceptron ∎ receives $N$ inputs from the outputs of perceptrons ∎$(k)$ ($k = 1, 2, \ldots, N$) in the previous layer, weighted by $\mathbf{J}_\blacksquare = (J_\blacksquare^1, J_\blacksquare^2, \ldots, J_\blacksquare^N)$. Its output $\mathbf{S}_\blacksquare$ is given by,

$$S_\blacksquare^\mu = \text{sgn}\left( \frac{1}{\sqrt{N}} \sum_{k=1}^N J_\blacksquare^k S_{\blacksquare(k)}^\mu \right), \qquad \mu = 1, 2, \ldots, M. \tag{1}$$

We assume that the synaptic weights $J_\blacksquare^k$ take real numbers normalized such that,

$$\sum_{k=1}^N (J_\blacksquare^k)^2 = N. \tag{2}$$

For simplicity, we call the variable for the neurons $\mathbf{S}_{l,i}$'s as 'spins', and the synaptic weights $J_{\blacksquare}^k$s as 'bonds' in the present paper.

We will consider random input data of size

$$N_{\text{data}} = NM = N^2 \alpha, \tag{3}$$

while the number of parameters is

$$N_{\text{parameter}} = N_{\blacksquare} N = N^2 L. \tag{4}$$

Here we introduced a parameter

$$\alpha \equiv \frac{M}{N}. \tag{5}$$

The task of *learning* is to design the synaptic weights $J_{\blacksquare}^k$ to build a mapping (function) between the imposed random input data and output data, which can be completely different, by a network of width $N$ and depth $L$.

We will consider the limit $N, M \to \infty$ with fixed $\alpha$. This scaling is known for the single perceptron [5] and we will find that it continues to be the key parameter for the bigger network much like the inverse temperature for condensed matters. Apparently the system is over parametrized $N_{\text{parameter}} > N_{\text{input}}$ if it is deep enough $L > \alpha$. ( Actually our results imply that typical storage capacity grows exponentially with the depth $L$ as we see later so that the system is essentially over-parametrized if $L > \ln \alpha$. ( see sec. 3.3.4)) We note that there might be other possible scalings different from Eq. (5). For example studies on some types of two-layer perceptron networks suggest other scaling such as $M = \alpha N^2$ is also possible (see Chap 12 of [12]). However, in the present paper we will limit ourselves to the scaling of the form Eq. (5).

The trajectories of such highly non-linear mapping as Eq. (1) along the random deep network is known to be highly chaotic [31,32]: small differences in the input data lead to rapid decorrelation of the resulting trajectories. This feature is considered as responsible for the high expressive power of DNNs [32]. Similarly, small changes made on the weights $J_{\blacksquare}^k$ also leads to chaotic decorrelation of trajectories [33]. But then we immediately face the obvious question: how the high generalization ability observed in DNNs can be explained when the system is so chaotic? In the present paper we construct a statistical mechanics point of view to answer such questions. Out of the set of all possible realizations of random deep networks, which typically give chaotic dynamics, we focus on a substantially smaller sub-manifold of it in which all trajectories (accidentally) meet the externally imposed boundary conditions put at the two opposite ends. This *selection* (learning) may have significant consequences on the properties of the resultant ensemble.

Following the pioneering work by Gardner [5,6] we consider the volume of the design space of the system associated with a given set of inputs/outputs patterns represented by $\mathbf{S}_0$ and $\mathbf{S}_L$, which can be expressed as,

$$V\left(\mathbf{S}_0, \mathbf{S}_L\right) = e^{NM\mathcal{S}(\mathbf{s}_0, \mathbf{s}_l)} = \left(\prod_{\blacksquare} \text{Tr}_{\mathbf{J}_{\blacksquare}}\right)\left(\prod_{\blacksquare \backslash \text{output}} \text{Tr}_{\mathbf{S}_{\blacksquare}}\right) \prod_{\mu=1}^{M} \prod_{\blacksquare} e^{-\beta V(r_{\blacksquare}^{\mu})}, \tag{6}$$

where

$$e^{-\beta V(r)} = \theta(r) \tag{7}$$

and we introduced the 'gap',

$$r_{\blacksquare}^{\mu} \equiv S_{\blacksquare}^{\mu} \sum_{i=1}^{N} \frac{J_{\blacksquare}^i}{\sqrt{N}} S_{\blacksquare(i)}^{\mu}. \tag{8}$$

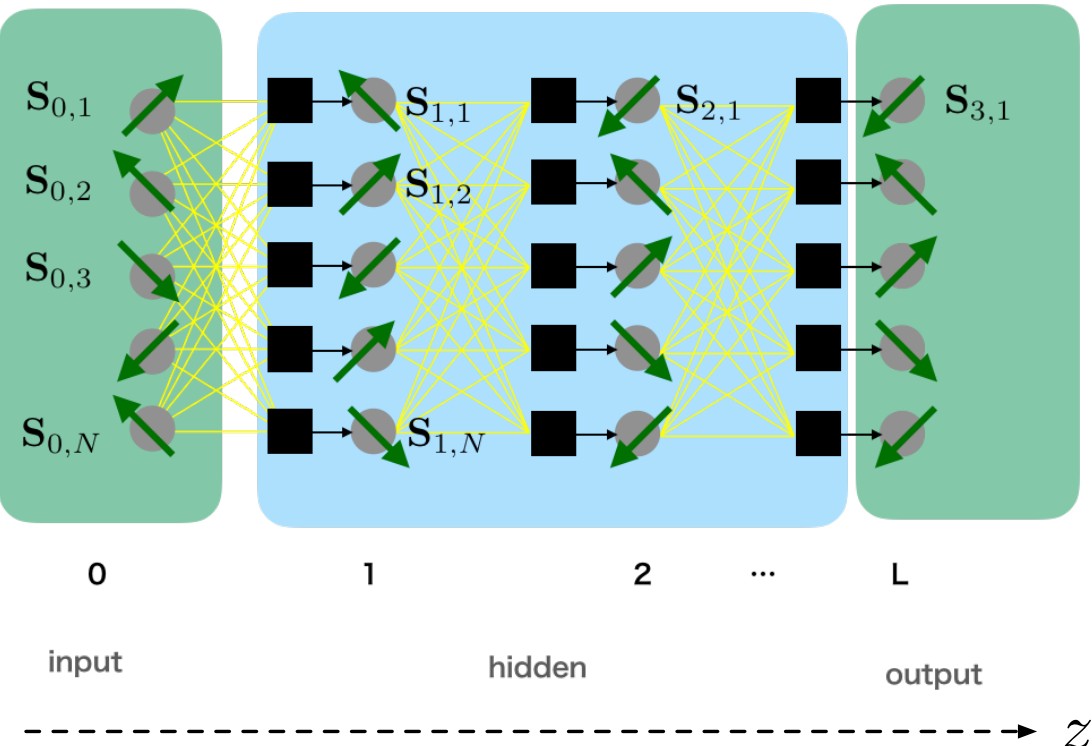

Figure 1: A simple multi-layer perceptron network of depth $L$ and width $N$. In this example the depth is $L = 3$. Each arrow represents a $M$-component vector spin $\mathbf{S}_i = (S_i^1, S_i^2, \ldots, S_i^M)$ with its component $S_i^\mu = \pm 1$ representing the state of a 'neuron' in the $\mu$-th pattern.

The trace over the spin and bond configurations can be written explicitly as,

$$\mathrm{Tr}_{\mathbf{S}} = \prod_{\mu=1}^{M} \sum_{S^\mu = \pm 1} \tag{9}$$

and

$$\mathrm{Tr}_{\mathbf{J}} = \int_{-\infty}^{\infty} \prod_{j=1}^{N} dJ^j \delta \left( \sum_{k=1}^{N} (J^k)^2 - N \right) = N \int_{-i\infty}^{i\infty} \frac{d\lambda}{2\pi} e^{N\lambda} \prod_{j=1}^{N} \int_{-\infty}^{\infty} dJ^j e^{-\lambda (J^j)^2}. \tag{10}$$

Note that in Eq. (6) summations are took not only over the bonds (synaptic weights) but also over the spins (neurons) in the hidden layers. This is the internal representation [24] which allows us to avoid viewing the system as a system with long-ranged interaction between the input and output through a highly convoluted non-linear mappings but rather as a system with short-ranged interactions between adjacent layers. From a physicist's point of view, this is far more convenient. Indeed we can now write the effective Hamiltonian of the system as

$$\mathcal{H}_{\mathrm{eff}} = \sum_{\mu=1}^{M} \sum_{\blacksquare} V \left( S_{\blacksquare}^\mu \sum_{i=1}^{N} \frac{J_{\blacksquare}^i}{\sqrt{N}} S_{\blacksquare(i)}^\mu \right). \tag{11}$$

This simple trick works because of the simple 'sgn' activation function Eq. (1) we consider in the present paper. Let us emphasize here that both the spins $S_{\blacksquare s}^\mu$ and bonds $J_{\blacksquare}^i$ are dynamical variables, except for the spins on the boundaries $l = 0, L$ which are frozen.

Now our task is to analyze the equilibrium statistical mechanics of the system of many variables with the effective Hamiltonian Eq. (11). In this point of view, we can forget about the 'feed-forwardness' of the original dynamical representation Eq. (1). If we imagine an inifinitely deep network without terminals or a network with the periodic boundary condition, one can regard the system as a globally homogenous $1(+\infty)$ dimensional system. With the boundaries, some inhomogeneity should emerge close to the boundaries.

The problem at our hands is similar to the statistical mechanics of an assembly of hard-spheres. Each of the configurations which meet the hard-core constraint Eq. (7) represents a valid trajectory (more precisely a set of $M$ perceptron trajectories all of which meet the corresponding inputs/outputs boundary conditions) of the original feed-forward problem. Like in the statistical mechanics of hard-spheres [11], everything that matters here is the entropy effect. For instance, we can expect that assembly of trajectories which consists of many nearby valid trajectories (which meet the same inputs/outputs boundary conditions) have richer (local) entropy so that they make important contributions to the total entropy. This corresponds to the notion of 'free-volume' of an assembly of hard-spheres [34]. Such an equilibrium statistical mechanics may not be merely academic. Indeed the standard schemes of deep learning involve Stochastic Gradient Descent (SGD) algorithms [1,35] which explores the solution space of DNNs in a stochastic way. There also trajectories with richer local entropy would appear more often during the sampling. In this paper we will find often that the analogy with the physics of hard-sphere glass [11] is very useful to understand our results in physical terms.

## 2.2 Two scenarios for inputs/outputs patterns

For the input and output patterns $\mathbf{S}_0$ and $\mathbf{S}_L$, we consider the following two scenarios.

### 2.2.1 Random inputs/outputs

As the simplest setting, we consider the case of completely random inputs/outputs patterns, which is the standard setting to study the storage capacity of the perceptrons [5,12]. More precisely all components of $\mathbf{S}_{0,i} = (S_{0,i}^1, S_{0,i}^2, \ldots, S_{0,i}^M)$ and $\mathbf{S}_{L,i} = (S_{L,i}^1, S_{L,i}^2, \ldots, S_{L,i}^M)$ for $i = 1, 2, \ldots, N$ are assumed to be iid random variables which take Ising values $\pm 1$. As we noted in the introduction, this setting can be regarded as a random constraint satisfaction problem (CSP).

### 2.2.2 Teacher-student setting

As a complementary approach, we consider the teacher-student setting, which is a standard setting to study statistical inference problems [8]. We consider two machines : a teacher machine and student machine and assume that they have exactly the same architecture, i.e. the same width $N$ and the depth $L$.

We assume that the teacher is a 'quenched-random teacher': the set of the synaptic weights $\{(J_\blacksquare^k)_{\text{teacher}}\}$ of the teacher machine are iid random variables which obey the normalization Eq. (2). Such a teacher machine is subjected to a set of random inputs, which are iid random variables, $\mathbf{S}_{0,i} = (S_{0,i}^1, S_{0,i}^2, \ldots, S_{0,i}^M)$ for $(i = 1, 2, \ldots, N)$ and produces the corresponding set of outputs,

$$(\mathbf{S}_{L,i})_{\text{teacher}} = ((S_{L,i}^1)_{\text{teacher}}, (S_{L,i}^2)_{\text{teacher}}, \ldots, (S_{L,i}^M)_{\text{teacher}}). \tag{12}$$

The task of the student machine is to try to infer the synaptic weights $\{(J_\blacksquare^k)_{\text{teacher}}\}$ of the teacher machine, by adjusting its own synaptic weights $\{(J_\blacksquare^k)_{\text{student}}\}$ such that it successfully reproduces all the outputs of the teacher $(\mathbf{S}_{L,i})_{\text{teacher}}$ starting from the same input data as the teacher.

Note that the student is given the full information of the input $\mathbf{S}_{0,i}$ and the output of the teacher $(\mathbf{S}_{L,i})_{\text{teacher}}$ plus full information on the architecture of the teacher. In the context of statistical inference, this is an idealized situation called as Bayes optimal case [8] and we limit ourselves to this in the present paper for simplicity.

## 3 Replica theory

Now let us formulate a replica approach to study the solution space of the deep neural network. To study the case of random inputs/outputs (sec. 2.2.1) we consider $n$ replicas $a = 1, 2, \ldots, n$ which are independent machines subjected to the common set of inputs $\mathbf{S}_{0,i}$ and outputs $\mathbf{S}_{L,i}$ for $i = 1, 2, \ldots, N$. For the case of the teacher-student setting (sec. 2.2.2) we consider $n = 1 + s$ replicas, with the replica $a = 0$ to represent the teacher machine and other replicas $a = 1, 2, \ldots, s$ to represent the replicas of the student.

### 3.1 Order parameters

For the setting with random inputs/outputs, which is a constraint satisfaction problem, we anticipate that that the solution space exhibits clustering (glass transition) as we noted in the introduction. Thus it is natural to consider order parameters that detect the glass transitions. Given the dense connections of the network, we naturally introduce 'local' glass order parameters (see [20]),

$$Q_{ab,\blacksquare} = \frac{1}{N}\sum_{i=1}^{N}(J_{\blacksquare}^{i})^{a}(J_{\blacksquare}^{i})^{b}, \qquad q_{ab,\blacksquare} = \frac{1}{M}\sum_{\mu=1}^{M}(S_{\blacksquare}^{\mu})^{a}(S_{\blacksquare}^{\mu})^{b}. \tag{13}$$

Note that the normalization condition for the bonds Eq. (2) and the spins (which take Ising values $\pm 1$) implies $Q_{aa,\blacksquare} = q_{aa,\blacksquare} = 1$.

For the teacher-student setting, we continue to use the above order parameters for $a = 0, 1, 2, \ldots, s$ replicas where 0-th replica is for the teacher machine. Thus $Q_{0a} = Q_{a0}$ and $q_{0a} = q_{a0}$ for $a = 1, 2, \ldots, s$ represent the overlap between the teacher machine and student machines.

There are two comments regarding some trivial symmetries left in the system. First, the system is symmetric under permutations of the labels put on the data $\mu = 1, 2, \ldots, M$. The labels put on different replicas could be permuted differently. In the 2nd equation of Eq. (13) it is assumed that all replicas follow the same labels breaking this permutation symmetry. Second, the system is symmetric under permutations of perceptrons $\blacksquare$ within the same layer and the permutations could be done differently on different replicas. In Eq. (13), this permutation symmetry is also broken. Note that solutions with other permutations regarding the two symmetries mentioned above give exactly the same free-energy so that one choice is enough.

### 3.2 Replicated Gardner volume

The Gardner's volume Eq. (6) fluctuates depending on the realizations of the boundaries $\mathbf{S}_0$ and $\mathbf{S}_L$. In the present paper we wish to analyze the *typical* behavior for stochastic realizations of the boundaries. To this end we consider the replicated phase space volume (the Gardner volume),

$$
\begin{aligned}
V^{n}(\mathbf{S}_0, \mathbf{S}_L) &= e^{NM\mathcal{S}_n(\mathbf{s}_0,\mathbf{s}_l)} \\
&= \prod_{a=1}^{n}\left(\prod_{\blacksquare}\mathrm{Tr}_{\mathbf{J}_{\blacksquare}^{a}}\right)\left(\prod_{\blacksquare\backslash\text{output}}\mathrm{Tr}_{\mathbf{S}_{\blacksquare}^{a}}\right)\left\{\prod_{\mu,\blacksquare,a}e^{-\beta V(r_{\blacksquare,a}^{\mu})}\right\},
\end{aligned}
\tag{14}
$$

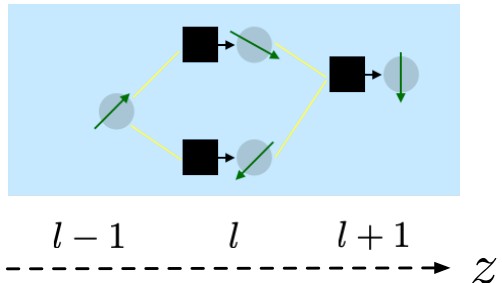

$$l-1 \qquad l \qquad l+1$$

$$\cdots\cdots\cdots\cdots\cdots\cdots\cdots\cdots\longrightarrow z$$

Figure 2: A loop of interactions in a DNN extended over 3 layers, through 3 perceptrons and 4 bonds. We neglect effects of such loops (and more extended ones) in our theory.

with

$$r^{\mu}_{\blacksquare,a} \equiv (S^{\mu}_{\blacksquare})^a \sum_{i=1}^{N} \frac{(J^i_{\blacksquare})^a}{\sqrt{N}} (S^{\mu}_{\blacksquare(i)})^a. \tag{15}$$

The *typical* behavior can be studied by considering the $n \to 0$ limit [36], i.e. $\partial_n \overline{V^n(\mathbf{S}_0, \mathbf{S}_L)}^{\mathbf{S}(0),\mathbf{S}(L)}\big|_{n=0}$ where the overline represents the average over the different realizations of the boundaries (see below for the details.)

As shown in appendix A, following similar steps as in [20], we obtain the replicated free-entropy functional $s_n(\{Q_{\blacksquare}, q_{\blacksquare}\})$ in terms of the order parameters $Q_{\blacksquare}$ and $q_{\blacksquare}$ defined in Eq. (13) in the limit $N, M \to \infty$ with fixed $\alpha = M/N$. For simplicity, we limit ourselves to a tree-approximation which neglects the effects of interaction-loops along the $z$-axis such as the one shown in Fig. 2. The tree approximation has two essential problems: 1) it cannot describe faithfully 1-dimensional fluctuations along $z$-axis 2) it misses microscopic details close to the boundaries where we naturally expect inhomogeneities. Especially it fails to capture the difference of the two opposite boundaries. In principle, this accidental symmetry can be removed taking into account loop-corrections. Indeed the loop shown in Fig. 2 is not symmetric with respect to the interchange of the left and right hand sides.

Given the structure of the network (see Fig. 1), it is natural to assume that order parameters are uniform within each layer $l = 0, 1, 2, \ldots, L$,

$$Q_{ab,\blacksquare} = Q_{ab}(l), \qquad q_{ab,\blacksquare} = q_{ab}(l). \tag{16}$$

To represent the quenched boundaries, we impose the boundary conditions on the inputs/outputs layers by simply putting $q_{ab}(0) = q_{ab}(L) = 1$ (see below).

The above general formulation can be adapted for the two scenarios introduced in sec. 2.2 as follows,

- Random inputs/outputs

  In the case of random inputs/outputs (sec. 2.2.1) we consider the free-energy functional,

$$\frac{-\beta F[\{\hat{Q}(l), \hat{q}(l)\}]}{NM} = \frac{\partial_n \overline{V^n(\mathbf{S}_0, \mathbf{S}_L)}^{\mathbf{S}_0, \mathbf{S}_L}\big|_{n=0}}{NM} = \partial_n s_n[\{\hat{Q}(l), \hat{q}(l)\}]\big|_{n=0}. \tag{17}$$

  The presence of the imposed random inputs/outputs can be specified by providing values of $q_{ab}(0)$ and $q_{ab}(L)$. Since all replicas are subjected to the same inputs and outputs, we can simply set,

$$q_{ab}(0) = q_{ab}(L) = 1. \tag{18}$$

As we discuss later we will also consider the case of fluctuating boundary conditions.

- Teacher-student setting

  In the case of the teacher-student setting (sec. 2.2.2) we consider instead the so called Franz-Parisi potential [37],

$$
\frac{-\beta F_{\text{teacher-student}}[\{\hat{Q}(l), \hat{q}(l)\}]}{NM} = \frac{\partial_s \overline{V^{1+s}(\mathbf{S}_0, \mathbf{S}_L(\mathbf{S}_0, \mathcal{J}_{\text{teacher}})))}^{\mathbf{S}_0, \mathcal{J}_{\text{teacher}}}\Big|_{s=0}}{NM}
$$
$$
= \partial_s s_{1+s}[\{\hat{Q}(l), \hat{q}(l)\}]\Big|_{s=0}, \tag{19}
$$

where the over-line denotes the average over the imposed random inputs imposed commonly on both the teacher and student machines. The outputs are just those of the teacher machine $a = 0$, $\mathbf{S}_L(\mathbf{S}_0, \mathcal{J}_{\text{teacher}})$ which are of course functions of the inputs $\mathbf{S}_0$ and the synaptic weights of the teacher machine $\mathcal{J}_{\text{teacher}} = \{(J_{\blacksquare}^k)_{\text{teacher}}\}$. Since both the teacher and student machines are subjected to the same inputs, we set,

$$
q_{ab}(0) = 1 \tag{20}
$$

for $a, b = 0, 1, \ldots, s$. In addition, since the outputs of the student machine are forced to agree perfectly with that of the teacher machine we set,

$$
q_{ab}(L) = 1 \tag{21}
$$

for $a, b = 0, 1, \ldots, s$.

### 3.3  Random inputs/outputs

Now we analyze the case of random inputs/outputs introduced in sec. 2.2.1 by the replica theory using the Parisi's ansatz explained in sec. A.5.1.

We assume the Pairisi's ansatz with $k$-step RSB (see sec. A.5) for the order parameters of the bonds $Q_i(l)$ for $l = 1, 2, \ldots, L$ and spins $q_i(l)$ for $l = 1, 2, \ldots, L-1$ which characterize the Parisi's matrices (see Fig. 19). We solve the saddle point equations numerically to obtain the glass order parameters as described in sec. B.3.3. For $i = 0, 1, 2, \ldots, k$ we have parameter $m_i$ (see Eq. (81)). In the $k \to \infty$ limit, $Q_i(l)$s become continuous functions $Q(x, l)$ which can be well approximated by $Q_i(l)$ plotted vs $m_i$ for large enough $k$ (See Fig. 19 d)). The same holds for the order parameter of spins $q_i(l)$s, i. e. we obtain continuous functions $q(x, l)$ in $k \to \infty$ limit. From the functions $Q(x, l)$ and $q(x, l)$, we can obtain the overlap distribution functions $P(q, l)$ and $P(Q, l)$ (see Eq. (82)). The boundary condition Eq. (18) (see Fig. 3) amounts to set,

$$
q_0(0) = q_0(L) = 1, \tag{22}
$$
$$
q_i(0) = q_i(L) = 0 \qquad (i = 1, 2, \ldots, k). \tag{23}
$$

In the following we present results using $k = 100$ step RSB and the depth of the system $L = 5 - 20$. Because of the tree-approximation and the choice of the boundary condition, the system becomes symmetric with respect to reflections at the center: we confirmed that the solutions satisfy $q_i(l) = q_i(L-l)$ and $Q_i(l) = Q_i(L-l)$.

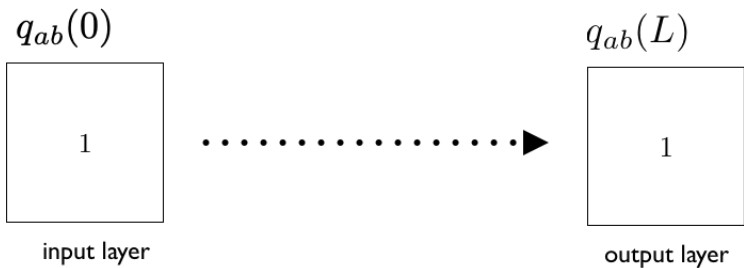

Figure 3: "Quenched" boundary

### 3.3.1 Liquid phase

For small $\alpha = M/N$ we find the whole system is in the liquid phase where the glass order parameters are all zero: for $i = 1, 2, \ldots, k$ $q_i(l) = 0$ ($l = 1, 2, \ldots, L-1$) and $Q_i(l) = 0$ ($l = 1, 2, \ldots, L$). This means that the parameter space is so large that there are simply too many solutions compatible with the constraints. Here the replica symmetry is not broken. This means that the solution space looks like a giant continent in which all typical solutions are continuously connected to each other.

### 3.3.2 The 1st glass transition

With increasing $\alpha$, the system becomes more constrained. We find a continuous (2nd order) glass transition at $\alpha_g(1) \simeq 2.03$ on the 1st layers $l = 1, L-1$ just beside the "quenched" inputs/outputs boundaries as shown in Fig. 4 a). The emergence of the finite glass order parameters signals that the solution space is shrinking there. The rest of the system ($l = 2, 3, \ldots, L-2$) remains in the liquid phase $q_{\text{EA}}(l) = Q_{\text{EA}}(l) = 0$ at this stage. As shown in Fig. 4 b) the Edwards-Anderson (EA) order parameters of the spins $q_{\text{EA}}(l) = q_k(l)$ and bonds $Q_{\text{EA}}(l) = Q_k(l)$ at the 1st layer $l = 1$ grow continuously across the critical point $\alpha_g(1)$. Exactly the same happens on the other side at $l = L-1$. The fact that the glass transition takes place in a *continuous* way, is different from the random first-order transition (RFOT) in structural glass models [9–11, 38–41].

Since the transition is a 2nd order transition, the liquid sate $(Q, q) = (0, 0)$ becomes unstable and a glass state can emerge smoothly at the transition. Then what would play the role of symmetry breaking field (see sec. A.1.3) to pick up a particular glass state out of many candidates? In the learning dynamics, the random inputs/outputs data imposed at the boundaries ($l = 0$ and $l = L$) and choices of the initial condition for learning will play the role of the symmetry breaking field.

The fact that the glassy regions emerge next to the boundaries is reasonable because the effect of constraints should be strongest there. The situation does not change even in the limit $L \to \infty$ where the two boundaries are infinitely separated. But this may appear bizarre. Why specification of the just the initial condition or finial condition for the dynamics Eq. (1) can constrain the 1st layers ($l = 1, L-1$) so much? With such a huge liquid-like region left in the bulk, any information starting from the input layer will be completely randomized before reaching the output layer. Here let us remind ourselves that we are considering statistical mechanics of the solution space which is like the statistical mechanics of hard-spheres as we noted below Eq. (11). The reason for the glass transition on the 1st layers is an entirely entropic reason: a certain set of configurations of the bonds in the 1st layers ($l = 1, L-1$) allow exceedingly larger fluctuation in the hidden layers compared with others so that they dominate the entropy of the solution space. In this sense it is a glass version of entropy-driven

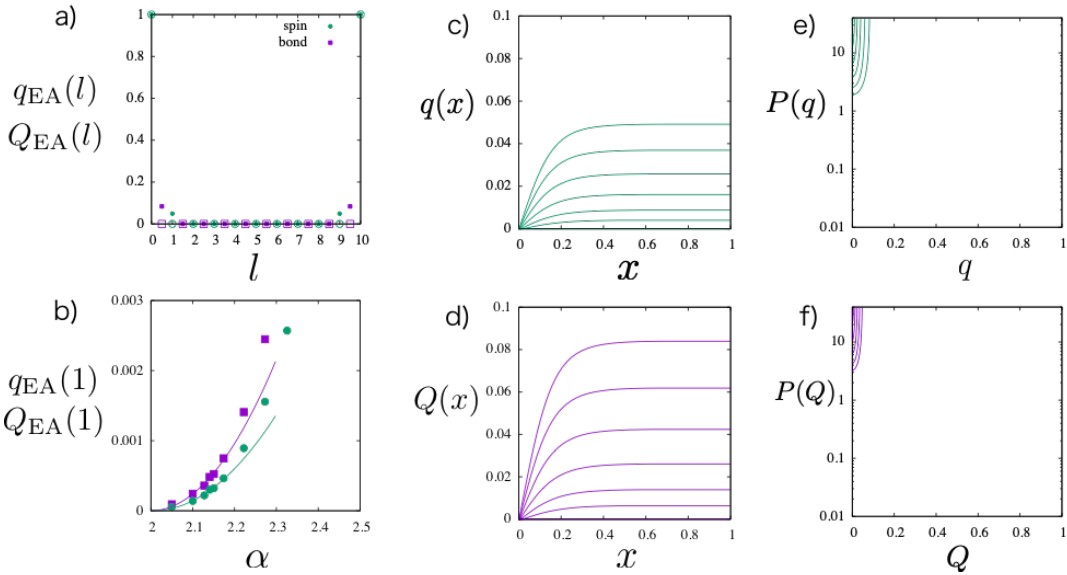

Figure 4: The 1st glass transition: a) spatial profile of the Edwards-Anderson (EA) order parameter for spins $q_{\text{EA}}(l)(= q_{k=100}(l))$ and bonds $Q_{\text{EA}}(l)(= Q_{k=100}(l)$ slightly before $\alpha = 2.0$ (empty symbols)/after $\alpha = 3.125$ (filled symbols) the 1st glass transition. The depth is $L = 10$ in this example. b) Evolution of the EA order parameters $q_{\text{EA}}(1) = q_k(1)$ and $Q_{\text{EA}}(1) = Q_k(1)$ at the 1st layer after passing the critical point of the 1st glass transition $\alpha_g(1) \simeq 2.03$. c),d) Glass order parameter function $q(x, l)$ for spins and $Q(x, l)$ for bonds at the 1st layer $l = 1$ at around the 1st glass transition. Here $\alpha = 3.13, 2.94, 2.78, 2.63, 2.50, 2.38$ from the top to the bottom. e),f) the overlap distribution function of spins $P(q) = dx(q)/dq$ and bonds $P(Q) = dx(Q)/dQ$ (see Eq. (82)).

ordering like the crystallization of hard-spheres (Alder transition) [42] and order-by-disorder transitions oftenly observed in frustrated magnets [43].

As shown in Fig. 4 c),d), the functions $q(x, l)$ an $Q(x, l)$ at the 1st layers $l = 1, L − 1$ are continuous functions of $x$ with plateaus at $q_{\text{EA}}$ and $Q_{\text{EA}}$ for some range $x_1(\alpha) < x < 1$ with $x_1(\alpha)$ decreasing with $\alpha$. Thus the replica symmetry is fully broken much as in the SK model for spin-glasses [13, 14]. Correspondingly the overlap distribution functions Eq. (82) $P(q) = dx(q)/dq$ and $P(Q) = dx(Q)/dQ$ shown in Fig. 4 e),f), exhibit delta peaks at $q = q_{\text{EA}}$, $Q = Q_{\text{EA}}$ plus non-trivial continuous parts extending down to $q = 0$ and $Q = 0$.

The RSB means that the solution space is now clustered, i. e. the giant continent of the solutions is split into mutually disconnected islands. The EA order parameters $q_{\text{EA}}$ and $Q_{\text{EA}}$ represent the size of the islands, i. e. larger EA order parameters mean smaller islands. The probability that two solutions sampled in equilibrium belong to the same island is given by $1 − x_1(\alpha)$. The continuously changing part of the functions $Q(x)$ and $q(x)$ in the range $0 < x < x_1(\alpha)$ means that the islands or clusters are organized into meta-clusters, meta-meta-clusters,... in a hierarchical way: the mutual overlap (distance in the phase space) between the islands is ultrametric [14–17, 44]. In general, the continuous RSB phase is marginally stable [14, 27, 45, 46].

The strong spatial heterogeneity of the glass order parameters is striking. It means that the solution space is clustered in the 1st layers ($l = 1, L − 1$) but the islands of solutions merge into a big continent in the rest of the system which remains in the liquid phase. The spatial heterogeneity is very interesting from the algorithmic point of view since this implies the learning dynamics is fast except next to the boundaries. Moreover, it is tempting to speculate

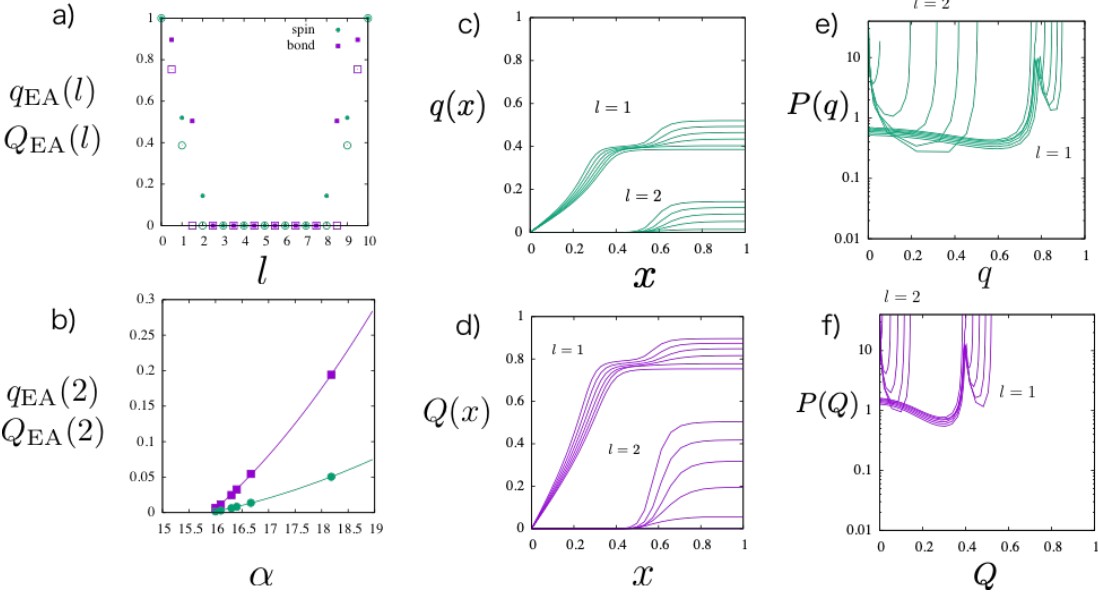

Figure 5: The 2nd glass transition: a) spatial profile of the Edwards-Anderson (EA) order parameter for spins $q_{\mathrm{EA}}(l)$ and bonds $Q_{\mathrm{EA}}(l)$ slightly before $\alpha = 15.38$ (empty symbols)/after $\alpha = 25$ (filled symbols) the 2nd glass transition. The depth is $L = 10$ in this example. b) Evolution of the EA order parameters $q_{\mathrm{EA}}(2)$ and $Q_{\mathrm{EA}}(2)$ at the 2nd layer after passing the critical point of the 2nd glass transition $\alpha_{\mathrm{g}}(2) \simeq 15.9$. c),d) Glass order parameter function $q(x,l)$ for spins and $Q(x,l)$ for bonds at the 1st and 2nd layers at around the 2nd glass transition. Here $\alpha = 25.0, 22.2, 20.0, 18.2, 16.7, 15.4$ from the top to the bottom. e),f) the overlap distribution function of spins $P(q) = dx(q)/dq$ and bonds $P(Q) = dx(Q)/dQ$ Eq. (82).

that the first dynamics in the liquid region will assist the equilibration of the glassy regions close to the boundaries.

### 3.3.3 The 2nd glass transition

Increasing $\alpha$ further we meet another glass transition at $\alpha_{\mathrm{g}}(2) \simeq 15.9$ by which the 2nd layers $l = 2, L-2$ become included in the glass phase while the rest of the system $l = 3, 4, \ldots, L-3$ still remains in the liquid phase as shown in Fig. 5 a). The glass phase has grown one step further into the interior. The transition is again a continuous one as can be seen in Fig. 5 b) where we display the EA order parameters $q_{\mathrm{EA}}(l) = q_k(l)$ and $Q_{\mathrm{EA}}(l) = Q_k(l)$ at $l = 2$. Exactly the same happens on the other side at $l = L-2$.

As shown in Fig. 5 c),d), the functions $q(x,l)$ an $Q(x,l)$ at the 2nd layers $l = 2, L-2$ are continuous functions of $x$ with plateaus at $q_{\mathrm{EA}}(2)$ and $Q_{\mathrm{EA}}(2)$ in some range $x_2(\alpha) < x < 1$ with $x_2(\alpha)$ decreasing with $\alpha$. A marked difference to the case of the 1st glass transition which happened at the 1st layers $l = 1, L-1$ is that the order parameters become finite only in some range $x_2(\alpha) \lessapprox x < 1$. As a result, it looks approximately like a step function with the step located at $x_2(\alpha)$. As shown in Fig. 5 e),f), this amounts to induce a delta peak not only at $q_{\mathrm{EA}}(2)$ ($Q_{\mathrm{EA}}(2)$) but also at $q = Q = 0$ in the distribution of the overlaps. In a sense the solution is approximately like one step RSB in the random energy model [47] or models for structural glasses [9–11, 38–41] if we neglect the smoothing part of the step like function. This means that, roughly speaking, the solution space in the 2nd layers are split into islands

that are completely dissimilar from each other. Two solutions sampled in equilibrium, in the 2nd layers, belong to the same island whose size is represented by $Q_{EA}(2)$ and $q_{EA}(2)$ with probability $1-x_2(\alpha)$. Otherwise, they belong to different islands which are very far from each other.

Remarkably, the 2nd glass transition induces another continuous glass transition on the 1st layers $l = 1, L-1$ which were already glassy. Physically, this is natural because the 1st layers are now more constrained than before having two glassy neighbors while they had just one glassy neighbor before. As can be seen in Fig. 5 c),d), an internal step-like structure emerges continuously within the region where the glass order parameter was flat $x_1(\alpha) < x < 1$ before the 2nd glass transition. As shown in Fig. 5 e),f), the emergence of the internal step amounts to a continuous splitting of the delta peak at $q_{EA}(1)$ ($Q_{EA}(1)$) into two peaks (plus a continuous part in between) meaning that the glass phase has become more complex. This means the smallest bundles or islands of the solutions have been split into multiple sub-bundles. In a sense, this is similar to the Gardner transition found originally in Ising $p$-spin spin-glass models [48] and in the hard-sphere glass in large-dimensional limit [10, 11, 26, 46].

We could say that the situation in the 1st layers is roughly like a 2 step RSB: if we neglect the smoothing parts, the functions $Q(x)$ and $q(x)$ look approximately like functions with two steps, one at $x_1(\alpha)$ and the other at $x_2(\alpha)$. This means that two solutions sampled in equilibrium, in the 1st layers, belong to the same island whose size is represented by $1-Q_{EA}(1) = 1-Q(x_2(\alpha),1)$ and $1-q_{EA}(1) = 1-q(x_2(\alpha),1)$ with probability $1-x_2(\alpha)$. Otherwise they belong to different islands. However, with a larger probability $1-x_1(\alpha)$, they belong at least to the same meta-cluster of islands whose size is represented by $1-Q(x_1(\alpha),1)$ and $1-q(x_1(\alpha),1)$ which are larger than $1-Q_{EA}$ and $1-q_{EA}$.

After the 2nd glass transition, the glass order parameters have become more heterogeneous in space. Interestingly the internal step of the glass order parameters on the 1st layers $l = 1, L-1$ is located around $x_2(\alpha)$ being synchronized with the step on the 2nd layers $l = 2, L-2$. This means that two solutions sampled in equilibrium belong to the same island in the 1st and 2nd layers with the *same* probability $1-x_2(\alpha)$. This implies that the same bundle of solutions continue in the 1st and 2nd layers. Since the EA order parameters are bigger in the 1st layers, the bundle becomes more spread out in the 2nd layers than in the 1st layers. The bundles are grouped into meta-bundles in the 1st layer which becomes dissociated in the 2nd layers. Finally, all bundles become dissociated and merge into a gigantic liquid continent after the 3rd layers. The two-step dissociation of the bundles of solutions is quite interesting in the context of learning.

### 3.3.4 More glass transitions

Now it is easy to imagine that glass phase will grow further invading the liquid phase by increasing $\alpha$ more. As we show in Fig. 6, this is indeed the case. We observe that the glass transition point $\alpha_g(l)$ of the $l$-th layer (and $L-l$ the layer) grows very rapidly, exponentially fast with $l$ as shown in Fig. 7,

$$\alpha_g(l) \sim 2.7(3)e^{1.03(2)l}. \tag{24}$$

In other words, the 'penetration depth' of the glass phase $\xi_{glass}$ grows very slowly with $\alpha$ as,

$$\xi_{glass}(\alpha) \sim \ln \alpha. \tag{25}$$

The results shown in this section is done on systems with $L = 20$ which is still larger than of $2\xi_{glass}(\alpha) \sim 18$ of $\alpha = 4000$ which is the largest $\alpha$ used in this section. Note that the system is under-parametrized, in the sense that the size of the data Eq. (3) is smaller than that of the parameters Eq. (4), only for $\alpha < 20$. However once the liquid phase is present at the center,

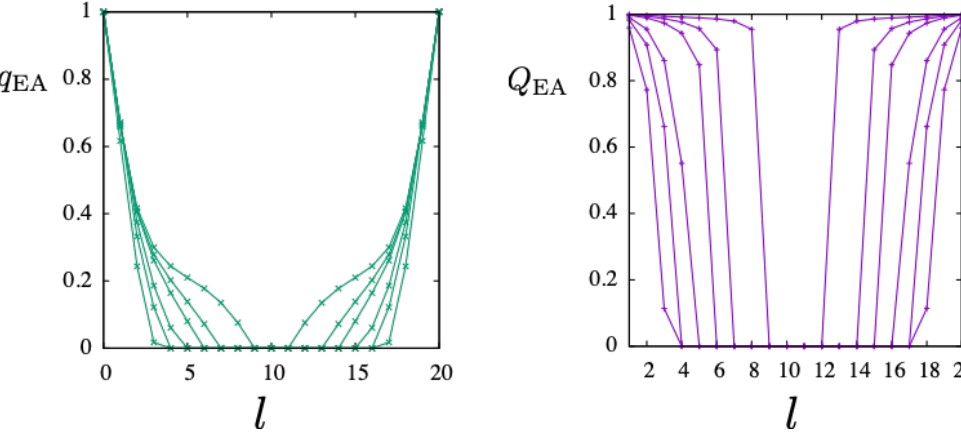

Figure 6: The spatial profile of the EA order parameters $q_{\text{EA}}(l) = q_k(l)$ and $Q_{\text{EA}}(l) = Q_k(l)$ at $\alpha = 50, 100, 200, 1000, 2000, 4000$. Here $L = 20$.

the solution for the glass phase does not change with larger $L$. So that the results presented in this section are essentially in the situation of over parametrization (for typical instances).

The exponential growth of the glass transition point $\alpha_g(l)$ with the depth $l$ implies that the storage capacity $\alpha_j(L)$, which should be greater than $\alpha_g(L)$ by definition, also grow exponentially fast with the depth $L$,

$$\alpha_j(L) \propto e^{\text{const}L}. \tag{26}$$

This is surprising because the worst case scenarios [30] would predict linear growth with $L$. This means the behavior of typical instances are very different from the worst ones in the DNN. Here it is instructive to recall the case of the single perceptron. The storage capacity of typical instances computed by the replica method [5] is $\alpha_j = 2$. The existence of solutions are guaranteed for all instances including the worst ones in the range $0 < \alpha < 1$ while there are exponentially rare $e^{-\text{const}N}$ samples which lacks solutions in the range $1 < \alpha < 2$ [49]. Our result implies the gap between the worst and typical ones become much more enhanced in deeper systems $L > 1$. This may be related to the so-called exponential expressivity [32]. The latter is due to the chaos effect of DNNs with non-linear activation functions like Eq. (1): trajectories starting from slightly different initial condition deccorrelate exponentially with the depth $l$. Perhaps this helps building a mapping (function) between the imposed input and output spin configurations, which are totally different, by a limited depth.

As $\alpha$ increases, the allowed phase space volume becomes suppressed. In Fig. 8 we display $x(Q, l) = \int_0^Q dQ P(Q, l)$ and $x(q, l) = \int_0^Q dq P(q, l)$ (see Eq. (82)). The latter is the probability that two replicas (two machines learning independently) subjected to the same inputs/outputs have a mutual overlap of the bonds (spins) at $l$-th layer smaller than $Q$ ($q$). As can be seen in the figure, the probability appears to decay as $1/\sqrt{\alpha}$ for all $l$, $Q$ and $q$. This implies two independently learning machines become more and more similar as the number of constraints increases.

We note however that the EA order parameter of the spins $q_{\text{EA}}(l)$ shown in Fig. 6 remain significantly smaller than that of the bonds $Q_{\text{EA}}(l)$. Apparently, it implies that even in the jamming limit where $Q_{\text{EA}}(l) \to 1^-$, $q_{\text{EA}}(l)$ does not reach 1. A possible reason is the chaos effect. As we noted before, trajectories of random perceptron network with non-linear activation functions are known to show chaotic behavior under infinitesimal changes on the input boundary [31, 32]. We confirmed it is always the case for the present model with the 'sgn' activation function Eq. (1) [33]. Moreover the system also shows a chaotic response against infinitesimal

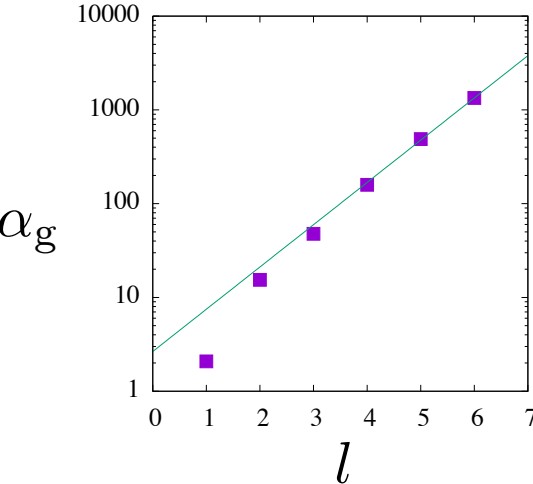

Figure 7: The glass transition point $\alpha_g(l)$ of internal layers. This is obtained by numerical analysis of the saddle point solutions. The solid line is the exponential fit Eq. (24).

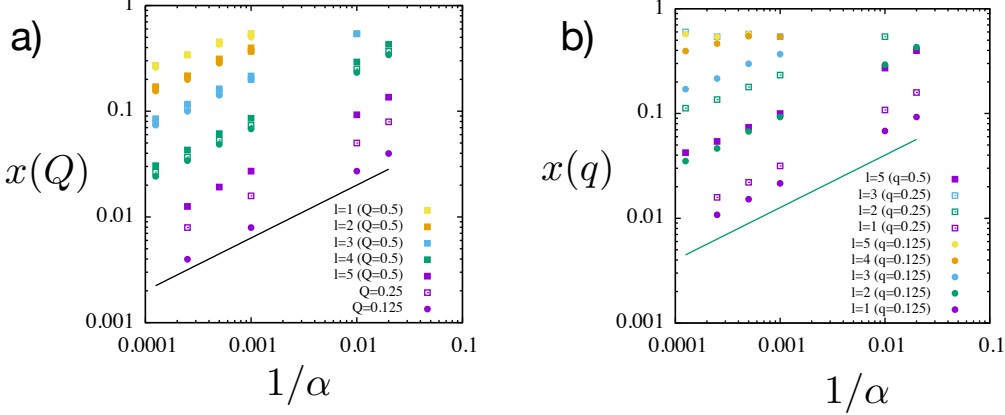

Figure 8: Decay of $x(Q)$ and $x(q)$ with increasing $\alpha$. Here values of $x(Q,l)$, which is the inverse function of $Q(x,l)$ and $q(x,l)$, are shown at various $Q$, $q$ and layers $l$. The slope of the straight line is $1/2$.

changes made on the bonds [33]. Thus even in $Q_{\mathrm{EA}}(l) \to 1^-$ limit, the spin configuration can fluctuate significantly.

As shown in Fig. 9, the glass order parameter functions become quite complex at large values of $\alpha$. Closer to the boundaries, the system has experienced larger numbers of successive glass transitions that leave behind river-terrace-like structure with many steps in the glass order parameter functions. This means distribution functions of the overlap with many delta peaks. The steps of the glass order parameter functions at different layers appear to be aligned with each other.

Now let us summarize the essential features of the glass order parameters $q(x,l)$ and $Q(x,l)$ shown in Fig. 4, 5 and 9. The essence of the river-terrace-like glass order parameter functions can be sketched schematically as shown in Fig. 10 a). Here we have simplified the picture representing the functions by staircases neglecting their rounding. Comparing the river-terraces at different layers we notice an interesting feature that the steps at different lay-

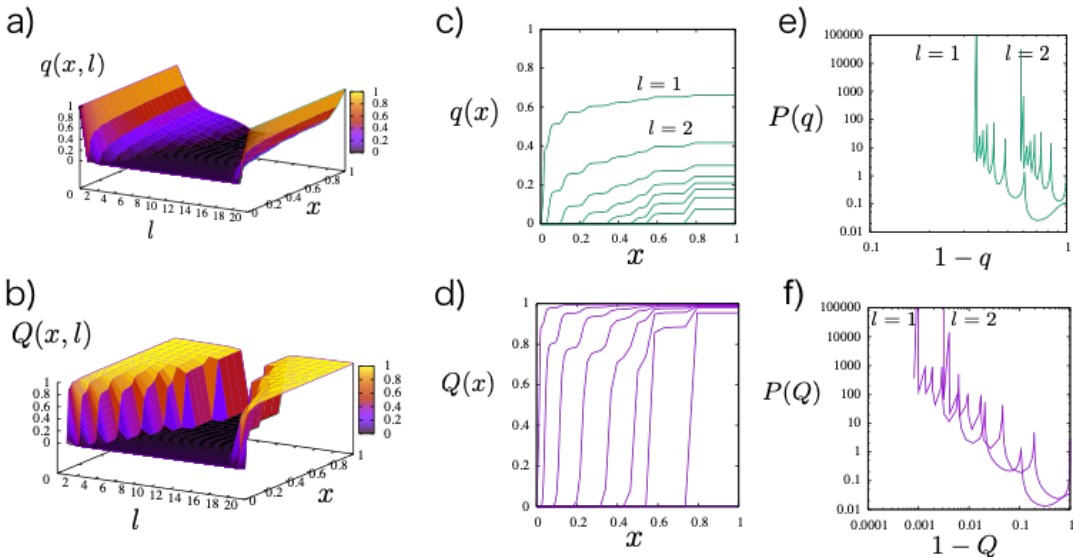

Figure 9: Glass order parameter functions under stronger constraints $\alpha = 4000$. In this example $L = 20$ and the central layers at $l = 8, 9$ still remain in the liquid phase. a),b) 3 dimensional plots of $q(x,l)$ and $Q(x,l)$. c),d) the same in 2 dimensional plots. e,f) the corresponding overlap distribution functions Eq. (82) at $l = 1, 2$ (for clarity others at $l = 3, 4, \dots$ are not shown).

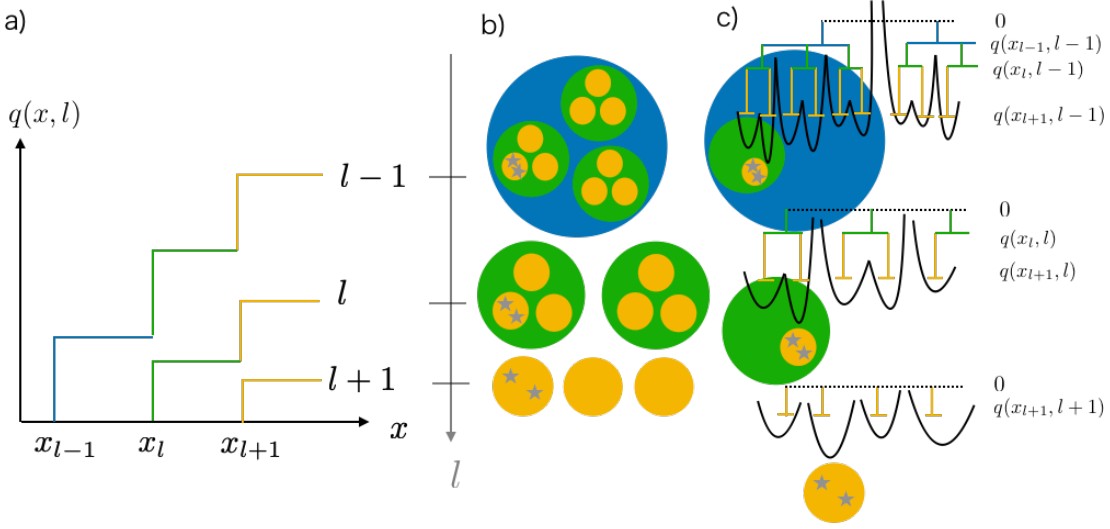

Figure 10: River-terrace like glass order parameter and its implications: a) schematic picture of the river-terrace-like glass order parameter $q(x)$ (or $Q(x)$) b) hierarchical clustering of replicas c) schematic free-energy landscape and trees representing ultrametric organization of overlaps between meta-stable states.

ers are synchronized: they are all located exactly at the same positions, $\dots, x_{l-1}, x_l, x_{l+1}, \dots$. The river-terraces reflect successive glass transitions in the following way. At the $n$-th glass transition, a finite glass order parameter emerges continuously in the interval $x_n(\alpha) < x < 1$ at the $n$-th (and $(L-n)$-th) layer. The glass order parameter functions at layers between the

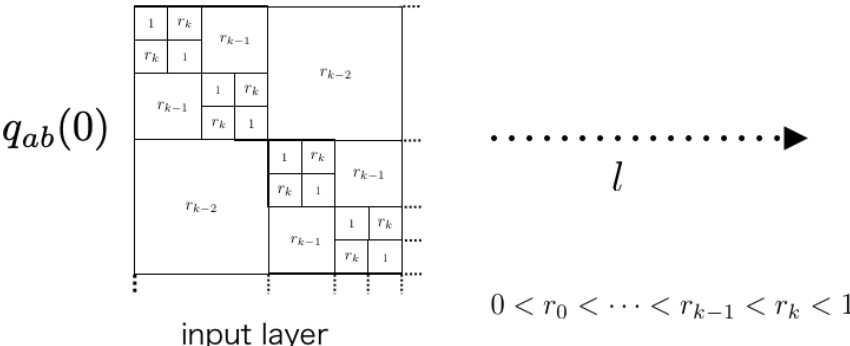

Figure 11: Fluctuating input layer with hierarchical overlap structure

$n$-th layer and the boundary, those at $l = 1, 2, \ldots, n-1$ (and the corresponding layers on the other side), which are already in the glass phase, acquire additional steps in the same interval $x_n(\alpha) < x < 1$. At a given $\alpha$, the layers included in the glass phase are $l = 1, 2, \ldots, n$ (and the corresponding ones on the other side) where $n$ is such that $\alpha_g(n) < \alpha < \alpha_g(n+1)$. Due to the successive glass transitions $1, 2, \ldots, n$, the $l$-th (and $L-l$ th) layer with $1 \leq l \leq n$ have a series of steps at $0 < x_l(\alpha) < x_{l+1}(\alpha) \ldots < x_n(\alpha) < 1$. Correspondingly the overlap distribution functions $P(Q, l)$ and $P(q, l)$ exhibit a series of delta peaks at $q(x_l, l) < q(x_{l+1}, l) \ldots < q(x_n, l)$ plus another delta peak at $q = 0$ for $2 \leq l \leq n$.

The river-terrace-like glass order parameter function $q(x, l)$ (and $Q(x, l)$) in Fig. 10 a), means spatial evolution of the hierarchical clustering of the solutions as shown schematically in Fig. 10 b). In panel b) clusters (and meta-clusters) of the same color represent those associated with a common value of $x$. Recalling the probabilistic meaning of $x$, it is natural to assume such a cluster represents a bundle of solutions that go together through different layers. Sampling two solutions in equilibrium, the two belong to such a common cluster with probability $1 - x$. The size of a cluster represents spreading of the solutions $1 - q(x, l)$, i.e. typical distance between the solutions belonging to the same cluster, which increases with decreasing $x$ and/or going away from the boundary $l = 1, 2, \ldots$. (Meta-)clusters with smaller $x$ represent those at a higher level in the hierarchy which includes sub-clusters associated with larger values of $x$. Going deeper into the bulk starting from the boundary, those clusters with smaller $x$ dissociate earlier.

This, in turn, implies the hierarchical free-energy landscape with basins, meta-basins,... which evolves in space as shown schematically in Fig. 10 c). The free-energy landscape evolves in space in such a way that it progressively becomes less complex and flatter as we go deeper into the interior. For a given $\alpha$, the penetration depth $\xi_{\text{glass}}(\alpha) \sim \ln \alpha$ is finite. So that in a deep enough network $L/2 > \xi_{\text{glass}}(\alpha)$, the interior remains in the liquid phase. Moreover, the fact that the river-terraces of the glass order parameter functions at different layers are synchronized to each other with common positions of the steps at $x_1 < x_2 \ldots$, suggests that the basic backbone structure of the free-energy landscape is preserved (but renormalized) moving away from the boundaries. It is tempting to speculate that these features have important consequences on learning in deep neural networks.

## 3.4 Fluctuating boundary

To obtain some further insights, we next analyze the case of fluctuating boundary: spin configurations on the boundaries are allowed to fluctuate during learning following certain probability distributions. Here we consider cases such that the overlap distribution of the spins

on the input layer ($l = 0$) exhibit a hierarchical structure as parametrized in the form of the Parisi's matrix Eq. (78) (Fig. 11). There are two motivations for this analysis:

- The perturbation may provide some hints on the stability of the characteristic free-energy landscape of the DNN we found above. Given that random neural networks are typically chaotic with respect to changes made on the inputs [32,33], it is very interesting to know how training make differences.

- In a typical setting of unsupervised learning, one would be interested with the probability distributions $P(\mathbf{S}_l)$ of hidden variables $\mathbf{S}_l$ ($l = 1, 2, \ldots$) when variables on the input boundary $\mathbf{S}_0$ are forced to obey some probability distribution $P(\mathbf{S}_0)$.

### 3.4.1 One RSB type boundary

Here we consider the simplest case of '1RSB'.

$$q_i(0) = \begin{cases} r & m_i < x_{\text{input}}, \\ 1 & m_i > x_{\text{input}}. \end{cases}$$

This means the system subjected to a slightly different input data instead of the original one, which has overlap $0 < r < 1$ with respect the original input data, from time-to-time with some small probability $x_{\text{input}}$.

It can be seen in Fig. 12 that the effect of the perturbation is strong only at $x < x_{\text{input}}$. This means that the trained system is not simply chaotic but the hierarchical organization in the solution space has a certain degree of robustness against perturbations on the inputs (as well as outputs).

### 3.4.2 Full RSB type boundary

Let us next consider the 'full RSB' case. More specifically we consider the simplest full RSB structure in the input layer,

$$q_i(0) = \min(am_i, 1), \tag{27}$$

with a certain constant $a > 0$. Thus $q(x, 0)$ function consists of two parts: 1) 'continuous part' $q(x, 0) = ax$ with slope $a$ in the interval $0 < x < 1/a$ and 2) 'plateau' $q(x, 0) = q_{\text{EA}}(0) = 1$ in the interval $1/a < x < 1$.

We analyze the saddle point solutions numerically as before (see sec. B.3.3). In the following, we present results using $k = 100$ step RSB and the depth of the system $L = 20$. We chose $1/a = 0.8$. As shown in Fig. 13, the glass phase grows increasing $\alpha$ much as in the case of "quenched" (RS) boundary condition discussed in sec. 3.3. We limit ourselves to $\alpha$ such that $\xi_{\text{glass}}(\alpha) < L/2$ so that we have a liquid phase left at the center of the system. In this circumstance the boundary condition on the other side $q_{ab}(L)$ is irrelevant.

A remarkable feature of the resulting glass order parameter is that the hierarchical structure put on the input propagates into the interior of the network preserving its basic hierarchical structure. The numerical solution suggests that the $q(x, l)$ function at a given layer $l$ consists of three parts: 0) $q(x, l) = 0$ for some interval $0 < x < x_l$ 1) 'continuous part' $q(x, l) = a(x - x_l)$ in the interval $x_l < x < 1/a$ with the same slope $a$ as in the input 2) 'plateau' $q(x, l) = q_{\text{EA}}(l) = 1 - ax_l$ in the last interval $1/a < x < 1$ as in the input. Correspondingly the overlap distribution function $P(q) = dx(q)/dq$ becomes,

$$P(q, l) = x_l \delta(q) + \frac{1}{a} + \left(1 - \frac{1}{a}\right) \delta(q - (1 - ax_l)), \tag{28}$$

which consists of three parts: 0) delta peak at $q = 0$ 1) constant part with height $1/a$ in the interval $x_l < x < 1/a$ as in the input 2) delta peak at $q = q_{\text{EA}}(l)$.

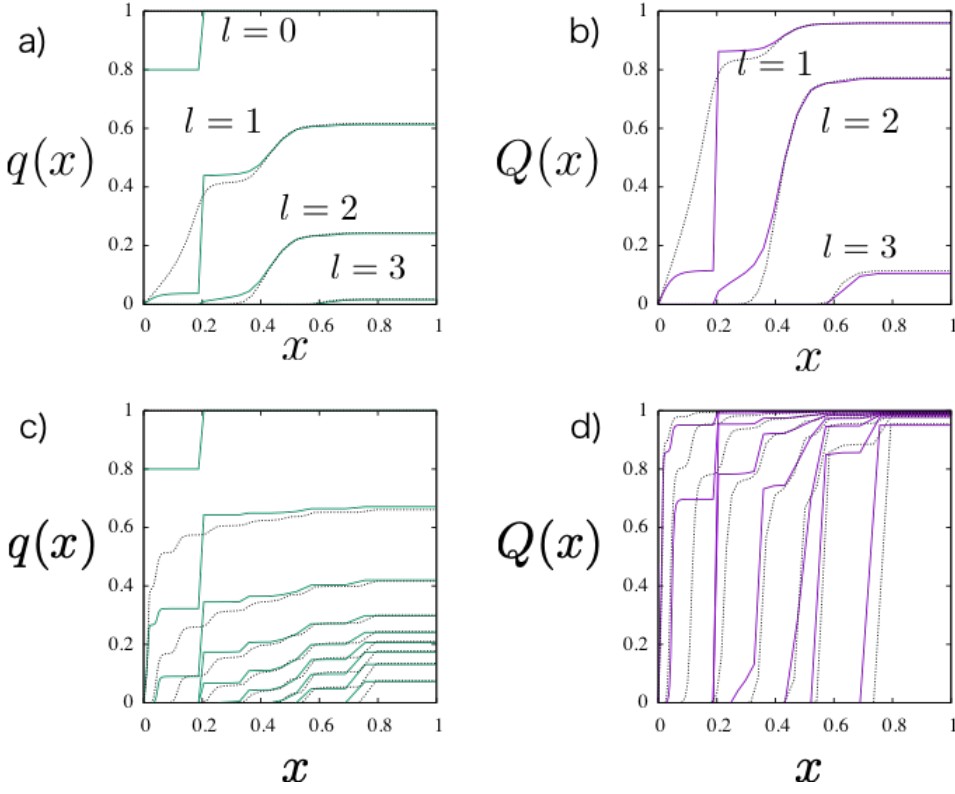

Figure 12: Glass order parameters with 1 RSB input. Here $L = 20$, $x_{\text{input}} = 0.2$ and $r = 0.8$ for the solid lines and $x_{\text{input}} = 0.2$ a),b) $\alpha = 50$ c),d) $\alpha = 4000$. Doted lines represent the glass order parameters with the frozen boundary.

Going deeper into the interior increasing $l$, we find $x_l$ grows and $q_{\text{EA}}(l) = 1 - a x_l$ decreases. We can regard this as a kind of 'renormalization' of the input data : the embedded overlap structure at low overlaps in the input data become progressively renormalized into the $q = 0$ sector in the hidden layers, keeping only the important part of the hierarchical structure at higher overlaps. It will be very interesting to study further the implication of this result in the context of data clustering where the idea of ultrametricity is very useful.

## 3.5 Teacher-student setting

Now let us turn to analyze the teacher-student setting introduced in sec. 2.2.2 by the replica theory using the ansatz explained in sec. A.5.2.

Since we are limiting ourselves to the Bayes optimal case, it is sufficient to consider the replica symmetric ($k = 0$) ansatz so that the Nishimori condition holds [8,50,51], which reads in the present system as,

$$r(l) = q_0(l), \qquad R(l) = Q_0(l). \tag{29}$$

The saddle point equations in sec. C.3 admit such solutions.

In Fig. 14 we show the profile of the solutions obtained at various $\alpha = M/N$. Remarkably the spatial profile of the order parameters are very similar to those of random inputs/outputs (See Fig. 6). This is again due to successive layer-by-layer, 2nd order 'crystalline' phase transitions which start from the boundaries. The overlap of the student machine to the teacher

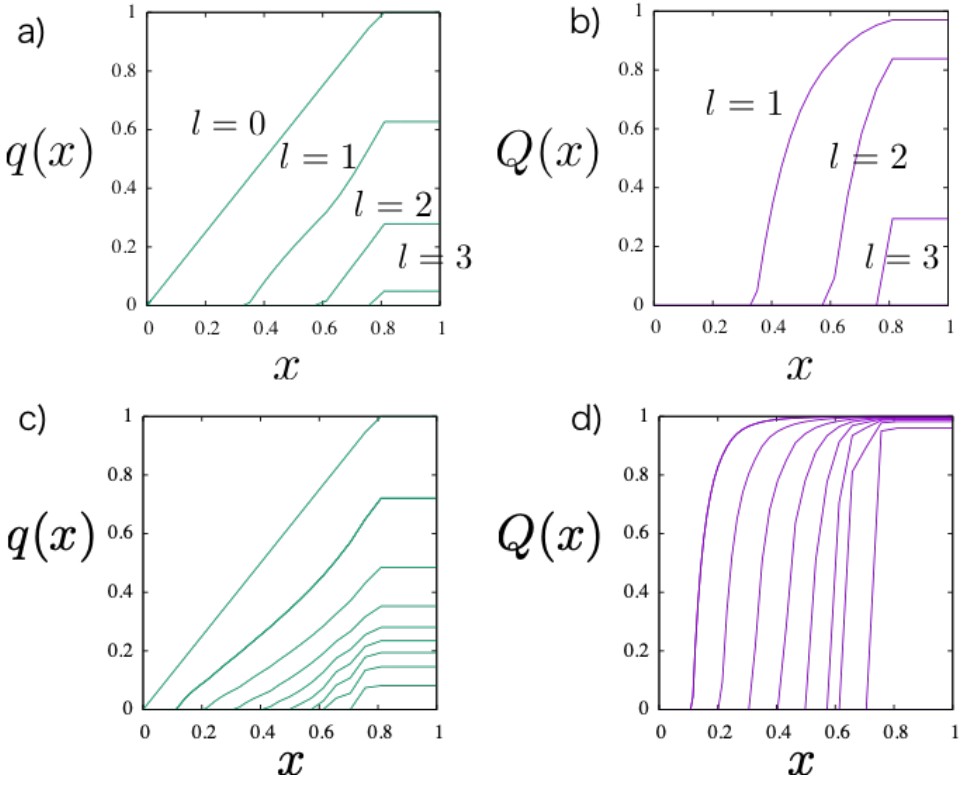

Figure 13: Glass order parameters with full RSB input. Here $L = 20$ and $1/a = 0.8$.
a),b) $\alpha = 50$ c),d) $\alpha = 4000$.

machine grows from the boundary and the penetration depth grows again as

$$\xi_{\text{teacher–student}} \propto \ln \alpha. \tag{30}$$

Remarkably the central part of the student machine remains de-correlated from the teacher machine if the system is deep enough, i. e. $L > \xi$. The solution (for the case $L > \xi$) in the crystalline region does not change even in $L \to \infty$ limit. The reason for the crystalline transition starting from the 1st layers ($l = 0, L-1$) is again the entropic effect: some set of configurations of the bonds in the 1st layers ($l = 0, L-1$) allow exceedingly larger fluctuation in the hidden layers compared with others so that they dominate the entropy of the solution space.

Now let us discuss what the above theoretical results mean for practice. The fact that the transitions are 2nd order transitions is a very good news. This is because it implies that inference will not be too difficult [8]: we do not need to worry about the possibility to be trapped in the solution of $R = 0$ (failure of inference) because it becomes unstable at the transition. However, very importantly, we have to ask what would play the role of symmetry breaking field by which the student machine can detect the teacher's configuration during learning. In our theory, we had the convenient 'fictitious' symmetry breaking field (see sec. A.1.1) but it must be realized by some 'real' field (in the computer!). Actually, if the central part remains really random, how can the student machine ever develop some finite overlap to the teacher machine at the opposite ends disconnected by the liquid phase in between? Our analysis for the case of the random inputs/outputs would suggest otherwise: the student machine should not be able to pick up the minima planted by the teacher machine correctly

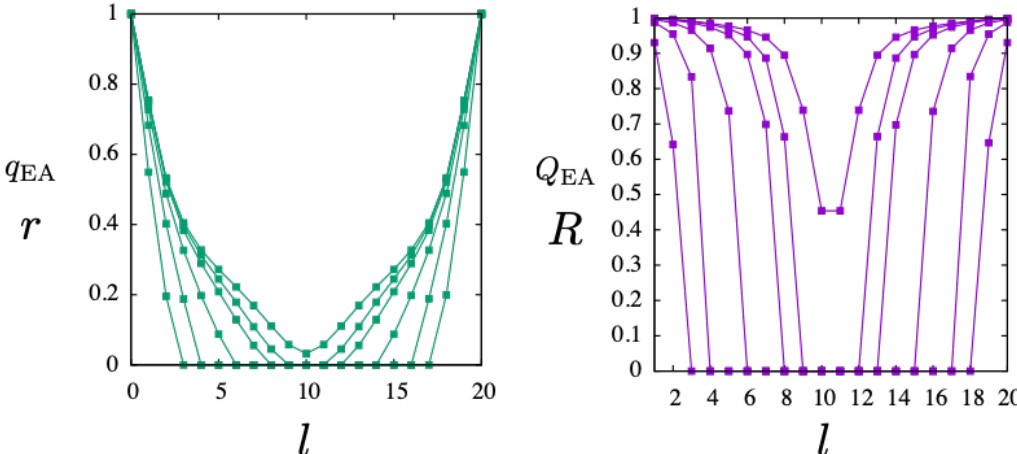

Figure 14: The spatial profile of the EA order parameters $q_{EA}(l) = q_0(l) = r(l)$, $Q_{EA}(l) = Q_0(l) = R(l)$ (RS solution $k = 0$) at $\alpha = 25, 100, 250, 500, 625, 714$. Here $L = 20$. For the largest $\alpha$, $\xi > L$ and the liquid phase disappear, which is a 'finite depth' effect .

hidden in the ocean of many (wrong) minima, all of which correctly satisfy the constraints on the input and output boundaries.

At the moment we do not have a proposal for the real symmetry breaking field which works during learning. Instead, we can think of the following *unlearning*. Suppose that we give the student machine the complete configuration of the teacher machine at the beginning then let the student machine relax under the constraint by the training data of size $M = \alpha N$. Our theory implies that the student machine will keep the configuration of the teacher machine close the boundaries over the region of size $\xi \propto \ln \alpha$ but forget the teacher machine in the center.

However, during unlearning, some weak correlation between the teacher and student machines of order, say $O(\log(N)/N)$ which does not contribute the order parameter Eq. (13) in the limits $N \to \infty$ (with fixed $\alpha = M/N$), can remain in the central part of the system. Once established this would play the role of the symmetry-breaking field: the free-energy of the selected state (teacher's configuration) will be lowered by an amount of order $O(\log(N))$ to other low lying (wrong) states. In this way the teacher's configuration may survive close to the boundaries. Such logarithmic correction naturally arises by integrating out the fluctuation of the order parameters around the saddle point. We leave the detailed analysis of the correction for future studies.

The next question is how the performance of the student-machine compares with the output of the teacher machine with respect to unseen test data. Increasing $\alpha$, i. .e. the size of training data, not only the thickness of the crystal phase $\xi$ grows but also the the amplitude of the bias field, that is the polarization of the student machine with respect to the teacher machine in the liquid-like region will become larger. Because of these two aspects, we expect the output of the student machine against the unseen test data is not totally decorrelated from that of the teacher machine even in the over-parametrized situation but the similarity of their outputs (generalization ability of the student machine) increases with the size of the training data $\alpha$.

The above scenario based on unlearning is obviously artificial (we are not interested in unlearning but learning!) but may help us to understand better generalization.

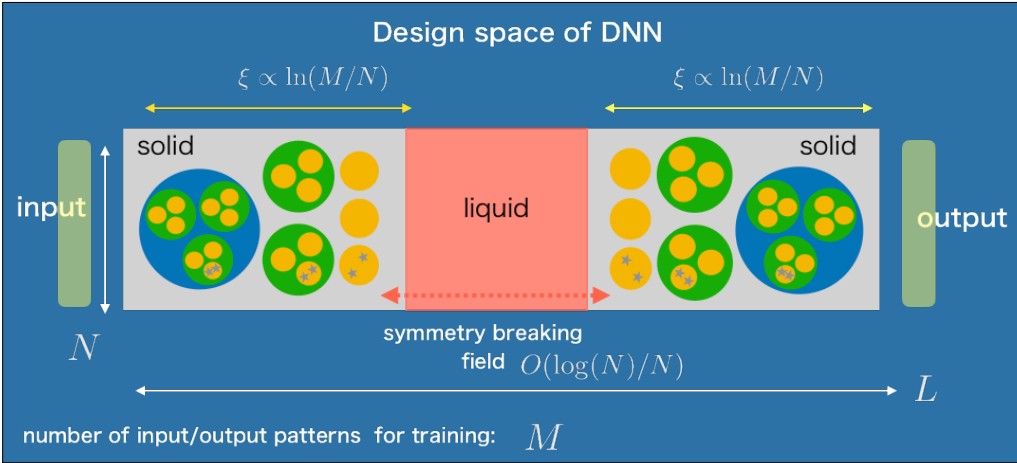

Figure 15: Schematic picture of the design space of deep neural network based on the present theory

### 3.6 Summary

The non-linear dynamics Eq. (1) in random perceptron networks is known to be highly chaotic [31, 32]. Among all such random perceptrons, which typically give chaotic dynamics, we considered statistical mechanics on the ensemble of extremely *rare* samples which happen to meet a large number of externally imposed inputs/outputs boundary conditions. Our theory predicts that such a *selection* (learning) on the ensemble of chaotic trajectories give rise to a hierarchical clustering of the trajectories (solutions) which evolves in space as shown in Fig. 15. The presence of the liquid phase in the center is consistent with the chaos. The spatial evolution of RSB smoothly connects the free-liquid like center and strongly constrained boundaries.

Imagine that we are monitoring the behavior of multiple machines that are subjected to the same inputs/outputs boundary condition but evolving (learning) independently from each other. Their configurations are represented by 'stars' in Fig. 15. Starting from the input layer, we notice that they progressively become more separated going deeper into the bulk but they become closer again approaching the output layer. The initial part, where different clusters of solutions (machines) merge into bigger cluster look like forgetting the detailed differences (renormalization) and the latter is the reverse: it amplifies mutual differences to produce the desired output (label). This picture appears to be consistent with some intuitions gained in some studies of machine learning in DNNs [52].

Usually (by definition) chaotic systems are extremely weak against perturbations. It is very interesting to ask what happens if selections (learning) come into play. Our theory implies that, during learning, a machine can diffuse chaotically within the huge continent of solutions (liquid) at the center without violating the imposed boundary conditions. Larger fluctuation means entropic stability. Thus it is not inconceivable that the combination of the strong internal chaos and the selection made at the boundaries can create a machine whose output is strong against perturbations. Our theoretical results suggest this is the case.

One can view Fig. 15 as a picture of the phase space of hard-spheres bounded by two walls made by frozen particles. The frozen boundaries act like pinning field for the particles and successive layer-by-layer glass transitions start from the boundaries as the pressure is increased. This is similar to the layer-by-layer growth by physical adsorption on substrates [53]. As the glass region grows in space, the interior of the glass region experiences further glass transitions (like the Gardner transition [48]) by which their phase space become split further. In the teacher-student setting, one of the glass corresponding to that of the teacher

and the student tries to find it.

## 4 Simulations of learning

Now we turn to discuss some numerical simulations to examine our theoretical predictions regarding the setting of a random constraint satisfaction problem with random inputs/outputs at boundaries.

In sec. 3.3 we found theoretically that the free-energy landscape of the perceptron network subjected to random constraints on the boundaries exhibit spatially heterogeneous structure: it is very complicated close to the boundaries but very simple in the central part. This naturally implies that learning dynamics is also heterogeneous in space.

To examine the nature of the learning dynamics we perform Monte Carlo simulations of the multi-layer neural network with depth $L$, width $N$ and randomly quenched inputs/outputs spins. The effective Hamiltonian of the system Eq. (11) reads as,

$$H = \sum_{\mu} \sum_{\blacksquare} V(r_{\blacksquare}^{\mu}), \qquad r_{\blacksquare}^{\mu} \equiv \sum_{i=1}^{N} \frac{J_{\blacksquare}^{i}}{\sqrt{N}} S_{\blacksquare(i)}^{\mu} S_{\blacksquare}^{\mu}. \tag{31}$$

For convenience for the simulation, we replace the hard-core potential Eq. (7) by a soft-core potential,

$$V(h) = \epsilon h^2 \theta(-h), \tag{32}$$

where $\epsilon$ is the unit of energy. Note that the statistical mechanics of a system with the soft-core potential becomes the same as the hardcore potential in the zero-temperature limit $k_B T/\epsilon \to 0$, where $k_B$ is the Boltzmann's constant and $T$ is the temperature, in the region where all the constraints are satisfied (SAT).

The dynamical variables are the $M$-component vector spins and bonds,

$$S_{\blacksquare}^{\mu} \qquad (\mu = 1, 2, \dots, M) \qquad (\blacksquare = 1, 2, \dots, N(L-1)), \tag{33}$$

$$J_{\blacksquare}^{i} \qquad (i = 1, 2, \dots, N) \qquad (\blacksquare = 1, 2, \dots, NL). \tag{34}$$

(Here we excluded the spins on the boundaries which are fixed.) Each component of the spins only takes Ising values $\pm 1$ while each of the bonds takes continuous values. In order to satisfy the normalization condition Eq. (2) $\sum_{i=1}^{N}(J_{\blacksquare}^{i})^2 = N$, we assume that $J_{\blacksquare}$ follows a Gaussian distribution with 0 mean and variance 1. We performed simple Metropolis updates of the dynamical variables at very low temperatures $T$ to simulate the relaxational dynamics. In a sense the finite temperature Monte Carlo dynamics mimic the 'stochastic' nature of the standard Stochastic Gradient Descent (SGD) algorithms used for training of DNNs [1]. To propose a new spin configuration for the Metropolis algorithm, first we select a spin component $S_{\blacksquare}^{\mu}$ randomly out of the $N \times L \times M$ possibilities and then flip it as $S_{\blacksquare}^{\mu} \to -S_{\blacksquare}^{\mu}$. To update the bond configuration, first we select a bond $J_{\blacksquare}^{i}$ randomly out of the $N \times L \times N$ possibilities and then shift its value as,

$$J_{\blacksquare}^{i} \to \frac{J_{\blacksquare}^{i} + rx}{\sqrt{1+r^2}}, \tag{35}$$

where $x$ is a random number following the Gaussian distribution with zero mean and variance 1. We set $r = 0.1$ in our simulations. Within 1 MCS (Monte Carlo Step), the unit step of the Monte Carlo simulation, we try updates of the spins $N \times L \times M$ times and updates of the bonds $N \times L \times N$ times.

At first the configurations of the frozen spins on the input $l = 0$ and the output $l = L$ layers are generated randomly. The initial configurations of the mobile spins are bonds are also

generated randomly. Then spins and bonds are updated using the Metropolis algorithm at a low temperature $T$. In our simulations we set $k_B T/\epsilon = 0.015$. Here we prepare two machines $a$ and $b$ which evolves from the same initial configurations, common boundary configuratinos for the spins on the boundaries. The two machines are updated independently by the Monte Calro method using independent series of random numbers.

We measure the following overlaps between the two machines (replicas),

$$Q(t,l) = \frac{1}{N^2} \sum_{\blacksquare \in l} \sum_{i=1}^{N} \overline{(J_\blacksquare^i)^a(t)(J_\blacksquare^i)^b(t)}, \qquad q(t,l) = \frac{1}{MN} \sum_{\blacksquare \in l} \sum_{\mu=1}^{M} \overline{(S_\blacksquare^\mu)^a(t)(S_\blacksquare^\mu)^b(t)}, \qquad (36)$$

where $\blacksquare \in l$ stands for summation over the perceptrons within the $l$-th layer. Since the two replicas start from the same initial conditions $q(0,1) = Q(0,l) = 1$ and the overlaps decay with time $t$. As we noted in sec. 3.1 the system is symmetric under permutations of the perceptrons $\blacksquare$ within the same layer. This does not matter here as long as we limit ourselves on the time scales where the correlation functions defined above remain positive. The overline $\overline{\cdots}$ represents the average over different samples: the realization of the inputs/outputs spins are chosen randomly for each sample. In the following, we used $60 - 240$ samples.

In Fig. 16 we show data of replica overlap of the spins and bonds plotted against linear time $t$. Here the width is $N = 20$. From the panels in the row a), it can be seen that the dynamics is actually faster in the center and slower close to the boundaries as we expected. From the panels in row b), it can be seen that the dynamics become slower as the strength of the constraints $\alpha$ increases. This is consistent with the theoretical expectation that the system becomes more glassy with larger $\alpha$. From the panels in row c), it can be seen that relaxation is apparently faster in the deeper system. Interestingly this happens even in the layer just next to the boundaries. Presumably, this implies that the deeper system is more flexible in the center and the fast dynamics at the center assists the relaxation of the whole system.

In the data shown above, the overlap of the spins and both tend to decay down to 0 at the long times. This implies the finite $N$ system is in the liquid phase everywhere at long enough time scales, as it should be. In Fig. 17 we show data of the overlaps of various width $N$ against *logarithmic* time so that we can also see dynamics at shorter time scales. Apparently system with smaller width $N$ decay faster suggesting that there is a finite relaxation time $\tau(N,\alpha,l)$ which increases with the width $N$ (increases also with $\alpha$ and becomes smallest at the center $l \sim L/2$). It can bee seen that at short enough time scales $t \ll \tau(N,\alpha,l)$, the data do not depend on width $N$ suggesting there is a limiting curve $N \to \infty$ with fixed $\alpha$. This is consistent with our theory in which the parameter $\alpha$ is the essential control parameter. Some of such limitting curves suggest complex dynamics with plateaus which are signatures of glassy dynamics [9]. Overlap of the bonds appear to be larger than those of spins. These features are consistent with our theory. We consider that the slow dynamics at the shorter time scales $t \ll \tau(N,\alpha,L)$ reflect the complex free-landscape and the truncation of the slow dynamics at longer time scales is a finite width $N$ effect. It is interesting to note that somewhat similar truncation of the slow dynamics has been observed in a study of SGD dynamics in DNNs [21].

Finally, in Fig. 18, we display the relaxation of energy $e(t,l) = E(t,l)/N$ at each layer $l$. Here $E(t,l)$ is the contribution of $l$ th layer to the energy (see Eq. (31)-Eq. (32)) at time $t$ (MCS). It can be seen again that relaxation is slower closer to the boundaries. The data also suggest that there is a $N \to \infty$ limit curve with fixed $\alpha$. Note also that the behavior of the system is not symmetric to the exchange of input and output sides. The asymmetry becomes stronger closer to the boundaries as one naturally anticipates.

To summarize the numerical observation of the relaxations is qualitatively consistent with the theoretical prediction which implies spatially heterogeneous dynamics.

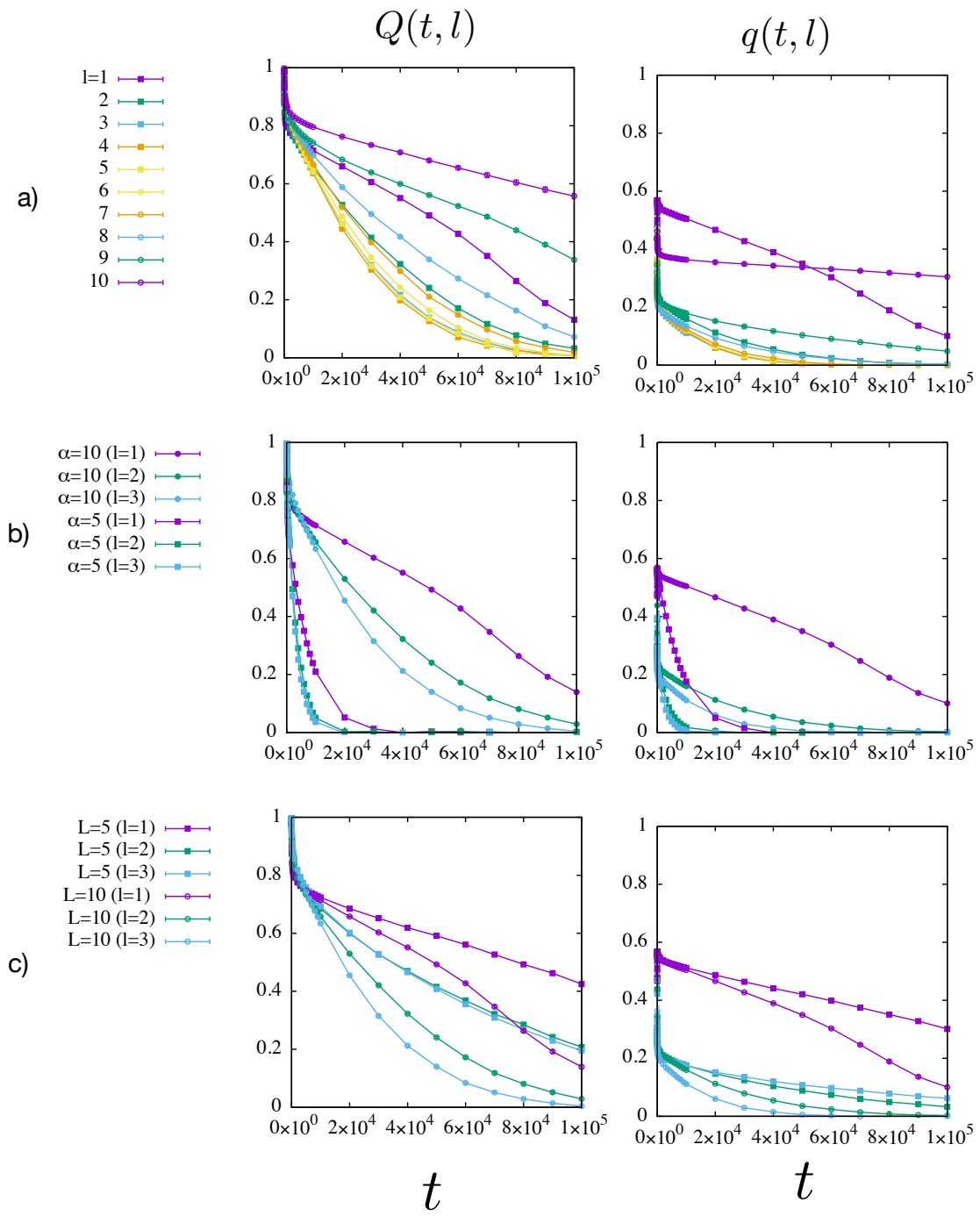

Figure 16: Relaxation of the replica overlaps of the bonds $Q(t,l)$ and spins $q(t,l)$. The unit of time is 1 MCS. In the 1st row a) data at different layers $l = 1, 2, \ldots, 10$ ($L = 10$, $N = 20$, $\alpha = 10$) are shown. In the 2nd row b) data for $\alpha = 5$ and $10$ ($L = 10$, $N = 20$) are shown. In the 3rd row c) data obtained with different depth $L = 5$ and $10$ ($N = 20$, $\alpha = 10$) are shown. Error bars are smaller than the size of the symbols.

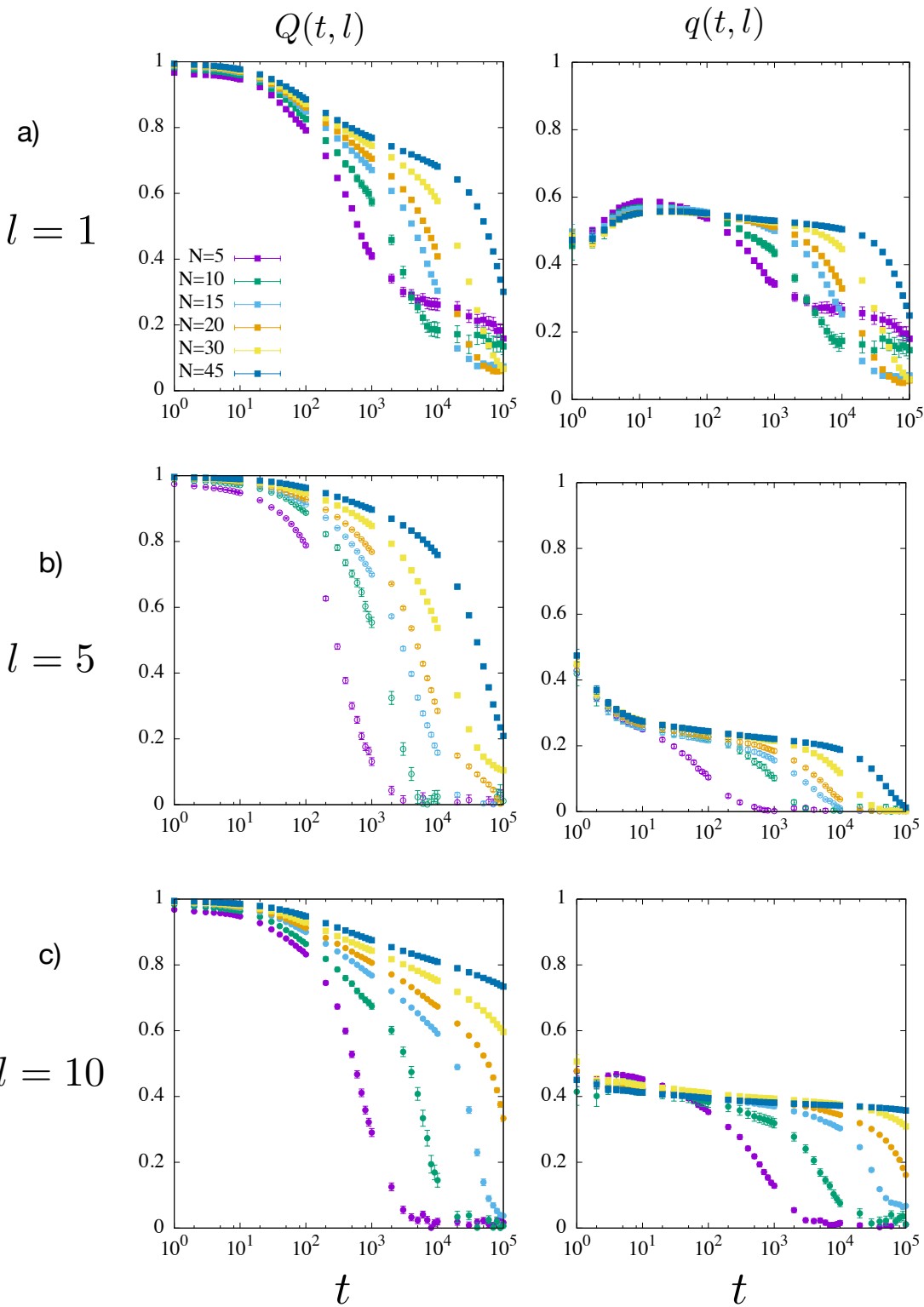

**Figure 17:** Relaxation of the replica overlaps of the bonds $Q(t,l)$ and spins $q(t,l)$ at various width $N$ plotted against logarithmic time. The unit of time is 1 MCS. Here $L = 10$, $T = 0.015$ and $\alpha = 5$. Data at different layers (a) close to the input $l = 1$, (b) at the center $l = 5$ and (c) close to the output $l = 10$ are shown.

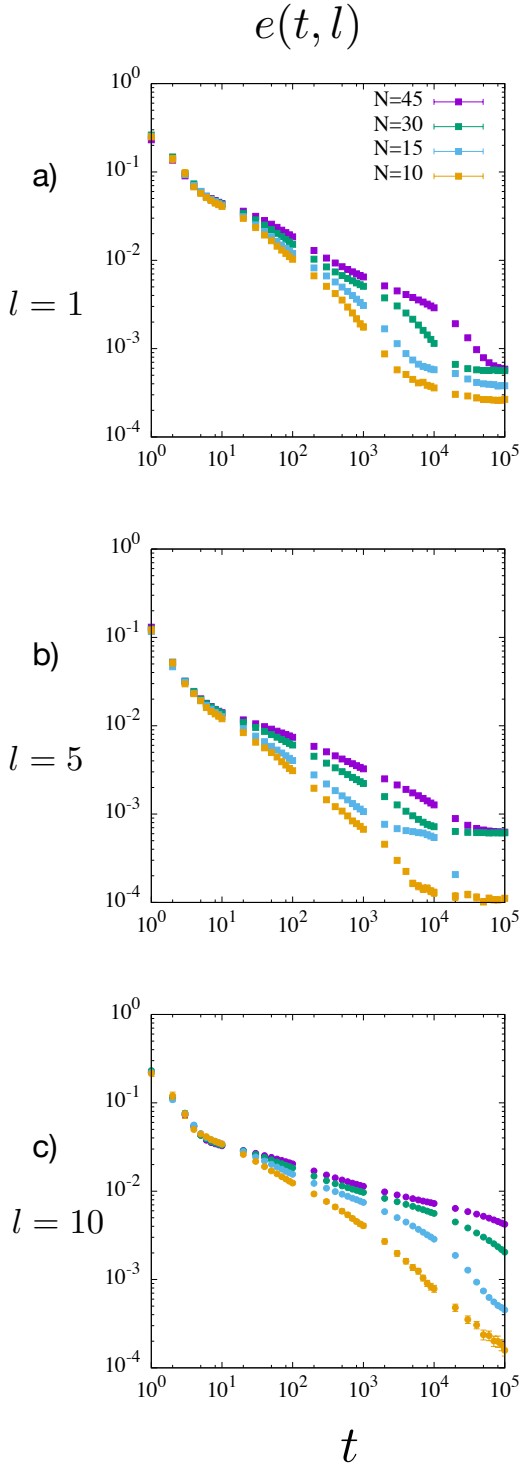

Figure 18: Relaxation of energy $e(t,l)$ of various width $N$. Here $L = 10$, $T = 0.015$ and $\alpha = 10$. The unit of time is 1 MCS. Data at different layers (a) close to the input $l = 1$, (b) at the center $l = 5$ and (c) close to the output $l = 10$ are shown. Error bars are smaller than the size of the symbols.

# 5 Conclusion and outlook

## 5.1 Conclusion

In the present paper, we constructed a statistical mechanical theory for the solution space of a deep perceptron network of depth $L$ and width $N$ subjected to $M = \alpha N$ patterns of data using the internal representation, based on the replica method in the limit $N, M \to \infty$ with fixed $\alpha$. We studied two scenarios :

1) Random inputs/outputs, which is a random constraint satisfaction problem

2) Teacher-student setting, which is a statistical inference problem.

In addition, we performed simulations to examine the theoretical predictions on 1).

The main outcome of the theory is the prediction of the strongly heterogeneous spatial profile of states inside the layered network : 1) 'glass-liquid-glass' in the case of the random inputs/outputs 2) 'crystal-liquid-crystal' in the case of the teacher-student setting. We find $\alpha = M/N$ is the key parameter which plays the role similar to the the inverse temperature in condensed matters. The thickness of the glass/crystal phase grows as $\xi \propto \ln \alpha$. This implies exponential growth of the storage capacity of DNN with the depth $\alpha_j(L) \propto e^{\text{const} L}$ for typical instances. Moreover, in the case of setting 1) random inputs/outputs, even the pattern of the replica symmetry breaking (RSB) varies in space: it is most complex close to the boundaries with $k$(+continuous)-RSB, which becomes $k-1$(+continuous)-RSB in the next layer, ... down to a replica symmetric (0 RSB) state in the central part. The hierarchical structures in different layers are synchronized. We argued that in the 2) teacher-student setting, the small positive overlap can remain in the liquid phase as a finite $N$ correction and plays the role of symmetry-breaking field.

There are some weak points in our theory which must be clarified by further works,

- Our theory assumes the wide limit $N \to \infty$ (with $\alpha = M/N$ fixed) while real networks have *finite* width $N$ so that 'phase transitions' we found here become at most just crossovers. Nonetheless, we believe our results still provide useful guidelines to understand real DNNs.

- Technically our theory relies on the tree-approximation which does not take into account the 1-dimensional inter-layer fluctuations along the $z$-axis (See Fig. 1) faithfully. We believe that the qualitative nature of the system in the limit $N \to \infty$ do not change by loop-corrections much as the Ginzburg-Landau theory in $1 + \infty$ dimension (here 1 is for the $z$-axis and $\infty$ dimension for the transverse directions) do not change the mean-field nature. However consideration of loop-corrections would improve the theory at small scales especially close to the boundaries where spatial heterogeneity is strongest. It is also important in systems with smaller width $N$ where we expects the effects of inter-layer fluctuations become larger. Indeed if we take $L \to \infty$ limit with finite width $N$, the system becomes truly an one dimensional system.

- Our theory is based on order parameters. This is a strong point but the theory itself does not answer what plays the role of symmetry breaking field conjugated to the order parameters in practice. This is a particularly important open question in the inference problem (teacher-student setting).

For 1), we also performed some simulations of the relaxational dynamics. We found crossover from the complex glassy slow dynamics to rapid decay at longer time scales. For fixed $\alpha = N/M$, the crossover time increases with the width $N$. As the theory suggests, the

glassy dynamics is controlled by the strength of $\alpha$ and spatially heterogeneous. It is faster and simpler in the center and slower and more complex closer to the boundaries in agreement with the theory. We leave numerical investigation for the teacher-student setting 2) for future works.

To summarize, both the theory and the simulation suggest spatially heterogeneous free-energy landscape in DNNs which is controlled by the parameter $\alpha = M/N$ (See Fig. 15). We speculate that this is responsible for the efficiency of DNNs in three respects:

a) The presence of the liquid phase in the center may facilitate the equilibration (learning) of the whole system. This is an interesting point which deserves further studies. In this respect, it is interesting to note that the so-called ultra-stable glasses [54–56] are created by vapor deposition which allows rapid equilibration at layers close to the surface. As we noted in sec. 3.6, the analogy to the hard-spheres is suggestive. One could also think about the analogy with the replica-exchange Monte Carlo method [57] which dramatically accelerates the equilibration of complex systems.

b) Despite the over-parametrization the system may still generalize because of the crystalline phase close to the boundaries and weak bias field in the liquid phase. However, how the bias field can be prepared in practice is an important open question.

c) Hierarchical free-energy landscapes with ultra-metric structure has been known in glass physics since the discovery of replica symmetry breaking [14, 15, 44]. Our theory suggests that it evolves in space in the DNN, during learning, progressively from the complex to simple ones going deeper into the interior from both the input and output boundaries. Probably the spatial 'renormalization' of the hierarchical clustering and the presence of the liquid phase at the center stabilizes the system against external perturbations or incompleteness of equilibration and contributes positively to the generalization ability of DNNs. As discussed in sec. 3.4.2, it would also be very interesting to study implications for unsupervised learning including in particular hierarchical data clustering [58].

## 5.2 Outlook

There are many possibilities for further investigations including the following. First of all, it will be very interesting to perform extensive numerical simulations with realistic algorithms and realistic large-scale data to examine our predictions. Second, a more detailed theoretical/numerical analysis of the remnant bias field in the liquid phase is needed. Third, the storage capacity and critical properties at jamming (SAT/UNSAT transition) [18–23] can be studied in detail by analyzing the regime $\xi \gg L$. Lastly, it will be interesting to extend the present work to other cases regarding the activation function, architecture, and learning methods.

Our system may be view as a $1+\infty$ dimensional glass which is an interesting playground to analyze spatial heterogeneity with mean-field theoretical approaches [53, 59–61]. The present work may also have implications on various complex systems with spatial heterogeneity, including ultra-stable glass [54–56] mentioned above, gene regulatory networks [62–64] which are often viewed like perceptrons and allosteric systems [65–67]. The central liquid region may be related to the idea of *neutral space* [62, 63] which is considered as responsible for robustness of biological systems.

# Acknowledgments

We thank Giulio Biroli, Silvio Franz, Koji Hashimoto, Sungmin Hwang, Kyogo Kawaguchi, Macoto Kikuchi, Kota Mitsumoto, Tomoyuki Obuchi, Haruki Okazaki, Akinori Tanaka, Pierfrancesco Urbani, Takaki Yamamoto, Lenka Zdeborová and Francesco Zamponi for useful discussions.

Numerical analysis in this work has been done using the supercomputer systems OCTOPUS and SX-ACE at the Cybermedia Center, Osaka University. The author thanks the Simons collobration on "cracking the glass problem" for opportunities of stimulating discussions. The author thanks the Yukawa Institute for Theoretical Physics at Nyoto University for discussions during the YITP workshop YITP-W-19-18 "Deep Learning and Physics 2019".

**Funding information** This work was supported by KANENHI (No. 19H01812) from MEXT, Japan.

# A   Replicated free-energy

The replicated phase space volume (the Gardner volume) can be written as,

$$
\begin{aligned}
V^n\left(\mathbf{S}_0, \mathbf{S}_L\right) &= e^{NM\mathcal{S}_n(\mathbf{s}_0, \mathbf{s}_l)} \\
&= \prod_{a=1}^{n}\left(\prod_{\blacksquare}\mathrm{Tr}_{\mathbf{J}_{\blacksquare}^a}\right)\left(\prod_{\blacksquare\backslash\text{output}}\mathrm{Tr}_{\mathbf{S}_{\blacksquare}^a}\right)\left\{\prod_{\mu,\blacksquare,a}\int\frac{d\eta_{\mu,\blacksquare,a}}{\sqrt{2\pi}}W_{\eta_{\mu,\blacksquare,a}}e^{i\eta_{\mu,\blacksquare,a}(r_{\blacksquare}^{\mu})^a}\right\} \\
&= \left(\prod_{\mu,\blacksquare,a}\int\frac{d\eta_{\mu,\blacksquare,a}}{\sqrt{2\pi}}W_{\eta_{\mu,\blacksquare,a}}\right)\left(\prod_{\blacksquare,a}\mathrm{Tr}_{\mathbf{J}_{\blacksquare}^a}\right)\left(\prod_{\blacksquare\backslash\text{output},a}\mathrm{Tr}_{\mathbf{S}_{\blacksquare}^a}\right) \\
&\qquad\prod_{\mu,\blacksquare,a}e^{i\eta_{\mu,\blacksquare,a}(S_{\blacksquare}^{\nu})^a\sum_{i=1}^{N}\frac{(J_{\blacksquare}^i)^a}{\sqrt{N}}(S_{\blacksquare(i)}^{\nu})^a},
\end{aligned}
\tag{37}
$$

where we introduced a Fourier representation,

$$
e^{-\beta V(r)} = \int\frac{d\eta}{\sqrt{2\pi}}W_{\eta}e^{-i\eta r}.
\tag{38}
$$

In the following we derive the free-energy functional of the replicated system starting from Eq. (37), following similar steps as in [20].

## A.1   Basic strategy

Before going to the details of the computations, let us sketch the basic strategy to extract properties of glassy phases using the replica approach in the present work as well as [20]. Very importantly, this applies to systems without the quenched disorder. Actually, this strategy lies behind the replica approach to structural glasses [11, 29, 37, 68, 69]. This is an important point for our present problem which is essentially disorder-free except for the boundaries.

### A.1.1   Explicit replica symmetry breaking

For simplicity, suppose that we have a generic system which consists of $N$ degrees of freedom $\{x\} = (x_1, x_2, \dots, x_N)$ whose Hamiltonian is $H[\{x\}]$, which can be with/without the quenched

disorder. Let us introduce $n$ replicas $a = 1, 2, \ldots, n$ and a replicated Hamiltonian,

$$H_n[\hat{\epsilon}] = \sum_{a=1}^{n} H[\{x_i^a\}] - \sum_{a<b} \epsilon_{ab} \sum_{i=1}^{N} x_i^a x_i^b. \tag{39}$$

Here we introduced the 2nd term on the r.h.s. which represts an artificial attractive coupling $\epsilon_{ab} > 0$ between replicas. The field $\epsilon_{ab}$ explicitly breaks the replica symmetry, i.e. the permutation symmetry of replica index. The free-energy of the replicated system can be defined as,

$$-\beta G[\hat{\epsilon}] = \ln \prod_{a=1}^{n} \prod_{i=1}^{N} \text{Tr}_{x_i^a} e^{-\beta H_n[\hat{\epsilon}]}. \tag{40}$$

This allows us to evaluate the overlap between the replicas,

$$q_{ab} = \frac{1}{N} \sum_{i=1}^{N} \langle x_i^a x_i^b \rangle_\epsilon = -\frac{1}{N} \frac{\partial G_\epsilon}{\epsilon_{ab}}. \tag{41}$$

We are interested with,

$$\lim_{\hat{\epsilon} \to 0} \lim_{N \to \infty} q_{ab} \tag{42}$$

and consider that this is the glass order parameter of the system. This is the idea of explicit replica symmetry breaking (RSB) by Parisi and Virasoro [28]. Similarly to the magnetic field $h$ for magnetization in ferromagnetic systems, the field $\epsilon_{ab}$ is conjugated to the glass order parameter $q_{ab}$ and plays the role of symmetry breaking field. One can define a 'glass' susceptibility,

$$\chi_{ab,cd} \equiv -\frac{1}{N} \frac{\partial}{\partial \epsilon_{ab} \partial \epsilon_{cd}} \beta G[\hat{\epsilon}] = \frac{\partial q_{ab}}{\partial \epsilon_{cd}} = \frac{1}{N} \sum_{i,j=1}^{N} \left[ \langle x_i^a x_i^b x_j^c x_j^d \rangle_\epsilon - \langle x_i^a x_i^b \rangle_\epsilon \langle x_j^c x_j^d \rangle_\epsilon \right]. \tag{43}$$

Instability toward spontaneous replica symmetry breaking may accompany divergence of the glass susceptibility at $\epsilon = 0$.

Let us then consider the Legendre transform of the free-energy,

$$-\beta F[\hat{q}] = -\beta G[\hat{\epsilon}^*] - N \sum_{a<b} \epsilon_{ab}^* q_{ab}, \tag{44}$$

where $\hat{\epsilon}^* = \hat{\epsilon}^*[\hat{q}]$ is defined such that,

$$\frac{1}{N} \frac{\partial}{\partial \epsilon_{ab}} (-\beta G[\hat{\epsilon}]) \bigg|_{\hat{\epsilon}=\hat{\epsilon}^*[\hat{q}]} = q_{ab}. \tag{45}$$

The inverse of the Legendre transform is,

$$-\beta G[\hat{\epsilon}] = -\beta F[\hat{q}^*] + N \sum_{a<b} \epsilon_{ab} q_{ab}^*, \tag{46}$$

where $\hat{q}^* = \hat{q}^*[\hat{\epsilon}]$ is defined such that,

$$\frac{1}{N} \frac{\partial}{\partial q_{ab}} (-\beta F[\hat{q}]) \bigg|_{\hat{q}=\hat{q}^*[\hat{\epsilon}]} = -\epsilon_{ab}. \tag{47}$$

The last expression tells us that the order parameter of our intest, which detects the spontaneous RSB Eq. (42), can be obtained by minimizing the free-energy $F[\hat{q}]$ which yields $\epsilon_{ab} = 0$. Related to Eq. (43) is the Hessian matrix,

$$H_{ab,cd} \equiv \frac{1}{N} \frac{\partial^2}{\partial q_{ab} \partial q_{cd}} (\beta F[\hat{q}]) \bigg|_{\hat{q}} = (\chi^{-1})_{ab,cd}. \tag{48}$$

The divergence of the glass susceptibility Eq. (43) imply vanishig of the eigen value(s) of the Hessian matrix [70]. Thermodynamic stability implies $\chi_{ab,cd} < \infty$ or positive (semi-) definiteness of the eigenvalues of the Hessian matrix.

### A.1.2  Ergodicity breaking

If the field does not depend on the replica indexes $\epsilon_{ab} = \epsilon$, we are not breaking the replica symmetry. But the replica symmetric (RS) field can be used at least to detect the *ergodicity breaking* where the Edwards-Anderson (EA) order parameter [71],

$$q_{\text{EA}} = \frac{1}{N} \sum_{i=1}^{N} \langle x_i \rangle^2 \tag{49}$$

becomes non-zero. Here $\langle \ldots \rangle$ represents an appropriate thermal average. For example, in a model with 3-body interaction $H = -J/N^2 \sum_{i,j,k=1}^{N} x_i x_j x_k$, the liquid phase where $q_{\text{EA}} = 0$ is realized at high enough temperatures while crystalline or glassy phases where $q_{\text{EA}} > 0$ emerge at lower temperatures [20,72].

In the context of relaxational dynamics, the EA order parameter can be considered as the long-time limit of the time autocorrelation function [71],

$$q_{\text{EA}} = \lim_{t \to \infty} C(t), \qquad C(t) \equiv \frac{1}{N} \sum_{i=1}^{N} \langle x_i(0) x_i(t) \rangle. \tag{50}$$

In the liquid phase the auto-correlation function decays down to 0 after finite relaxation time. The latter diverges at the transition leading to $q_{\text{EA}} > 0$. Thus the EA order parameter detects the Ergodicity breaking (either due to crystalline or glass transitions). Note that the RSB discussed previously in sec. A.1.1 automatically also means an ergodicity breaking, but of a more complicated version involving the hierarchical organization of relaxations [73–77].

In some cases, like a model with 2-body interaction $H = -J/N \sum_{i,j,k=1}^{N} x_i x_j$, the system is symmetric under global 'spin flip' $x_i \to -x_i$ for $\forall i$, the perturbation Eq. (39) with the RS field $\epsilon_{ab} = \epsilon$ then breaks this symmetry. In our present problem Eq. (11), the 'spins' have this symmetry. One observes that the system is no longer invariant under global flip in one replica, say $a$, $x_i^a \to -x_i^a$ for $\forall i$. Thus the role played by the RS field $\epsilon_{ab} = \epsilon$, in this case, is like the magnetic field conjugated to the magnetization which is the order parameter for usual ferromagnets.

### A.1.3  What is the symmetry breaking field?

In the theoretical formulation, we introduced conveniently the symmetry-breaking fields $\epsilon_{ab}$ as Eq. (39). But we have to ask ourselves what plays the role of the somewhat fictitious field in reality. The perturbations like Eq. (39) can be introduced by considering 'random pinning fields' [28,78]. Some sorts of weak random pinning fields may exist in nature. But what about computers? In the context of machine learning, the role of symmetry breaking field may be played by 1) choices of boundary conditions (inputs/outputs data) 2) choices of initial condition for learning.

### A.1.4  Plefka expansion

Now our task is to compute the free-energy $F[\hat{q}]$ defined in Eq. (44). To this end, we will follow the idea of Plefka expansion [79]. The computations presented in the following sections follow this strategy.

Suppose that the effect of the interactions between the dynamical variables $x_i$ ($i = 1, 2, \ldots, N$) can be treated perturbatively which enable the following decompositions,

$$F = F_0 + \lambda F_1 + \ldots \qquad G = G_0 + \lambda G_1 + \ldots \qquad \epsilon_{ab} = (\epsilon_0)_{ab} + \lambda (\epsilon_1)_{ab} + \ldots . \tag{51}$$

Here the quantities with suffix 0 repsent those which are present in the absence of interactions (like the ideal gass free-energy) and those with suffix 1 repsent those due to interactions. Here we omitted the higher-order terms. The parameter $\lambda$, which is introduced to organize a perturbation theory, is put back to $\lambda = 1$ in the end.

The Legendre transform Eq. (44) becomes, at $O(\lambda^0)$,

$$-\beta F_0[\hat{q}] = -\beta G_0[\hat{\epsilon}_0^*] - N \sum_{a<b} (\epsilon_0^*)_{ab} q_{ab}, \tag{52}$$

where $(\epsilon_0^*)_{ab}$ is defined such that,

$$\frac{1}{N} \frac{\partial}{\partial \epsilon_{ab}} (-\beta G_0[\hat{\epsilon}]) \bigg|_{\hat{\epsilon} = \hat{\epsilon}_0^*[\hat{q}]} = q_{ab}. \tag{53}$$

Then at $O(\lambda)$ we find,

$$
\begin{aligned}
-\beta F_1[\hat{q}] &= -\beta G_1[\hat{\epsilon}_0^*[\hat{q}]] + \sum_{a<b} \frac{\partial}{\partial \epsilon_{ab}} (-\beta G_0[\hat{\epsilon}]) \bigg|_{\hat{\epsilon} = \hat{\epsilon}_0^*[\hat{q}]} (\epsilon_1^*)_{ab} - N \sum_{a<b} (\epsilon_1^*)_{ab} q_{ab} \\
&= -\beta G_1[\hat{\epsilon}_0^*[\hat{q}]].
\end{aligned}
\tag{54}
$$

In the 2nd equation we used Eq. (53). Minimization of the free-energy $F[\hat{q}]$ (see Eq. (47)) implies $(\epsilon_0^*)_{ab} = -\lambda (\epsilon_1^*)_{ab}$ up to this order.

If higher order terms $O(\lambda^2)$ in the expansion Eq. (51) vanish in $N \to \infty$ limit, the treatment described above is sufficient. This happens in the derivation of the the exact free-energy functional of a family of glassy spin models [20] which include the family of p-spin (Ising/spherical) mean-field spin-glass models (with or without the quenched disorder) (see section 6 of [20]), glassy hard-spheres [11] in large-dimensional limit and aspherical particles [29] in large-dimensional limit. Unfortunately, in the present system with the layered geometry, we will find that $O(\lambda^3)$ do not vanish because of the loop effects across different layers (see Fig. 2). To make an analytical progress we invoke a tree-approximation which neglects contributions of such loops.

## A.2 Evaluation of the entropic part of the free-energy

We introduce 'local' order parameters [20], for each perceptron ■,

$$Q_{ab,\blacksquare} = \frac{1}{N} \sum_{i=1}^{N} (J_{\blacksquare}^i)^a (J_{\blacksquare}^i)^b \qquad q_{ab,\blacksquare} = \frac{1}{M} \sum_{\mu=1}^{M} (S_{\blacksquare}^{\mu})^a (S_{\blacksquare}^{\mu})^b \tag{55}$$

through the identities,

$$
\begin{aligned}
1 &= \int_{-\infty}^{\infty} \int_{-i\infty}^{i\infty} \prod_{a<b} \left( \frac{N}{2\pi} \right) dQ_{ab,\blacksquare} d\epsilon_{ab,\blacksquare} e^{N \sum_{a<b} \epsilon_{ab,\blacksquare} \left( Q_{ab,\blacksquare} - N^{-1} \sum_{i=1}^{N} (J_{\blacksquare}^i)^a (J_{\blacksquare}^i)^b \right)}, \\
1 &= \int_{-\infty}^{\infty} \int_{-i\infty}^{i\infty} \prod_{a<b} \left( \frac{M}{2\pi} \right) dq_{ab,\blacksquare} d\varepsilon_{ab,\blacksquare} e^{M \sum_{a<b} \varepsilon_{ab,\blacksquare} \left( q_{ab,\blacksquare} - M^{-1} \sum_{\mu=1}^{M} (S_{\blacksquare}^{\mu})^a (S_{\blacksquare}^{\mu})^b \right)},
\end{aligned}
\tag{56}
$$

by which we can write the summation over the configurations of the bonds and spins of each replica which appear in Eq. (37) as,

$$\prod_a \mathrm{Tr}_{J_\blacksquare^a} \cdots = \left( \prod_{a<b} \int_{-\infty}^{\infty} d(Q_{ab,\blacksquare}) \right) e^{N s_{\mathrm{ent,bond}}[\hat{Q}_\blacksquare]} \prod_{i=1}^{N} \langle \cdots \rangle_{J_\blacksquare^i}, \tag{57}$$

$$\prod_a \mathrm{Tr}_{S_\blacksquare^a} \cdots = \left( \prod_{a<b} \int_{-\infty}^{\infty} d(q_{ab,\blacksquare}) \right) e^{M s_{\mathrm{ent,spin}}[\hat{q}_\blacksquare]} \prod_{\mu=1}^{M} \langle \cdots \rangle_{S_\blacksquare^\mu}, \tag{58}$$

where we have performed integrations over $\epsilon_{ab}$ and $\varepsilon_{ab}$ by the saddle point method assuming $N \gg 1$ and $M \gg 1$. We dropped irrelevant prefactors. In Eq. (57) and Eq. (58), in the products $\prod_{i=1}^{N} \langle \cdots \rangle_{J^i}$ and $\prod_{\mu=1}^{M} \langle \cdots \rangle_{S^\mu}$, the symbol $\cdots$ refere to quantities factorized in terms of $i$ and $\mu$. Note that Eq. (55)-Eq. (58) are defined for each perceptron $\blacksquare$, which are assumed to be independent from each other following the prescription formulated in sec. A.1.4. Thus in the following, we dropp the subscript $\blacksquare$ for simplicity.

For the trace over the configurations of bonds, we find using Eq. (10),

$$s_{\mathrm{ent,bond}}[\hat{Q}] = \frac{1}{2} \sum_{a,b} \epsilon_{ab}^* Q_{ab} + \ln \prod_{c=1}^{n} \int_{-\infty}^{\infty} dJ^c e^{-\frac{1}{2} \sum_{a,b} \epsilon_{ab}^* J^a J^b}, \tag{59}$$

$$\langle \cdots \rangle_{J^i} = \frac{\prod_{c=1}^{n} \int_{-\infty}^{\infty} d(J^i)^c e^{-\frac{1}{2} \sum_{a,b} \epsilon_{ab}^* (J^i)^a (J^i)^b} \cdots}{\prod_{c=1}^{n} \int_{-\infty}^{\infty} d(J^i)^c e^{-\frac{1}{2} \sum_{a,b} \epsilon_{ab}^* (J^i)^a (J^i)^b}}, \tag{60}$$

where we introduced $Q_{aa} = 1$ and $\epsilon_{aa} = \lambda_a$ to include the integral Eq. (10) which enforces the spherical constraint Eq. (2). Simiarly, for the trace over the spin configuration we find using Eq. (9),

$$s_{\mathrm{ent,spin}}[\hat{q}] = \frac{1}{2} \sum_{a,b} \varepsilon_{ab}^* q_{ab} + \ln \prod_{c=1}^{n} \sum_{S^c = \pm 1} e^{-\frac{1}{2} \sum_{a,b} \varepsilon_{ab}^* S^a S^b} \tag{61}$$

$$\langle \cdots \rangle_{S^\mu} = \frac{\prod_{c=1}^{n} \sum_{(S^\mu)^c = \pm 1} e^{-\frac{1}{2} \sum_{a,b} \varepsilon_{ab}^* (S^\mu)^a (S^\mu)^b} \cdots}{\prod_{c=1}^{n} \sum_{(S^\mu)^c = \pm 1} e^{-\frac{1}{2} \sum_{a,b} \varepsilon_{ab}^* (S^\mu)^a (S^\mu)^b}}, \tag{62}$$

where we introduced $\varepsilon_{aa} = 0$. The saddle points $\epsilon_{ab}^* = \epsilon_{ab}^*(\hat{Q})$ and $\varepsilon_{ab}^* = \varepsilon_{ab}^*(\hat{q})$ are determined by

$$Q_{ab} = \frac{\prod_c \mathrm{Tr}_{J^c} e^{-\sum_{a<b} \epsilon_{ab} J^a J^b} J^a J^b}{\prod_c \mathrm{Tr}_{J^c} e^{-\sum_{a<b} \epsilon_{ab} J^a J^b}} \Bigg|_{\epsilon_{ab} = \epsilon_{ab}^*(\hat{Q})}, \qquad q_{ab} = \frac{\prod_c \mathrm{Tr}_{S^c} e^{-\sum_{a<b} \varepsilon_{ab} S^a S^b} S^a S^b}{\prod_c \mathrm{Tr}_{S^c} e^{-\sum_{a<b} \varepsilon_{ab} S^a S^b}} \Bigg|_{\varepsilon_{ab} = \varepsilon_{ab}^*(\hat{q})}. \tag{63}$$

Let us note that Eq. (63) correspond to Eq. (53) in the program outlined in sec. A.1.4. The $\varepsilon^*$ and $\varepsilon^*$ determined here are $\varepsilon_0^*[\hat{q}]$ and $\epsilon_0^*[\hat{Q}]$ which are evaluated in the absence of the interaction term.

The above equations imply in particular,

$$\langle (J_\blacksquare^i)^a \rangle_{J^i} = 0, \qquad \langle (J_\blacksquare^i)^a (J_{\blacksquare'}^j)^b \rangle_{J^i} = Q_{ab} \delta_{ij} \delta_{\blacksquare,\blacksquare'}, \tag{64}$$

$$\langle (S_\blacksquare^\mu)^a \rangle_{S^\mu} = 0, \qquad \langle (S_\blacksquare^\mu)^a (S_{\blacksquare'}^\nu)^b \rangle_{S^\mu} = q_{ab} \delta_{\mu\nu} \delta_{\blacksquare,\blacksquare'}. \tag{65}$$

### A.2.1 Entropic part of the 'bonds'

The entropic contribution of the bonds Eq. (60) can be readily evaluated as (see (77) of [20]),

$$s_{\mathrm{ent,bond}}[\hat{Q}] = \frac{n}{2} + \frac{n}{2} \ln(2\pi) + \frac{1}{2} \ln \det \hat{Q}. \tag{66}$$

### A.2.2 Entropic part of the 'spins'

The spins are 'Ising spins'. The entropic part of the free-energy of the spins are (see (79) of [20]),

$$s_{\text{ent,spin}}[\hat{q}] \;=\; \frac{1}{2}\sum_{a,b}\varepsilon^*_{ab}q_{ab} + \ln e^{-\frac{1}{2}\sum_{a,b}\varepsilon^*_{ab}\frac{\partial^2}{\partial h_a\partial h_b}}\prod_a 2\cosh(h_a)\Bigg|_{\{h_a=0\}}, \qquad (67)$$

with $\epsilon_{aa}=0$. Here we performed the spin trace formally as

$$\mathrm{Tr}_{\mathbf{S}^c}\,e^{-\frac{1}{2}\sum_{a,b}\varepsilon_{ab}S^aS^b} = \mathrm{Tr}_{\mathbf{S}^c}\,e^{-\frac{1}{2}\sum_{a,b}\varepsilon_{ab}S^aS^b} = \mathrm{Tr}_{\mathbf{S}^c}\,e^{-\frac{1}{2}\sum_{a,b}\varepsilon_{ab}\frac{\partial^2}{\partial h_a\partial h_b}}\,e^{\sum_a h_a S^a}\Bigg|_{\{h_a=0\}}$$

$$= e^{-\frac{1}{2}\sum_{a,b}\varepsilon_{ab}\frac{\partial^2}{\partial h_a\partial h_b}}\prod_a 2\cosh(h_a)\Bigg|_{\{h_a=0\}}.$$

For the integration over $\varepsilon_{ab}$, the saddle point $\varepsilon^*_{ab} = \varepsilon^*_{ab}[\hat{q}]$ is obtained formally as,

$$q_{ab} = -\frac{\delta}{\delta\varepsilon_{ab}}\ln e^{-\frac{1}{2}\sum_{a,b}\varepsilon_{ab}\frac{\partial^2}{\partial h_a\partial h_b}}\prod_a 2\cosh(h_a)\Bigg|_{\{h_a=0\}}\Bigg|_{\varepsilon_{ab}=\varepsilon^*_{ab}[\hat{q}]} \qquad (a\neq b). \qquad (68)$$

## A.3 Evaluation of interaction part of the free-energy

Now we wish to evaluate the partition function Eq. (37) using the tools developed above. What we are doing below is the evaluation of the interaction part of the free-energy Eq. (54) in the program outlined in sec. A.1.4.

In Eq. (58) we notice that different spin components $\mu$ are decoupled in the average $\prod_\mu\langle\ldots\rangle_{S^\mu}$. Then we obtain the following cumulant expansion which will become very useful. For any observable $A = A(S^\mu)$ and writing $\langle\ldots\rangle_{S^\mu} = \langle\ldots\rangle$ for simplicity we find,

$$\ln\langle e^{\frac{1}{\sqrt{M}}\sum_{\mu=1}^M A_\mu}\rangle \;=\; \sqrt{M}\langle A_\mu\rangle + \frac{1}{2!}(\langle A_\mu^2\rangle - \langle A_\mu\rangle^2)$$

$$+\frac{1}{3!\sqrt{M}}(\langle A_\mu^3\rangle - 3\langle A_\mu^2\rangle\langle A_\mu\rangle + 2\langle A_\mu\rangle^3) + \ldots. \qquad (69)$$

Here we just used the fact that $\langle A^\mu A^\nu\rangle = \langle A^\mu\rangle\langle A^\nu\rangle$ holds for $\mu\neq\nu$. Thus in the $M\to\infty$, the lowest non-vanishing cumulant dominates the r.h.s.. For instance if $\langle A_\mu\rangle = 0$ and $\langle A_\mu^2\rangle \neq 0$, then $\lim_{M\to\infty}\ln\langle e^{\frac{1}{\sqrt{M}}\sum_{\mu=1}^M A_\mu}\rangle = \frac{1}{2!}\langle A_\mu^2\rangle$. Note that the same property also holds for the averaging in the 'bond space' $\langle\ldots\rangle_{J^i}$ in Eq. (57).

Now we are ready to evaluate Gardner's volume Eq. (14). We find, introducing a small

parameter $\lambda$,

$$
\ln\left\langle \prod_{\mu,\blacksquare,a} \exp\left[ i\eta_{\mu,\blacksquare,a} \sum_{i=1}^{N} \sqrt{\frac{\lambda}{N}} (J_\blacksquare^i)^a (S_\blacksquare^\mu)^a (S_{\blacksquare(i)}^\mu)^a \right] \right\rangle_{J^i,S^\mu}
$$

$$
= \ln\left\langle 1 + \sum_a \sum_\blacksquare \sqrt{\frac{\lambda}{N}} \sum_{i,\mu} i\eta_{\mu,\blacksquare,a} (J_\blacksquare^i)^a (S_\blacksquare^\mu)^a (S_{\blacksquare(i)}^\mu)^a \right.
$$

$$
\left. + \frac{1}{2!} \sum_{a,b} \sum_{\blacksquare,\blacksquare'} \frac{\lambda}{N} \sum_{i,j,\mu,\nu} i\eta_{\mu,\blacksquare,a} i\eta_{\nu,\blacksquare',b} (J_\blacksquare^i)^a (J_{\blacksquare'}^j)^b (S_\blacksquare^\mu)^a (S_{\blacksquare'}^\nu)^b (S_{\blacksquare(i)}^\mu)^a (S_{\blacksquare'(j)}^\nu)^b + \dots \right\rangle_{J^i,S^\mu}
$$

$$
= \frac{1}{2!} \sum_{a,b} \sum_{\blacksquare,\blacksquare'} \frac{\lambda}{N} \sum_{i,j,\mu,\nu} i\eta_{\mu,\blacksquare,a} i\eta_{\nu,\blacksquare',b} \delta_{\blacksquare,\blacksquare'} \delta_{ij} \delta_{\mu,\nu} Q_{ab,\blacksquare} q_{ab,\blacksquare} q_{ab,\blacksquare(i)}
$$

$$
+ \frac{1}{4!} \sum_{a,b,c,d} \sum_{\blacksquare_1,\blacksquare_2,\blacksquare_3,\blacksquare_4} \left(\frac{\lambda}{N}\right)^2 \sum_{i,j,k,l,\mu_1,\mu_2,\mu_3,\mu_4} i\eta_{\mu_1,\blacksquare_1,a} i\eta_{\mu_2,\blacksquare_2,b} i\eta_{\mu_4,\blacksquare_3,c} i\eta_{\mu_4,\blacksquare_4,d}
$$

$$
\times \delta_{\blacksquare_1,\blacksquare_2} \delta_{\blacksquare_1,\blacksquare_3} \delta_{\blacksquare_1,\blacksquare_4} \delta_{ij} \delta_{ik} \delta_{il} \delta_{\mu_1,\mu_2} \delta_{\mu_1,\mu_3} \delta_{\mu_1,\mu_4} \times
$$

$$
\left[ \langle (J_\blacksquare^i)^a (J_\blacksquare^i)^b (J_\blacksquare^i)^c (J_\blacksquare^i)^d \rangle_{J^i} \langle (S_\blacksquare^\mu)^a (S_\blacksquare^\mu)^b (S_\blacksquare^\mu)^c (S_\blacksquare^\mu)^d \rangle_{S^\mu} \langle (S_{\blacksquare(i)}^\mu)^a (S_{\blacksquare(i)}^\mu)^b (S_{\blacksquare(i)}^\mu)^c (S_{\blacksquare(i)}^\mu)^d \rangle_{S^\mu} \right.
$$

$$
- Q_{ab,\blacksquare} Q_{cd,\blacksquare} q_{ab,\blacksquare} q_{cd,\blacksquare} q_{ab,\blacksquare(i)} q_{cd,\blacksquare(i)} - Q_{ac,\blacksquare} Q_{bd,\blacksquare} q_{ac,\blacksquare} q_{bd,\blacksquare} q_{ac,\blacksquare(i)} q_{bd,\blacksquare(i)}
$$

$$
\left. - Q_{ad,\blacksquare} Q_{bc,\blacksquare} q_{ad,\blacksquare} q_{bc,\blacksquare} q_{ad,\blacksquare(i)} q_{bc,\blacksquare(i)} \right] + \dots + \text{``loop correction''}
$$

$$
= \frac{1}{2} \sum_{a,b} \sum_\blacksquare \sum_\mu i\eta_{\mu,\blacksquare,a} i\eta_{\mu,\blacksquare,b} q_{ab,\blacksquare} Q_{ab,\blacksquare} \frac{\lambda}{N} \sum_{i=1}^{N} q_{ab,\blacksquare(i)} + \text{``loop correction''}. \tag{70}
$$

In the last equation we assumed $N \gg 1$ by which only the 2nd order term in the cumulant survives. The contribution of 4th order term of order $O(\lambda^2)$ writtten explicitly above is smaller than the 2nd order term by a factor $O(\lambda/N)$ so that it can be neglected in the $N \to \infty$ limit.

However, it is easy to realize that the correction due to the interaction loop shown in Fig. 2 make a contribution of order $O(\lambda^3)$ and it does *not* vanish in the $N \to \infty$ limit. In the present paper, we invoke a tree-approximation discarding correction terms due to such loops and more extended ones. Within this approximation, the expansion of the free-energy Eq. (51) stops at $O(\lambda)$ so that Eq. (54) gives the free-energy.

## A.4 Total free-energy

To sum up, we find,

$$
V^n(\mathbf{S}_0, \mathbf{S}_L) = \int \prod_{a<b} \left( \prod_\blacksquare dQ_{ab,\blacksquare} e^{NS_{\text{ent,bond}}[\hat{Q}_\blacksquare]} \prod_{\blacksquare \backslash \text{output}} dq_{ab,\blacksquare} e^{MS_{\text{ent,spin}}[\hat{q}_\blacksquare]} \right)
$$

$$
\prod_\blacksquare \left\{ \left( \prod_a \int \frac{d\eta_a}{\sqrt{2\pi}} W_{\eta_a} \right) e^{-\frac{1}{2}\sum_{a,b} \eta_a \eta_b Q_{ab,\blacksquare} q_{ab,\blacksquare} \frac{\sum_{i=1}^N q_{ab,\blacksquare(i)}}{N}} \right\}^M
$$

$$
= \int \prod_{a<b} \left( \prod_\blacksquare dQ_{ab,\blacksquare} e^{NS_{\text{ent,bond}}[\hat{Q}_\blacksquare]} \prod_{\blacksquare \backslash \text{output}} dq_{ab,\blacksquare} e^{MS_{\text{ent,spin}}[\hat{q}_\blacksquare]} \right)
$$

$$
\prod_\blacksquare \left\{ e^{-\frac{1}{2}\sum_{a,b} \frac{\partial^2}{\partial h_{\blacksquare,a} \partial h_{\blacksquare,b}} Q_{ab,\blacksquare} q_{ab,\blacksquare} \frac{\sum_{i=1}^N q_{ab,\blacksquare(i)}}{N}} \prod_a e^{-\beta V(h_{\blacksquare,a})} \bigg|_{h_{\blacksquare,a}=0} \right\}^M. \tag{71}
$$

Given the structure of the network, it is natural to assume that the saddle point values only depend on the layer $l = 0, 1, 2, \ldots, L$, as Eq. (16),

$$Q^*_{ab,\blacksquare} = Q_{ab}(l), \qquad q^*_{ab,\blacksquare} = q_{ab}(l). \tag{72}$$

Then we find,

$$s_n[\{\hat{Q}(l), \hat{q}(l)\}] = \frac{1}{\alpha} \sum_{l=1}^{L} s_{\text{ent,bond}}[\hat{Q}(l)] + \sum_{l=1}^{L-1} s_{\text{ent,spin}}[\hat{q}(l)] - \sum_{l=1}^{L} \mathcal{F}_{\text{int}}[\hat{q}(l-1), \hat{Q}(l), \hat{q}(l)], \tag{73}$$

with

$$-\mathcal{F}_{\text{int}}[\hat{q}(l-1), \hat{Q}(l), \hat{q}(l)] = \ln e^{\frac{1}{2} \sum_{ab} q_{ab}(l-1)Q_{ab}(l)q_{ab}(l)\partial_{h_{l,a}}\partial_{h_{l,b}}} \prod_{a=1}^{n} e^{-\beta V(h_{l,a})} \Big|_{h_{l,a}=0}. \tag{74}$$

The order parameters must verify saddle point equations,

$$\frac{\partial}{\partial Q_{ab}(l)} \partial_n s_n[\{\hat{Q}(l), \hat{q}(l)\}] \Big|_{n=0} = 0 \qquad l = 1, 2, \ldots, L, \tag{75}$$

$$\frac{\partial}{\partial q_{ab}(l)} \partial_n s_n[\{\hat{Q}(l), \hat{q}(l)\}] \Big|_{n=0} = 0 \qquad l = 1, 2, \ldots, L-1. \tag{76}$$

## A.5 Parisi's ansatz

### A.5.1 Random inputs/outputs

In the case of random inputs/outputs (sec. 2.2.1) we have $n$ replicas $a = 1, 2, \ldots, n$. Then it is natural to consider the standard Parisi's ansatz with $k$-step RSB (including RS as $k = 0$ and continuous RSB as $k = \infty$) [13, 14] (See Fig. 19),

$$Q_{ab}(l) = \sum_{i=0}^{k+1} Q_i(l)(I^{m_i}_{ab} - I^{m_{i+1}}_{ab}) \qquad l = 1, 2, \ldots, L, \tag{77}$$

$$q_{ab}(l) = \sum_{i=0}^{k+1} q_i(l)(I^{m_i}_{ab} - I^{m_{i+1}}_{ab}) \qquad l = 1, 2, \ldots, L-1, \tag{78}$$

$$\varepsilon_{ab}(l) = \sum_{i=0}^{k} \varepsilon_i(l)(I^{m_i}_{ab} - I^{m_{i+1}}_{ab}) \qquad l = 1, 2, \ldots, L-1, \tag{79}$$

where $I^m_{ab}$ is a generalized ('fat') Identity matrix of size $n \times n$ composed of blocks of size $m \times m$. Here we aassumed $q_{k+1}(l) = Q_{k+1}(l) = 1$ and $I^{m_{k+2}}_{ab} = 0$. In the Parisi's ansatz one considers

$$1 = m_{k+1} < m_k < \ldots < m_1 < m_0 = n, \tag{80}$$

which becomes

$$0 = m_0 < m_1 < \ldots < m_k < m_{k+1} = 1 \tag{81}$$

in the $n \to 0$ limit. In the $k \to \infty$ limit, the matrix elements can be parametrized by functions $q(x, l)$, $Q(x, l)$ and $\epsilon(x, l)$ defined in the range $0 \le x \le 1$ (See Fig. 19 d)).

The order parameter functions encode characteristics of the complex free-energy landscape [14]. For example the distribution functions of the overlaps between two replicas (two independent machines) can be related to the order parameter functions as,

$$P(q, l) = \frac{dx(q, l)}{dq} \qquad x(q, l) = \int_0^q dq' P(q', l),$$

$$P(Q, l) = \frac{dx(Q, l)}{dQ} \qquad x(Q, l) = \int_0^Q dQ' P(Q', l). \tag{82}$$

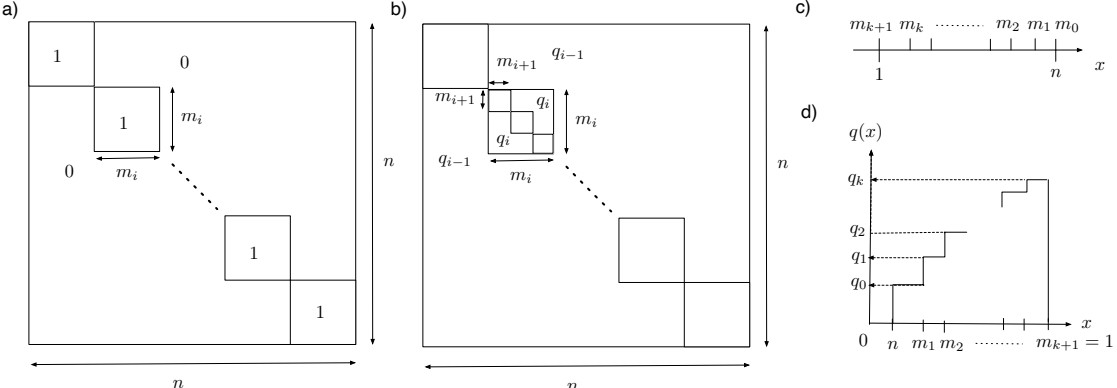

Figure 19: Parametrization of the Parisi's matrix a) the 'fat' identity matrix $I_{ab}^{m_i}$ b) Parisi's matrix given by Eq. (78) (Eq. (77) and Eq. (79) have the same structure with $q_i$'s replaced by $Q_i$'s and $\epsilon_i$'s.) c) the hierarchy of the sizes $m_i$ of the sub-matrices d) the $q(x)$ function with $0 < n < 1$ ($Q(x)$,$\epsilon(x)$ functions have the same structure).

Thus $x(q,l)$ ($x(Q,l)$) is the probability that the mutual overlap of the spin (bond) patterns at $l$-th layer between two machine are smaller than $q$ ($Q$). Equivalently $1-x(q,l)$ ($1-x(Q,l)$) is the probability that the mutual overlap of the spin (bond) patterns at $l$-th layer between two machine are *larger* than $q$ ($Q$).

The functions $q(x,l)$ and $Q(x,l)$ are expected to have a 'plateau' close to $x = 1$, which gives rise to a delta function in the overlap distribution functions $P(q,l)$ and $P(Q,l)$. As usual we regard the plateau values as the self-overlaps of the meta-stable states or the Edwards-Anderson order parameters $q_{\text{EA}}(l)$ and $Q_{\text{EA}}(l)$. In practice we will use the values of $q_k$ and $Q_k$ in the $k$-RSB ansatz as the Edwards-Anderson order parameters.

Analysis of the free-energy functional $-\beta F[\{\hat{Q}(l),\hat{q}(l)\}]/MN$ Eq. (17) can be done using these matrices in Eq. (73). In appendix B, we present details of the RSB solution.

### A.5.2  Teacher-student setting

For the teacher-student setting (sec. 2.2.2) we have to modify the matrices $\hat{Q}$,$\hat{q}$ and $\hat{\epsilon}$ slightly to include $a = 0$ for the teacher machine in addition to $a = 1, 2, \dots, s$ for the student as shown in Fig. 20. We denote the modified matrices as $\hat{Q}^{1+s}$,$\hat{q}^{1+s}$ and $\hat{\epsilon}^{1+s}$. The sub-matrices $\hat{q}^s$, $\hat{Q}^s$ and $\hat{\epsilon}^s$ are for the student for which we assume the same hierarchical structure as before Eq. (77)-Eq. (79).

Analysis of the Franz-Parisi potential $-\beta F_{\text{teacher-student}}[\{\hat{Q}(l),\hat{q}(l)\}]/MN$ Eq. (19) can be done using these matrices in Eq. (73). In appendix C, we present details of the RSB solution.

## B   RSB solution for the random inputs/outputs

.

Here we derive the RSB solution using the Parisi's ansatz explained in sec. A.5.1.

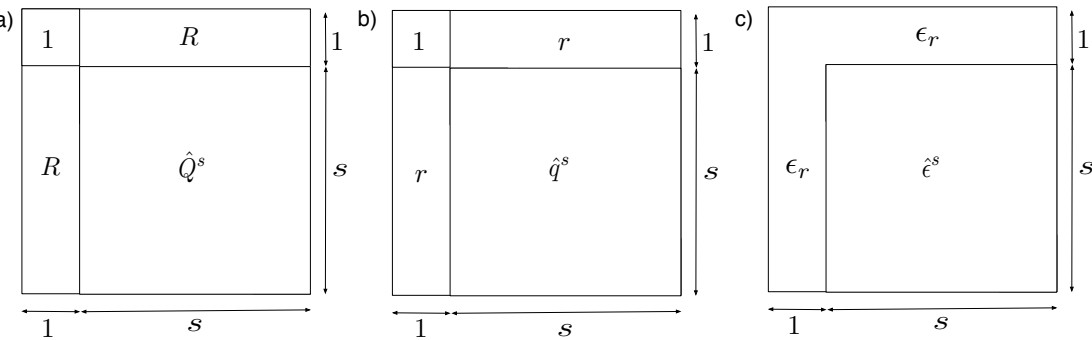

Figure 20: Parametrization of the Parisi's matrices for the teacher-student setting : a) $\hat{Q}^{1+s}$ b) $\hat{q}^{1+s}$ and c) $\hat{\epsilon}^{1+s}$. For the sub-matrices $\hat{q}^s$, $\hat{Q}^s$ and $\hat{\epsilon}^s$ of size $s \times s$ we assume the same hierarchical structure as those in Eq. (77)-Eq. (79) (see Fig. 19) but with $n$ replaced by $s$.

## B.1 Entropic part of the free-energy

### B.1.1 Entropic part of the free-energy due to 'bonds'

In the $k$-RSB ansatz, the entropic part of the free-energy Eq. (66) due to the 'bonds' can be evaluated as follows. We find [20, 80],

$$
\ln \det \hat{Q} = \ln \left( 1 + \sum_{j=0}^{k} (m_j - m_{j+1}) Q_j \right) \tag{83}
$$

$$
+ \; n \sum_{i=0}^{k} \left( \frac{1}{m_{i+1}} - \frac{1}{m_i} \right) \ln \left( 1 + \sum_{j=i}^{k} (m_j - m_{j+1}) Q_j - m_i Q_i \right). \tag{84}
$$

Remembering that $m_0 = n$ we find,

$$
\partial_n S_{\text{ent,bond}}[\hat{Q}] \Big|_{n=0} = \frac{1}{2} \, \partial_n \ln \det \hat{Q} \Big|_{n=0} = \frac{1}{2} \frac{Q_0}{G_0} + \frac{1}{2} \frac{1}{m_1} \ln G_0
$$

$$
+ \; \frac{1}{2} \sum_{i=1}^{k} \left( \frac{1}{m_{i+1}} - \frac{1}{m_i} \right) \ln G_i, \tag{85}
$$

with

$$
G_i = 1 + \sum_{j=i}^{k} (m_j - m_{j+1}) Q_j - m_i Q_i \qquad i = 0, 1, \ldots, k, \tag{86}
$$

which implies

$$
Q_i = 1 - G_k + \sum_{j=i+1}^{k} \frac{1}{m_j} (G_j - G_{j-1}) \qquad i = 0, 1, \ldots, k. \tag{87}
$$

### B.1.2 Entropic part of the free-energy due to 'spins'

In the $k$-RSB ansatz, the entropic part of the free-energy Eq. (67) due to the 'spins' can be evaluated as follows

$$
\begin{aligned}
S_{\text{ent,spin}}[\hat{\epsilon}, \hat{q}] &= \frac{n}{2} \sum_{i=0}^{k} \epsilon_i q_i (m_i - m_{i+1}) + \frac{n}{2} \epsilon_k \\
&\quad + \ln \prod_{i=0}^{k} \exp \left[ \frac{\Lambda_i^{\text{Ising}}}{2} \sum_{a,b=1}^{n} I_{ab}^{m_i} \frac{\partial^2}{\partial h_a \partial h_b} \right] \prod_{a=1}^{n} (2 \cosh(h_a)) \Bigg|_{\{h_a=0\}}, \quad (88)
\end{aligned}
$$

which implies

$$
\begin{aligned}
\partial_n S_{\text{ent,spin}}[\hat{\epsilon}, \hat{q}] \Big|_{n=0} &= \frac{1}{2} \sum_{i=0}^{k} \epsilon_i q_i (m_i - m_{i+1}) + \frac{\epsilon_k}{2} - f_{\text{Ising}}(m_0 = 0, 0) \\
&= \frac{1}{2} \sum_{i=0}^{k} \epsilon_i q_i (m_i - m_{i+1}) + \frac{\epsilon_k}{2} - \int Dz_0 f_{\text{Ising}}(m_1, \sqrt{\Lambda_0^{\text{Ising}}} z_0), \quad (89)
\end{aligned}
$$

where $\epsilon_i$'s must be fixed through saddle point equations with respect to variations of them (see below).

In the last two-equation of Eq. (89) we used a family of functions which can be obtained recursively as follows [81]. Using

$$
\Lambda_i^{\text{Ising}} = \begin{cases} -\epsilon_0 & (i = 0) \\ -\epsilon_i + \epsilon_{i-1} & (i = 1, 2, \ldots, k) \end{cases}, \quad (90)
$$

we introduce a family of functions defined recursively for $i = 0, 1, 2, \ldots, k$,

$$
\begin{aligned}
e^{-m_i f_{\text{Ising}}(m_i, h)} &= e^{\frac{\Lambda_i^{\text{Ising}}}{2} \frac{\partial^2}{\partial h^2}} e^{-m_i f_{\text{Ising}}(m_{i+1}, h)} \\
&= \gamma_{\Lambda_i^{\text{Ising}}} \otimes e^{-m_i f(m_{i+1}, h)} \\
&= \int Dz_i e^{-m_i f_{\text{Ising}}(m_{i+1}, h - \sqrt{\Lambda_i^{\text{Ising}}} z_i)}, \quad (91)
\end{aligned}
$$

with the initial condition

$$
f_{\text{Ising}}(m_{k+1}, h) = -\ln 2 \cosh(h). \quad (92)
$$

Here we used an identity

$$
\exp \left( \frac{a}{2} \frac{\partial^2}{\partial h^2} \right) A(h) = \gamma_a \otimes A(h) \quad (93)
$$

and the following short hand notations: $\gamma_a(x)$ is a Gaussian

$$
\gamma_a(x) = \frac{1}{\sqrt{2\pi a}} e^{-\frac{x^2}{2a}}, \quad (94)
$$

by which we write a convolution of a function $A(x)$ with the Gaussian as,

$$
\gamma_a \otimes A(x) \equiv \int dy \frac{e^{-\frac{y^2}{2a}}}{\sqrt{2\pi a}} A(x - y) = \int \mathcal{D}z A(x - \sqrt{a} z), \quad (95)
$$

where

$$\int \mathcal{D}z \ldots \equiv \int dz \frac{e^{-\frac{z^2}{2}}}{\sqrt{2\pi}} \cdots . \tag{96}$$

The saddle point equation with respect to variations of $\hat{\epsilon}$ Eq. (68) becomes in the $k$-RSB ansatz, for $i = 0, 1, 2, \ldots, k$,

$$\begin{aligned}
q_i &= \frac{2}{m_i - m_{i+1}} \left[ \left( -\frac{\partial}{\partial \epsilon_i} \right)(-f_{\text{Ising}}(m_0 = 0, 0)) - \frac{1}{2}\delta_{ik} \right] \\
&= \int dh P_{\text{Ising}}(m_i, h)(-f'_{\text{Ising}}(m_i + 1, h))^2, \tag{97}
\end{aligned}$$

where $f'_{\text{Ising}}(m, h) = \partial_h f_{\text{Ising}}(m, h)$ and we used (see [20] appendix C)

$$\left( -\frac{\partial}{\partial \epsilon_i} \right)(-f_{\text{Ising}}(m_0 = 0, 0)) = \frac{1}{2}(m_i - m_{i+1}) \int dh P_{\text{Ising}}(m_i, h)(-f'_{\text{Ising}}(m_{i+1}, h))^2 + \frac{1}{2}\delta_{i,k}, \tag{98}$$

with

$$P_{\text{Ising}}(m_i, h) \equiv \frac{\delta f_{\text{Ising}}(m_0, 0)}{\delta f_{\text{Ising}}(m_{i+1}, h)}, \tag{99}$$

which follows a recursion formula (see [20] sec. 8.3.1),

$$P_{\text{Ising}}(m_j, h) = e^{-m_j f_{\text{Ising}}(m_{j+1}, h)} \gamma_{\Lambda_j^{\text{Ising}}} \otimes_h \frac{P_{\text{Ising}}(m_{j-1}, h)}{e^{-m_j f_{\text{Ising}}(m_j, h)}} \qquad j = 1, 2, \ldots, k, \tag{100}$$

with the 'boundary condition'

$$P_{\text{Ising}}(m_0, h) = \frac{1}{\sqrt{2\pi\Lambda_0^{\text{Ising}}}} e^{-\frac{h^2}{2\Lambda_0}}. \tag{101}$$

In Eq. (100) $\otimes_h$ stands for a convolution with respect to the variable $h$.

The derivatives $f'_{\text{Ising}}(m, h) = \partial_h f_{\text{Ising}}(m, h)$ can also be obtained recursively. From Eq. (91) and Eq. (92) we find,

$$f'_{\text{Ising}}(m_i, h) = e^{m_i f_{\text{Ising}}(m_i, h)} \gamma_{\Lambda_i^{\text{Ising}}} \otimes f'_{\text{Ising}}(m_{i+1}, h) e^{-m_i f_{\text{Ising}}(m_{i+1}, h)}, \tag{102}$$

for $i = 1, 2, \ldots, k$ with the 'boundary condition',

$$f'_{\text{Ising}}(m_{k+1}, h) = -\tanh(h). \tag{103}$$

## B.2 Interaction part of the free-energy

The interaction part of the free-energy Eq. (74) becomes in the $k$-RSB ansatz,

$$-\partial_n \mathcal{F}_{\text{int}} \left[ \hat{q}(l-1), \hat{Q}(l), \hat{q}(l) \right] \Big|_{n=0} \tag{104}$$

$$= \partial_n \ln \prod_{i=0}^{k+1} \exp \left[ \frac{\Lambda_i(l)}{2} \sum_{a,b=1}^{n} I_{ab}^{m_i} \frac{\partial^2}{\partial h_a \partial h_b} \right] \prod_{a=1}^{n} e^{-\beta V(r(h_a))} \Bigg|_{\{h_a = 0\}} \Bigg|_{n=0}$$

$$= f(m_0 = 0, 0, l)$$

$$= \int \mathcal{D}z_0 f(m_1, h - \sqrt{\Lambda_0}, l) \Big|_{h=0}, \tag{105}$$

with, for $l = 1, 2, \ldots, L$,

$$\Lambda(l) = \begin{cases} \lambda_0(l) & (i = 0), \\ \lambda_i(l) - \lambda_{i-1}(l) & (i = 1, 2, \ldots, k+1) \end{cases} \tag{106}$$

and

$$\lambda_i(l) = q_i(l-1)Q_i(l)q_i(l). \tag{107}$$

We introduced a family of functions defined recursively for $i = 0, 1, 2, \ldots, k$ and $l = 1, 2, \ldots, L$,

$$
\begin{aligned}
e^{-m_i f(m_i, h, l)} &= e^{\frac{\Lambda_i(l)}{2} \frac{\partial^2}{\partial h^2}} e^{-m_i f(m_{i+1}, h, l)} \\
&= \int Dz_i e^{-m_i f(m_{i+1}, h - \sqrt{\Lambda_i(l)} z_i, l)},
\end{aligned}
\tag{108}
$$

with the initial condition

$$f(m_{k+1}, l) = -\ln \gamma_{\Lambda_{k+1}(l)} \otimes e^{-\beta V(h)} = -\ln \int Dz_{k+1} e^{-\beta V(h - \sqrt{\Lambda_{k+1}(l)} z_{k+1})}. \tag{109}$$

For the hard-core potential Eq. (7) we find,

$$f(m_{k+1}, h, l) = -\ln \Theta\left( \frac{h}{\sqrt{2\Lambda_{k+1}(l)}} \right), \tag{110}$$

where

$$\Theta(x) = \int_{-\infty}^{x} \frac{dy}{\sqrt{\pi}} e^{-y^2}. \tag{111}$$

## B.3 Saddle point equations

### B.3.1 Variation of $q_i(l)$'s

The saddle point equations Eq. (76) becomes, for $i = 0, 1, 2, \ldots, k$ and $l = 1, 2, \ldots, L-1$,

$$
\begin{aligned}
0 &= \frac{\partial}{\partial q_i(l)} \partial_n S_n^{\text{ent,spin}}[\hat{q}, \hat{Q}] \Big|_{n=0} \\
&= \frac{1}{2} \epsilon_i(l)(m_i - m_{i+1}) - \frac{\partial}{\partial q_i(l)} \sum_{l'=1}^{L} \mathcal{F}_{\text{int}}[\hat{q}(l'-1), \hat{Q}(l'), \hat{q}(l')] \\
&= \frac{1}{2} \epsilon_i(l)(m_i - m_{i+1}) + \sum_{l'=1}^{L} \frac{\partial \lambda_i(l')}{\partial q_i(l)} \left( -\frac{\partial f(m_0 = 0, 0, l')}{\partial \lambda_i(l')} \right),
\end{aligned}
\tag{112}
$$

from which we find,

$$
\begin{aligned}
\epsilon_i(l) &= -\sum_{l'=1}^{L} \frac{\partial \lambda_i(l')}{\partial q_i(l)} \kappa_i(l') \\
&= -q_i(l-1)Q_i(l)\kappa_i(l) - Q_i(l+1)q_i(l+1)\kappa_i(l+1),
\end{aligned}
\tag{113}
$$

where we introduced, for $i = 0, 1, 2, \ldots, k$ and $l = 1, 2, \ldots, L$,

$$\kappa_i(l) \equiv \int dh P(m_i, h, l)(-f'(m_{i+1}, h, l))^2, \tag{114}$$

with

$$P(m_i, h, l) \equiv \frac{\delta f(m_0, 0, l)}{\delta f(m_{i+1}, h, l)}, \tag{115}$$

which follows a recursion formula (see [20] sec. 8.3.1),

$$P(m_j, h, l) = e^{-m_j f(m_{j+1}, h, l)} \gamma_{\Lambda_j} \otimes_h \frac{P(m_{j-1}, h, l)}{e^{-m_j f(m_j, h, l)}} \qquad j = 1, 2, \dots, k+1, \tag{116}$$

with the 'boundary condition'

$$P(m_0, h, l) = \frac{1}{\sqrt{2\pi\Lambda_0(l)}} e^{-\frac{h^2}{2\Lambda_0(l)}}. \tag{117}$$

The last equation of Eq. (112) is obtained using the following (see [20] appendix C)

$$\frac{\partial}{\partial \lambda_i(l)}(-f(m_0 = 0, 0, l)) = \frac{1}{2}(m_i - m_{i+1}) \int dh P(m_i, h, l)(-f'(m_{i+1}, h, l))^2. \tag{118}$$

The derivatives $f'(m, h, l) = \partial_h f(m, h, l)$ can also be obtained recursively. From Eq. (108) and Eq. (109) we find,

$$f'(m_i, h, l) = e^{m_i f(m_i, h, l)} \gamma_{\Lambda_i(l)} \otimes f'(m_{i+1}, h, l) e^{-m_i f(m_{i+1}, h, l)}, \tag{119}$$

for $i = 1, 2, \dots, k$ with the 'boundary condition',

$$f'(m_{k+1}, h, l) = -\frac{\int Dz_{k+1}(d/dh)(e^{-\beta V(h-\sqrt{\Lambda_{k+1}(l)})})}{\int Dz_{k+1} e^{-\beta V(h-\sqrt{\Lambda_{k+1}(l)})}}, \tag{120}$$

which becomes for the hardcore potential (using Eq. (110) and Eq. (111)),

$$f'(m_{k+1}, h, l) = -\frac{1}{\Theta\left(\frac{h}{\sqrt{2\Lambda_{k+1}(l)}}\right)} \frac{1}{\sqrt{2\pi\Lambda_{k+1}(l)}} \exp\left(-\frac{h^2}{2\Lambda_{k+1}(l)}\right). \tag{121}$$

### B.3.2   Variation of $G_i(l)$'s

For the saddle point equations Eq. (75) it is convenient to consider instead, for $i = 0, 1, 2, \dots, k$ and $l = 1, 2, \dots, L$,

$$0 = \frac{\partial}{\partial G_i(l)} \partial_n S[\hat{q}, \hat{Q}]\Big|_{n=0}, \tag{122}$$

where $G_i(l)$'s are defined in Eq. (86). We obtain (see [20] sec. 8.4),

$$\frac{Q_0(l)}{G_0^2(l)} = \alpha q_0(l) q_0(l-1) \kappa_0(l)$$

$$\frac{1}{G_i(l)} - \frac{1}{G_0(l)} = \alpha \left( \sum_{j=0}^{i-1} (m_j - m_{j+1}) q_j(l) q_j(l-1) \kappa_j(l) + m_i q_i(l) q_i(l-1) \kappa_i(l) \right), \tag{123}$$

for $i = 1, 2, \dots, k$.

### B.3.3 Procedure to solve the saddle point equations

The saddle point equations for a generic finite $k$-RSB ansatz with some fixed values of $0 < m_1 < m_2 < \ldots < m_k < 1$ can be solved numerically as follows.

0. Choose a boundary condition by fixing $q_i(0)$ and $q_i(L)$ for $i = 0, 1, 2, \ldots, k$.

1. Make some guess for the initial values of $q_i(l)$ ($l = 1, 2, \ldots, L - 1$) and $Q_i(l)$ ($l = 1, 2, \ldots, L$) for $i = 0, 1, 2, \ldots, k$. Then compute $G_i(l)$ for $i = 0, 1, \ldots, k$ and $l = 1, 2, \ldots, L - 1$ using Eq. (86).

2. Do the following (1)-(8) for $l = 1, 2, \ldots, L$. (1) Compute $\lambda_i(l)$ for $i = 0, 1, 2, \ldots, k$ and using Eq. (107). (2) Compute $\Lambda_i(l)$ for $i = 0, 1, 2, \ldots, k+1$ using Eq. (106). (3) Compute functions $f(m_i, h, l)$ recursively for $i = k, k-1, \ldots, 0$ using Eq. (108) with the boundary condition given by Eq. (109) (which is Eq. (110) for the hardcore potential). (4) Compute also the derivatives $f'(m_i, h, l)$ recursively for $i = k, k-1, \ldots, 2, 1$ using Eq. (119) with the boundary condition given by Eq. (120) (which is Eq. (121) for the hardcore potential). (5) Compute functions $P(m_i, h, l)$ recursively for $i = 1, \ldots, k$ using Eq. (116) with the boundary condition given by Eq. (117). (6) Compute $\kappa_i(l)$ for $i = 0, 1, \ldots, k$ using Eq. (114). (7) Compute $G_i(l)$ for $i = 0, 1, \ldots, k$ using Eq. (123). (8) Compute $Q_i(l)$ for $i = 0, 1, \ldots, k$ using Eq. (87).

3. Do the following (1)-(6) for $l = 1, 2, \ldots, L - 1$. (1) Compute $\epsilon_i(l)$ for $i = 0, 1, 2, \ldots, k$ using Eq. (113). (2) Compute $\Lambda_i^{\mathrm{Ising}}(l)$ for $i = 0, 1, 2, \ldots, k$ using Eq. (90). (3) Compute functions $f_{\mathrm{Ising}}(m_i, h, l)$ recursively for $i = k, k-1, \ldots, 0$ using Eq. (91) with the boundary condition given by Eq. (92). (4) Compute also the derivatives $f'_{\mathrm{Ising}}(m_i, h, l)$ recursively for $i = k, k-1, \ldots, 2, 1$ using Eq. (102) with the boundary condition given by Eq. (103). (5) Compute functions $P_{\mathrm{Ising}}(m_i, h, l)$ recursively for $i = 1, \ldots, k$ using Eq. (100) with the boundary condition given by Eq. (101). (6) Compute $q_i(l)$ for $i = 0, 1, \ldots, k$ using Eq. (97).

4. Return to 2.

The above procedure 1.-4. must be repeated until the solution converges. The values of $m_i$s ($0 < m_1 < m_2 \ldots < m_k < 1$ (see Fig. 19 c))) are chosen such that $\log m_i$s are equally spaced between $\log m_1$ and $\log m_{k+1} = 0$. We chose $m_1 = 0.0001$ in the numerical analysis shown in this paper. Numerical integrations are done by the simple rectangle rule with an integration step 0.01.

## C  RSB solution for the teacher-student setting

Here we derive the RSB solution using the Parisi's ansatz explained in sec. A.5.2.

### C.1  Entropic part of the free-energy

#### C.1.1  Entropic part of the free-energy due to 'bonds'

Within the ansatz for the teacher-student setting, the entropic part of the free-energy Eq. (66) due to the 'bonds' can be evaluated as follows. First we find,

$$\det \hat{Q}^{1+s} = \det(\hat{Q}^s - R^2). \tag{124}$$

Thus we find

$$
\begin{aligned}
\partial_s S_{\text{ent,bond}}[\hat{Q}^{1+s}]\Big|_{s=0} &= \frac{1}{2}\partial_s \ln\det(\hat{Q}^s - R^2)\Big|_{s=0} \\
&= \frac{1}{2}\frac{Q_0 - R^2}{G_0} + \frac{1}{2}\frac{1}{m_1}\ln G_0 \\
&\quad + \frac{1}{2}\sum_{i=1}^{k}\left(\frac{1}{m_{i+1}} - \frac{1}{m_i}\right)\ln G_i,
\end{aligned}
\tag{125}
$$

with

$$
G_i = 1 + \sum_{j=i}^{k}(m_j - m_{j+1})Q_j - m_i Q_i \qquad i = 0,1,\ldots,k,
\tag{126}
$$

which implies

$$
Q_i = 1 - G_k + \sum_{j=i+1}^{k}\frac{1}{m_j}(G_j - G_{j-1}) \qquad i = 0,1,\ldots,k.
\tag{127}
$$

Note that above equations slightly modify Eq. (86) and Eq. (87).

### C.1.2 Entropic part of the free-energy due to 'spins'

Within the same ansatz, the entropic part of the free-energy Eq. (67) due to the 'spins' can be evaluated as follows,

$$
\begin{aligned}
S_{\text{ent,spin}}[\hat{\epsilon}^{1+s}, \hat{q}^{1+s}] &= s\epsilon_r r + \frac{1}{2}\epsilon_r + \frac{s}{2}\sum_{i=0}^{k}\epsilon_i q_i(m_i - m_{i+1}) + \frac{s}{2}\epsilon_k \\
&\quad + \ln\exp\left[\frac{\Lambda_{\text{com}}^{\text{Ising}}}{2}\sum_{a,b=0}^{s}\frac{\partial^2}{\partial h_a \partial h_b}\right]\prod_{i=0}^{k}\exp\left[\frac{\Lambda_i^{\text{Ising}}}{2}\sum_{a,b=1}^{s}I_{ab}^{m_i}\frac{\partial^2}{\partial h_a \partial h_b}\right]\prod_{a=0}^{s}(2\cosh(h_a))\Bigg|_{\{h_a=0\}} \\
&= s\epsilon_r r + \frac{1}{2}\epsilon_r + \frac{s}{2}\sum_{i=0}^{k}\epsilon_i q_i(m_i - m_{i+1}) + \frac{s}{2}\epsilon_k \\
&\quad + \ln\gamma_{\Lambda_{\text{com}}}\otimes(2\cosh(h)\gamma_{\Lambda_0^{\text{Ising}}}\otimes e^{-sf^{\text{Ising}}(m_1,h)}\Big|_{h=0},
\end{aligned}
\tag{128}
$$

where $\epsilon_r$ and $\epsilon_i$'s must be fixed through saddle point equations with respect to variations of them (see below).

In Eq. (128) $I_{ab}^{m_i}$ is defined similarly as those used in Eq. (77)-Eq. (79) (see Fig. 19) but with size $s \times s$ instead of $n \times n$. We have also introduced,

$$
\begin{aligned}
\Lambda_{\text{com}}^{\text{Ising}} &= -\epsilon_r, \\
\Lambda_i^{\text{Ising}} &= \begin{cases} -\epsilon_0 + \epsilon_r & (i = 0) \\ -\epsilon_i + \epsilon_{i-1} & (i = 1,2,\ldots,k) \end{cases}
\end{aligned}
\tag{129}
$$

and used the family of functions defined recursively for $i = 0,1,2,\ldots,k$ using Eq. (91) and the initial condition Eq. (92). One must keep in mind that $\Lambda_0^{\text{Ising}}$ in Eq. (129) is shifted with respect to that in Eq. (90) due to $\epsilon_r$.

The saddle point equations with respect to variations of $\hat{\epsilon}^{1+s}$ Eq. (68) yield $q_i$'s and $r$. Variation with respect to $\epsilon_i$ yields the equation for the $q_i$'s, which is formally the same as Eq. (97),

$$q_i = \int dh P_{\text{Ising}}(m_i, h)(-f'_{\text{Ising}}(m_i, h))^2, \tag{130}$$

for $i = 0, 1, 2, \dots, k$. Here $P_{\text{Ising}}(m_i, h)$ can be obtained from the equation Eq. (100). However the initial condition is modified from Eq. (101) to

$$P_{\text{Ising}}(m_0, h) = \frac{\int Dz_{\text{com}} 2\cosh(\sqrt{\Lambda^{\text{Ising}}_{\text{com}}} z_{\text{com}}) \frac{1}{\sqrt{2\pi\Lambda^{\text{Ising}}_0}} e^{-\frac{(h-\sqrt{\Lambda_{\text{com}}}z_{\text{com}})^2}{2\Lambda^{\text{Ising}}_0}}}{\int Dz_{\text{com}} 2\cosh(\sqrt{\Lambda^{\text{Ising}}_{\text{com}}} z_{\text{com}})}. \tag{131}$$

Variation with respect to $\epsilon_r$ yields the equation for the $r$ as the following. Using

$$0 = \frac{\partial}{\partial \epsilon_r} S_{\text{ent,spin}}[\hat{\epsilon}, \hat{q}^{1+s}] = sr + \frac{1}{2} - \frac{1}{2}$$
$$- s\frac{\int Dz_{\text{com}} 2\sinh(\sqrt{\Lambda_{\text{com}}} z_{\text{com}}) \int Dz_0(-f'_{\text{Ising}}(m_1, \sqrt{\Lambda_{\text{com}}} z_{\text{com}} + \sqrt{\Lambda^{\text{Ising}}_0} z_0))}{\int Dz_{\text{com}} 2\cosh(\sqrt{\Lambda_{\text{com}}} z_{\text{com}})} + O(s^2). \tag{132}$$

Thus we find

$$r = \frac{\int Dz_{\text{com}} 2\sinh(\sqrt{\Lambda_{\text{com}}} z_{\text{com}}) \int Dz_0(-f'_{\text{Ising}}(m_1, \sqrt{\Lambda_{\text{com}}} z_{\text{com}} + \sqrt{\Lambda^{\text{Ising}}_0} z_0))}{\int Dz_{\text{com}} 2\cosh(\sqrt{\Lambda_{\text{com}}} z_{\text{com}})}. \tag{133}$$

## C.2 Interaction part of the free-energy

Within the same ansatz, the interaction part of the free-energy Eq. (74) becomes,

$$\partial_s \mathcal{F}_{\text{int}}\left[\hat{q}^{1+s}(l-1), \hat{Q}^{1+s}(l), \hat{q}^{1+s}(l)\right]\Big|_{s=0} \tag{134}$$

$$= \partial_s \ln \exp\left[\frac{\Lambda_{\text{com}}(l)}{2} \sum_{a,b=0}^{s} \frac{\partial^2}{\partial h_a \partial h_b}\right] \exp\left[\frac{\Lambda_{\text{teacher}}(l)}{2} \frac{\partial^2}{\partial h_0^2}\right]$$

$$\prod_{i=0}^{k+1} \exp\left[\frac{\Lambda_i(l)}{2} \sum_{a,b=1}^{s} I_{ab}^{m_i} \frac{\partial^2}{\partial h_a \partial h_b}\right] \prod_{a=0}^{s} e^{-\beta V(r(h_a))}\Bigg|_{\{h_a=0\}}\Bigg|_{s=0}$$

$$= \partial_s \ln \int Dz_{\text{com}} \int Dz_{\text{teacher}} e^{-\beta V(\sqrt{\Lambda_{\text{com}}(l)}z_{\text{com}} + \sqrt{\Lambda_{\text{teacher}}(l)}z_{\text{teacher}})}$$

$$\int Dz_0 f(m_1, \sqrt{\Lambda_{\text{com}}(l)}z_{\text{com}} + \sqrt{\Lambda_0(l)}z_0)\Bigg|_{s=0}, \tag{135}$$

with, for $l = 1, 2, \dots, L$. Here we introduced,

$$\Lambda_{\text{com}}(l) = r(l-1)R(l)r(l),$$
$$\Lambda_{\text{teacher}}(l) = 1 - r(l-1)R(l)r(l),$$
$$\Lambda_i(l) = \begin{cases} \lambda_0(l) - \Lambda_{\text{com}}(l) & (i=0), \\ \lambda_i(l) - \lambda_{i-1}(l) & (i=1,2,\dots,k+1), \end{cases} \tag{136}$$

with $\lambda_i(l)$'s defined in Eq. (107) which reads as,

$$\lambda_i(l) = q_i(l-1)Q_i(l)q_i(l). \tag{137}$$

We also used the family of functions $f(m_i, h)$ defined recursively for $i = 0, 1, 2, \ldots, k$ and $l = 1, 2, \ldots, L$ by Eq. (108) with the initial condition Eq. (109). Note that $\Lambda_0$ is shifted with respect to that in Eq. (106) due to $R$ and $r$.

### C.3 Saddle point equations

#### C.3.1 Variation of $q_i(l)$'s

For the saddle point equations Eq. (76), we find formally the same result as Eq. (113) which reads as,

$$\epsilon_i(l) = -q_i(l-1)Q_i(l)\kappa_i(l) - Q_i(l+1)q_i(l+1)\kappa_i(l+1), \tag{138}$$

with $\kappa_i$ defined as Eq. (114) which reads as,

$$\kappa_i(l) \equiv \int dh P(m_i, h, l)(-f'(m_{i+1}, h, l))^2. \tag{139}$$

The function $P(m_i, h, l)$ can be also be obtained by the same equations as before Eq. (116) but with the initial condition Eq. (117) modified as,

$$P(m_0, h, l) = \frac{\int Dz_{\text{com}} \int Dz_{\text{teacher}} e^{-\beta V(\sqrt{\Lambda_{\text{com}}(l)}z_{\text{com}} + \sqrt{\Lambda_{\text{teacher}}(l)}z_{\text{teacher}})} \frac{1}{\sqrt{2\pi\Lambda_0(l)}} e^{-\frac{(h-\sqrt{\Lambda_{\text{com}}(l)}z_{\text{com}})^2}{2\Lambda_0(l)}}}{\int Dz_{\text{com}} \int Dz_{\text{teacher}} e^{-\beta V(\sqrt{\Lambda_{\text{com}}(l)}z_{\text{com}} + \sqrt{\Lambda_{\text{teacher}}(l)}z_{\text{teacher}})}}. \tag{140}$$

Note also that $\Lambda_0$ is shifted as in Eq. (136). For the hardcore potential Eq. (7) we find,

$$P(m_0, h, l) = \frac{1}{\int Dz \Theta\left(\frac{\sqrt{\Lambda_{\text{com}}(l)}z}{\sqrt{2(\Lambda_{\text{teacher}}(l))}}\right)} \int Dz \Theta\left(\frac{\sqrt{\Lambda_{\text{com}}(l)}z}{\sqrt{2\Lambda_{\text{teacher}}(l))}}\right) \frac{1}{\sqrt{2\pi\Lambda_0(l)}} e^{-\frac{(h-\sqrt{\Lambda_{\text{com}}(l)}z)^2}{2\Lambda_0(l)}}, \tag{141}$$

with $\Theta(h)$ defined in Eq. (111).

#### C.3.2 Variation of $G_i(l)$'s

For the saddle point equations Eq. (75), we just need to modify slightly Eq. (123), with $G_i$'s defined in Eq. (126),

$$\frac{Q_0(l) - R^2(l)}{G_0^2(l)} = \alpha q_0(l)q_0(l-1)\kappa_0(l) \tag{142}$$

$$\frac{1}{G_i(l)} - \frac{1}{G_0(l)} = \alpha \left( \sum_{j=0}^{i-1}(m_j - m_{j+1})q_j(l)q_j(l-1)\kappa_j(l) + m_i q_i(l)q_i(l-1)\kappa_i(l) \right),$$

for $l = 1, 2, \ldots, L$.

### C.3.3 Variation of $r$

Here we consider variation of the free-energy Eq. (19) (see also Eq. (73)) with respect to $r(l)$ , for $l = 1, 2, \ldots, L-1$,

$$
\begin{aligned}
0 &= \frac{\partial}{\partial r(l)} \, \partial_s s_{1+s}[\{\hat{Q}(l), \hat{q}(l)\}]\Big|_{s=0} \\
&= \frac{\partial}{\partial r(l)} \, \partial_s S_{\text{ent,spin}}[\hat{q}^{1+s}(l)]\Big|_{s=0} - \frac{\partial}{\partial r(l)} \sum_{l'=1}^{L} \partial_s \mathcal{F}_{\text{int}}[\hat{q}^{1+s}(l'-1), \hat{Q}^{1+s}(l'), \hat{q}^{1+s}(l')]\Big|_{s=0} .
\end{aligned}
$$
(143)

Variation of the entropic part (spin) of the free-energy Eq. (128) yields,

$$
\frac{\partial}{\partial r} \, \partial_s S_{\text{ent,spin}}[\hat{q}^{1+s}]\Big|_{s=0} = \epsilon_r .
$$
(144)

On the other hand, variation of the interaction part of the free-energy Eq. (135) yields, for $l = 1, 2, \ldots, L-1$,

$$
\begin{aligned}
&-\frac{\partial}{\partial r(l)} \sum_{l'=1}^{L} \partial_s \mathcal{F}_{\text{int}}[\hat{q}^{1+s}(l-1), \hat{Q}^{1+s}(l), \hat{q}^{1+s}(l)]\Big|_{s=0} \\
&= -\sum_{l'=1}^{L} \left( \frac{\partial \Lambda_{\text{com}}(l')}{\partial r(l)} \frac{\partial}{\partial \Lambda_{\text{com}}(l')} + \frac{\partial \Lambda_{\text{teacher}}(l')}{\partial r(l)} \frac{\partial}{\partial \Lambda_{\text{teacher}}(l')} + \frac{\partial \Lambda_0(l')}{\partial r(l)} \frac{\partial}{\partial \Lambda_0(l')} \right) \\
&\qquad \partial_s \mathcal{F}_{\text{int}}[\hat{q}^{1+s}(l-1), \hat{Q}^{1+s}(l), \hat{q}^{1+s}(l)]\Big|_{s=0} \\
&= r(l-1) R(l) \kappa_{\text{inter}}(l) + R(l+1) r(l+1) \kappa_{\text{inter}}(l+1),
\end{aligned}
$$
(145)

where we introduced

$$
\kappa_{\text{inter}}(l) \equiv \frac{\int Dz_{\text{com}} g'_{\text{teacher}}(\sqrt{\Lambda_{\text{com}}(l)} z_{\text{com}}) \int Dz_0 (-f'(m_1, \sqrt{\Lambda_{\text{com}}(l)} z_{\text{com}} + \sqrt{\Lambda_0(l)} z_0))}{\int Dz_{\text{com}} g_{\text{teacher}}(\sqrt{\Lambda_{\text{com}}(l)} z_{\text{com}})} ,
$$
(146)

with

$$
g_{\text{teacher}}(h) \equiv \int Dz_{\text{teacher}} e^{-\beta V(h - \sqrt{\Lambda_{\text{teacher}}} z_{\text{teacher}})} .
$$
(147)

For the hardcore potential Eq. (7), $g_{\text{teacher}}(h) = \Theta(h/\sqrt{2\Lambda_{\text{teacher}}})$ with $\Theta(h)$ defined in Eq. (111) and $g'_{\text{teacher}}(h) = e^{-h^2/2\Lambda_{\text{teacher}}}/\sqrt{2\pi\Lambda_{\text{teacher}}}$.
Using the above results we find, for $l = 1, 2, \ldots, L-1$,

$$
\epsilon_r(l) = -r(l-1) R(l) \kappa_{\text{inter}}(l) - R(l+1) r(l+1) \kappa_{\text{inter}}(l+1) .
$$
(148)

### C.3.4 Variation of $R$

Finally we consider variation of the free-energy Eq. (19) (see also Eq. (73)) with respect to $R(l)$, for $l = 1, 2, \ldots, L$,

$$
\begin{aligned}
0 &= \frac{\partial}{\partial R(l)} \, \partial_s s_{1+s}[\{\hat{Q}(l), \hat{q}(l)\}]\Big|_{s=0} \\
&= \frac{\partial}{\partial R(l)} \frac{1}{\alpha} \, \partial_s S_{\text{ent,bond}}[\hat{Q}^{1+s}(l)]\Big|_{s=0} - \frac{\partial}{\partial R(l)} \mathcal{F}_{\text{int}}[\{\hat{Q}^{1+s}(l), \hat{q}^{1+s}(l)\}]\Big|_{s=0} .
\end{aligned}
$$
(149)

Variation of the entropic part (bond) of the free-energy Eq. (125) yields,

$$\frac{\partial}{\partial R}\, \partial_s S_{\text{ent,bond}}[\hat{Q}^{1+s}]\Big|_{s=0} = -\frac{R}{G_0}. \tag{150}$$

On the other hand, variation of the interaction part of the free-energy Eq. (135) yields,

$$
\begin{aligned}
&-\frac{\partial}{\partial R(l)}\, \partial_s \mathcal{F}_{\text{int}}[\hat{q}^{1+s}(l-1), \hat{Q}^{1+s}(l), \hat{q}^{1+s}(l)]\Big|_{s=0}\\
&= -\left( \frac{\partial \Lambda_{\text{com}}(l)}{\partial R(l)}\frac{\partial}{\partial \Lambda_{\text{com}}(l)} + \frac{\partial \Lambda_{\text{teacher}}(l)}{\partial R(l)}\frac{\partial}{\partial \Lambda_{\text{teacher}}(l)} + \frac{\partial \Lambda_0(l)}{\partial R(l)}\frac{\partial}{\partial \Lambda_0(l)} \right)\\
&\qquad\qquad \partial_s \mathcal{F}_{\text{int}}[\hat{q}^{1+s}(l-1), \hat{Q}^{1+s}(l), \hat{q}^{1+s}(l)]\Big|_{s=0}\\
&= r(l-1)r(l)\kappa_{\text{inter}}(l), \tag{151}
\end{aligned}
$$

with $\kappa_{\text{inter}}(l)$ defined in Eq. (146). Using these results in Eq. (149) we find,

$$R(l) = \alpha G_0 r(l-1)r(l)\kappa_{\text{inter}}(l). \tag{152}$$

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
