# Peer review of "From complex to simple : hierarchical free-energy landscape renormalized in deep neural networks"

_SciPost Physics, doi:SciPost Phys. Core 2, 005 (2020)_

## Round 1 · Referee Report · Anonymous (Referee 1) · 2019-12-2

Strengths

  1. The theoretical computation is remarkable and, as far as I can tell, novel
  2. Potentially very interesting and useful results
  3. Well organized and written

Weaknesses

  1. The approximation used to derive the results might prove too crude (it's hard to tell). I have some doubts in particular about the teacher-student case.
  2. Some numerical experiments (especially in the teacher-student scenario) are rather far from the theoretical analysis

Report

The author leverages an approximation that allows him to perform an in-depth replica analysis, at arbitrary level of RSB, of a neural network with an arbitrary number of layers of constant size and with sign activation functions. This is technically quite remarkable. The results are also very interesting, and seem to capture some important features of the network, since they seem to roughly agree qualitatively with the results of Monte Carlo numerical experiments.

My main doubt about the claims of the paper arises in the teacher-student scenario. In this case, the theory seems to produce a paradoxical situation in which the student "loses track" of the teacher in the middle of the network but then recovers it toward the end. The author proposes (p.22/23) a solution based on a vanishing symmetry breaking field. This is possible, and would be quite interesting if true. However, one may well suspect that this is the result of the Gaussian approximation used, which in turn forces a bi-directional symmetry that is not present in the original system. Personally, I find this latter scenario more likely, unless persuaded otherwise, simply because it seems to be the most straightforward explanation to me. The following discussion on the generalization performance reinforces in me this opinion, since (if I understand it correctly) it seems to imply perfect or near-perfect generalization of a random function (the teacher) even in the over-parametrized regime. In turn, that would imply that the inductive bias imposed by the architecture is extremely strong, meaning that any such architecture would tend to realize, with overwhelming probability, a tiny subset of all the possible mappings. I don't think that's the case. So am I missing something?

The numerical experiments tend to agree with the theory in a qualitative way, but they are performed on very small systems, and furthermore with binary weights (p.26, sec. 6.2). The binary weights case can potentially be quite different from the continuous case (and much harder). So this makes me wonder about the reason for this choice. (In the random scenario the weights were continuous.) Also, the networks used in the experiments are really small, with $L=6$ layers of $N=4$ units at most (by the way the text says "$N \times L$ values of the bonds", shouldn't it be "$N^2 \times L$" as in eq. (3), i.e. 96 at the most? Minus 20% that's 77 weights). I suppose this is due to the difficulty of training the student with MC. Still, I think that this may be problematic for the comparison with the theory, as finite-size effects could be very strong.

On top of this the teacher-student scenario has the problem of the permutation symmetry in the hidden units when measuring the overlaps. The text (p.27) reads "In order to remove this unwanted symmetry, we removed 20% of the bonds randomly in each layer, but exactly in the same way, from both the teacher and student machines." I don't understand this procedure, unless the bonds were removed even before the training phase. If so, it should be clarified. Also, this is another (probably minor?) difference with the theoretical analysis, and it might further exacerbate finite-size effects. Furthermore, on such small networks, removing at random 20% of the weights might not be enough to disambiguate all the units: it means removing less than 1 link per unit on average; from the point of view of two units in a layer, if their input and output links are masked in the same way they have the same local structure and they are thus interchangeable. Indeed, a very quick test shows that in a random mask on a network with $N=4$, $L=6$ there is about a 65% chance that at least one such ambiguity arises in one of the layers 2 to 5 (with $L=4$ the chance reduces to about 41%). This could contribute to reduce the apparent overlap considerably on average (each ambiguous pair that the student learns in the wrong order reduces the overlap of a layer by half with $N=4$, so very roughly it's a reduction by a factor of 0.75*0.65+0.35=0.84 on average).
[To me, the most natural (albeit definitely more expensive) approach without masks would be to find the permutation of the indices of the hidden units that maximizes the overlap with the teacher, starting from the first layer and moving up one layer at a time. This is still sub-optimal but computationally feasible (it amounts at solving a small bipartite matching problem at one layer; applying the permutation to that layer and the next; moving up and repeating).]

I think that the author should comment on this issue. But just to clarify, in my opinion even without any teacher-student analysis at all the results on the random patterns case would still be extremely valuable and interesting, so I don't consider this as a major problem for publication.

Apart from this, there are still a few unclear points for me. One is the definition of "over-parametrized regime", which is said (p.6) to be when $L > \alpha$. Yet, many experiments (e.g. fig. 7) use $L=20$ and $\alpha = 4000$, and the results suggest that the network is still "liquid" in the middle, thus that it is not at capacity. Either I'm not understanding what is meant with "over-parametrized regime" (and perhaps this is related to the teacher-student results discussed above), or this is an effect of the approximation. Relatedly: Is there a way to compute the critical capacity for the network in this framework? And how would it scale, and how would the order parameters look like? Etc. Or is there a reason not to compute it, technical or otherwise? I wish that the author clarified these points.

My only remaining issues are very minor points or typos, and I'll leave them for the "Requested changes" section.

Requested changes

Apart from the points mentioned in the main report, here follows a list of minor issues:

  1. (p.4): "using a Gaussian ansatz" -> I don't think it can be called an ansatz, I suggest to use "approximation"

  2. (p.6): the notation for eq. 5 is a little ambiguous and confusing. As written, it looks like the expression is factorized on the nodes. Probably, the easiest fix would be to add parentheses that clarify that the first two products are meant to represent a multidimensional integral. Alternatively, a more common notation I've seen for this is to bring the products inside the integral (maybe with parentheses to group the integration variable and the spherical constraint).

  3. (p.8): "In Eq. (11) it is assumed that all replicas follow the same labels breaking the permutation symmetry. Second, the system is symmetric under permutations of perceptrons [ ] within the same layer and the permutations could be done differently on different replicas. In Eq. (11) this permutation symmetry is also broken." -> I think that it should be clarified how this second breaking (the one on the units) is achieved, as it is not clear at all.

  4. (p.19, sec. 4.2): The second argument for the input overlap fluctuation structure reads "It is natural to consider the case that input data fluctuates during learning as it happens in the standard Stochastic Gradient Descent (SGD) algorithm." However, in SGD the fluctuations happen as a byproduct of the learning algorithm, whereas the optimization goal is still in principle to learn the whole, non-fluctuating training set. To me, what the author suggests seems more similar to adding noise in the inputs, or to some stochastic data augmentation techniques perhaps. Even then, assuming an ultrametric structure does not seem obvious or particularly "natural" for those scenarios to me. Unrelatedly, in the third argument, "Real data may not be totally random" is quite an understatement!

  5. (p.21, and analogously for the beginning of p.29 where the argument is repeated): "We can regard this as a kind of ’renormalization’ of the input data [...] This means that a DNN works naturally as a machine for renormalization, i.e. classifications and feature detections." I think I understand what the author is saying here, and it is certainly a very interesting observation. However, I think that the link between the "renomarlization" operation as shown by the results of the analysis and its interpretation in the context of classification and feature detection is not as straightforward as is being put here. In my opinion, this sentence would require either a slight reformulation or to be expanded with additional discussions and arguments.

  6. (p.24-26, sec. 6.1): Maybe I missed it but do the MC simulations reach zero error? A plot of both the loss and the error as a function of time would be useful.

  7. (p.45, on the numerical solution of the k-RSB equations). There are several points where the procedure requires to compute functions involving a variable h (e.g. f, f', P..), which is then integrated over. Was this done by sampling h (and m) at regular points on a grid? If so, at what interval? If not, how? Please expand a little on the numerical procedure.

  8. Some figures (e.g. figs. 15, 16) are very hard to read even when zoomed, due to poor resolution. The author seems to be using gnuplot; I suggest using a vector graphics terminal such as svg (it can then be converted easily to pdf) or eps rather than a raster terminal such as pngcairo.

  9. Typos:

  10. throughout the paper: "i. e." -> "i.e." p.4: "it can happen that solution space" -> missing "the" p.4: "to understand the efficiently" -> "...the efficiency" p.4: "a statistical inference problems" -> "...problem" p.4: "creases" -> "increases" p.5: "perceptrons A perceptron" -> missing full stop before "A" p.6: "where we introduced 'gap'" -> "...the 'gap'" p.7: "From physicist’s point of view" -> "...a physicist's..." p.7: "As a complimentary approach" -> "...complementary..." p.7: "by construction its own synaptic weights" -> something went wrong with this sentence, maybe "by adjusting its own synaptic weights" etc.? p.11: "a delta functions" -> "a delta function" p.15: "As the result it looks approximately like" -> "...as a result..." p.15: "this amount to induce" -> "...amounts..." p.16: "the 2nd glass transition induce" -> "...induces" p.16: "an internal step like structure emerge continuously" -> "...emerges..." p.16; "the emergence of the internal step amount" -> "...amounts" p.16: "have interesting implications" -> "has..." p.16: "the number of constraints increase, the allowed phase space become suppressed" -> "...increases...becomes..." p.16: "the probability appear to decay" -> "...appears..." p.17: "which leave behind river terrace like structure" -> "...behind a river-terrace-like structure..." p.17: "finite glass order parameter emerge continuously" -> "a finite glass order parameter emerges..." (or "...parameters...") p.17: "the layers included in the glass phase is" -> "...are" p.17: "and then it implies" -> "and this implies" p.17: "$1 - q(x,l)$ ,i. e." -> swapped comma/space p.18: "To understand meaning" -> "...the meaning" p.18: "The hierarchical organization of clusters imply" -> "...implies" p.18: "small valleys are group" -> "...grouped" p.18: "it progressively become" -> "...becomes" p.18: "in deep enough network" -> "...networks" (or "in a deep enough...") p.19: "suggests that basic hierarchical structure" -> "...that the basic..." p.19: "spin configuration on the boundaries are allowed" -> "...configurations..." p.19: "exhibit hierarchical, structure" -> "exhibits a hierarchical structure" p.20: "the resultant glass order parameter" -> "...resulting..." p.20: "the hierarchical structure put in the input propagate" -> "...propagates" p.21: "Numerical solution suggests" -> "The numerical..." p.21: "are progressively renormalized into $q = 0$ sector" -> "...into the $q = 0$ sector" p.22: "which read" -> "which reads" p.22: "grow from the boundary" -> "grows..." p.23: "Most likely scenario is" -> "The most..." p.23: "which do not contribute the order" -> "...does not contribute to..." p.23: "remain in the central part" -> "remains..." p.23: "naturally arise by" -> "...arises..." p.24: "Each component of the spins only take...each of the bonds take" -> "...takes...takes" p.25: "in the low a) ... in the low b) ... in the low c)" -> "in row a)..." etc. p.27: "we used systems with $L = 4$" -> it seems from fig. 16 that $L = 6$ was also used, and that $\alpha=4$ was only tested there. p.28: "We argued that small the positive overlap can remain" -> "...that the small positive..." p.28: "the opposite side of the system" -> "...sides..." p.28: "which appear to be compatible" -> "...appears..." p.28: "sclae" -> "scale" p.28: "Weak bias which remain" -> "...remains" p.30, caption of fig. 15: "In the 1st low" -> "...row"; same for 2nd and 3rd p.32, fig. 17, panel a): The points are centered at half-integer values? p.36 eq. 51: I suppose that $c$ in the first equation (and in eq. 52 and in the sentence after eq. 53) should be $N$? Also, shouldn't there be an (irrelvant) $1/2\pi$ factor? p.38 eq. 66: There's an extra power of 2 in the 3rd order term, it should be $\langle A^2 \rangle \langle A\rangle$ not $\langle A^2 \rangle^2 \langle A\rangle$ p.38 eq. 67 (also eq. 68): I think that a property is used here which is a generalization of eq. 59 to the case of different nodes. What about writing the general form in eq. 59, with an extra delta? p.45, point 2, step (7): should it be "compute $G_i(l)$ ... using Eq. (112)"? p.45, point 3, step (6): should "using eq. (18)" be "using eq. (87)"? p.47: "One just keep in mind" -> I guess "...must..."?

  • validity: high
  • significance: high
  • originality: top
  • clarity: top
  • formatting: good
  • grammar: good

Author:  Hajime Yoshino  on 2019-12-04  [id 664]

(in reply to Report 1 on 2019-12-02)
Category:
remark

I would like to thank the referee (report 1) for the very careful reading of the manuscript and many valuable comments.

The following is a short note which hopefully helps to clarify some concerns raised by the referee. I will include the following points in the revised manuscript.

Once we turn to the statistical mechanics of the 'solution space a la Gardner [5,6] for the perceptrons, there is no 'feed-forward'ness. One can look at the partition function (5) just as a partition function of a spin model defined on a certain asymmetric network with two boundaries. The constraints put on the boundaries, either on the input or output side, restrict the solution space. For example, one can ask what is the consequence in the solution space by a perturbation put on the output layer (back-propagation). It is not so surprising that phase transitions take place starting around the two boundaries where the effect of constraints are stronger.

But when we try to interpret the results obtained in such statistical mechanical analysis on the solution space, especially phase transitions, we may need some care. A 'phase' characterized by a non-zero order parameter breaks the corresponding symmetry of the original system. In the case of constraint satisfaction problems or statistical inference problems, one should ask what plays the role of symmetry breaking field in the algorithms used and initial conditions used for them which allow the algorithm to pick up the particular state. (In the case of DNN, the system is a feed-forward network and learning usually involves backward propagation.) This is particularly an interesting problem in the case of DNN if it has a 'liquid' phase in the center as our theory suggests.

The above points hold irrespective of what kind of approximation (or no approximation) one makes to evaluate the partition function (5). In our case, we made the Gaussian approximation which fails to capture the geometric asymmetry of the network. Better approximation should predict asymmetry reflected in the solution space. I agree with the referee that this is a serious drawback of our approximation, which we have stated it in the conclusion. Generalization-ability is certainly overestimated. I will add a warning on this point in the revised version.

I will respond to the specific comments of the referee later.

I attach below a preliminary data of additional simulation of the teacher-student setting (corresponding to Fig. 16) but removing 40% of the bonds instead of 20% (from the beginning, before training). Qualitatively speaking the result is not significantly different from the case of 20%. I will include the data in the revised manuscript.

Attachment:

fig_ising_perceptron_teacher_student.pdf

Anonymous on 2019-12-17  [id 687]

(in reply to Hajime Yoshino on 2019-12-04 [id 664])
Category:
correction

I checked higher-order terms in the cumulant expansion reported in appendix A.2.1 and found that they become actually negligible in N >> 1 limit (see the note attached below). So our result is not a Gaussian approximation. (However, we do not have a proof that we have exhausted all sets of relevant order parameters such as some crystalline order parameters. ) I will make corrections regarding these in the revised manuscript.

The theory gives the input/output symmetric results. Probably the geometric asymmetry we anticipate and see in numerical simulations disappears in large width N >> 1 limit. ( This is similar to the following: if we cut out a piece of a square of linear dimension L in an arbitrary way from an infinitely large triangular lattice, the edges of the square are not nicely matched with the triangular lattice. However, the effect will become negligible in large size limit L >>1. )

Attachment:

note1217.pdf

---

## Round 1 · Referee Report · Anonymous (Referee 2) · 2019-12-17

Strengths

1- The paper addresses a notoriously difficult problem 2- It proposes a new 'gaussian approximation' that allows analytic progress.

Weaknesses

1- The choice of the regime $M/N$ finite is not the most relevant one both for the storage problem and for generalization. 2- The paper concentrates on technical details much more than on physics. 3- The meaning, qualities and limitations of the main approximation are not discussed.

Report

This paper propose an approximate analytical treatment to study the Gardner volume of
multilayer neural networks. This is a notoriously difficult problem and any advances are welcome. The author considers networks of $L$ layers of $N$ formal neurons each, in the limit $N\to\infty$. The addressed problems are (1) the storage problem of random associations and (2) the generalization problem in the student-teacher setting. The author consider a number of examples $M$ given to the net which scales as $N$, namely $M=\alpha N$. This is far from the capacity in problem (1) and from the onset og generalization in problem (2) which both should scale as $M_c\sim N^2 L$.
The author introduces a 'gaussian approximation' that allows him an approximate analytic treatment through the replica method. Unfortunately, as a result of the approximation the networks looses its feed-forward directionality and becomes symmetric under exchange of input and output. For small $\alpha$ the authors find that the system is in a 'liquid phase'. Increasing $\alpha$ the author finds that freezing
of couplings and internal representations propagates from the boundary towards the interior of the network, with a characteristic 'penetration length' scaling as $\log \alpha$. A very detailed description of the various transitions is proposed.
The propagation of freezing from the boundary towards the interior is confirmed by Montecarlo numerical simulations. Quite surprisingly the author claims that some form of generalization is possible in problem (2) for finite $\alpha$. It is not clear to me if this is just an artefact of the gaussian approximation.

Though I am convinced that the results of the paper are potentially worth to be published, I found the paper very difficult to read, it concentrating on the
explanation of the details of the replica techniques used in the paper, and much less on the physical motivations of the choice of the models and regimes under study,
the implications of the various solutions for learning and so on. Above all, meaning, qualities and limitations of the gaussian approximation, that potentially is the main contribution of the paper are barely discussed at all.

I sincerely think that the results are potentially interesting, but the paper in the present form does not render justice to them. I would suggest the author to rewrite completely the paper concentrating much more on physically salient points rather then on replica details.

Requested changes

My suggestion is to rewrite completely the paper, with emphasis on physics rather than on technicalities.

  • validity: high
  • significance: high
  • originality: high
  • clarity: poor
  • formatting: below threshold
  • grammar: below threshold

Author:  Hajime Yoshino  on 2019-12-23  [id 692]

(in reply to Report 2 on 2019-12-17)
Category:
remark

I thank the referee (report 2) for useful comments and suggestions.

I agree with the referee to revise the main text so that it becomes less technical. I will send some of the technical parts to appendices and add more discussions in the main text.

I will add a remark that the 'liquid phase' is interesting also in the context of biology since it is related to the idea of 'neutral space'.

Concerning the 'Gaussian approximation' please see my previous comment on '2019-12-17' which can be found below in this web page. I reported there that higher-order terms of the cumulant expansion are actually absent so that the theory is not an approximation but exact in this respect.

Concerning 'Directionality', it becomes irrelevant once we consider the statistical mechanics of solution space irrespective of approximations. This is a completely separate issue as pointed out in an earlier comment on ' 2019-12-02' which can also be found on this webpage.

The above two points will be included in the revised manuscript.

The scaling $\alpha=M/N$ is well known for a single perceptron (Gardner 1987). Essentially I find this continues to be the key parameter (like inverse temperature for condensed matters) for bigger networks. In the present DNN system, the data size scales as $MN=N^{2} \alpha$ while the number of parameters scales as $N^{2}L$. On the other hand, it is interesting to note that the "expressivity" of a DDNs is known to grow exponentially with the depth $L$ due to the convolution of non-linear functions. ( see, for instance, Poole, Ben, et al. "Exponential expressivity in deep neural networks through transient chaos." Advances in neural information processing systems. 2016. arXiv:1606.05340). This implies for a given expressivity $\alpha'$, one needs only a finite depth $\xi' \propto \ln \alpha'$ system. This appears to be related with our finding that the depth of the 'solid-phase' grows as $\xi \propto \ln \alpha$. The capacity limit of DNN, which requires $\xi \gg L$, can also be studied by the present theory easily but I will leave it for another work since it is not of the main interest in the present work.

---

## Round 2 · Referee Report · Anonymous (Referee 1) · 2020-2-26

Strengths

  1. The theoretical computation is remarkable and, as far as I can tell, novel
  2. Potentially very interesting and useful results
  3. Well organized and written

Weaknesses

  1. I believe that the results are approximate (as was actually claimed in the previous version) rather than exact/quasi-annealed (as claimed in this version). Given that, I'll report my previous comment: The approximation used to derive the results might prove too crude (it's hard to tell). I have some strong doubts in particular about the teacher-student case.
  2. The setting of some of the numerical experiments (in the teacher-student scenario) are rather far from the theoretical analysis. Additionally, when the setting is pushed a little more in the direction of the theory (increasing N), the results go in an opposite direction than the theory suggests (generalization capability drops).

Report

The author has amended and expanded the paper and in doing so he has fixed most of the issues that I had raised in my previous review.

The new version however also includes now some stronger claims with respect to the previous one, as a result of a new computation (eq. 71) which shows that some terms of order higher than 2 become negligible in the bulk of the network. Based on this result, the author has removed the previous description of the computation as a "Gaussian approximation", claiming instead that the result is an "exact replica calculation" except for the boundaries where the calculation is not quenched but "slightly annealed". Since some features of the results would be surprising (mainly, the symmetry of the solution to an asymmetric model), the author presents some qualitative arguments and some analogies with other systems (e.g. hard spheres) in support to this claim.

I'm not convinced that this claim is correct. In brief, the procedure detailed in section A.1.4 called "Plefka expansion" is only exact in the limit of weak interactions (small $\lambda$), otherwise eq. 53 does not hold. But in that section it is written: "The parameter $\lambda$, which is introduced to organize a perturbation theory, is put back to $\lambda = 1$ in the end." This is a weak-interactions approximation. It seems to me that this procedure is crucial to derive the results of the paper, in particular eq. 62, which allows to derive eqs. 63-64 which in turn allow to perform the necessary simplifications in eq. 71, which is the basis of the new, stronger claim. Indeed, the author writes that eq. 62 corresponds to eq. 52, which is valid in absence of interactions.

(I should say at this point that I have some additional comments about the clarity of the derivation in the appendix; it's possible that I misunderstood something, see below.)

From the qualitative point of view, the author argues that the surprising symmetry of the result stems from the infinite $N$ limit, but I fail to see how that would happen here. For example, one argument is that "The system is locally asymmetric (at the scale of each layer) but globally homogeneous except at the boundaries." but I don't see this supposed homogeneity; the expression is very clearly still asymmetric. I don't dispute that it may happen that an asymmetry like this might become irrelevant in some limit for some models, I'm just not convinced that this is the case here, at least not exactly. The "entropic argument" given at p.14 is not unreasonable, but it's qualitative and an a-posteriori explanation; certainly it's not sufficient by itself to justify exact symmetry. (On the other hand, I still think that the results are relevant and probably qualitatively valid, and if so that the entropic argument gives a plausible mechanism).

The generalization result for the teacher-student scenario is still the most problematic in my opinion, as I still have a very hard time believing that near-perfect generalization is possible with a liquid phase in the middle of the network, just from "combinatorics-like" arguments (different networks should correspond to different $N$-bits-to-$N$-bits mappings, once permutation symmetry is accounted for). I had previously argued that, if one seeks an explicitly symmetric solution, imposing an overlap 1 at the initial end would obviously result in an overlap 1 at the other end. Now the author argues, from eq. 71, that the symmetry in the solution emerges spontaneously in the large $N$ limit rather than being enforced a priori from an approximation. Since as I wrote above I'm not convinced that that's the case, I maintain my previous opinion. One way to check if I'm right could be to test the equations at smaller values of $\alpha$: it the output overlap is always 1, implying perfect generalization for any size of the training set, which would be clearly absurd (one expects to see something like the single teacher-student perceptron with continuous weights, where the error decreases with $\alpha$), then the result is an artifact of the approximation.

Relatedly, the results of the simulations in the teacher-student case use extremely small values of $N$ and are thoroughly unconvincing. I have reimplemented the procedure described in the paper, and with the same parameters ($N=4$, $L=6$, $\alpha=1,2,4$), I obtain precisely the same results for the generalization overlap that are reported in fig. 17, panel c, right plot. However, increasing $N$ makes all the overlaps drop drastically. As an example of parameter settings that I could test quickly, for $L=10$ and $\alpha=1$, I obtain the following values for $N=4,8,12$: $0.65$, $0.21$, $0.09$. I see similar results with $L=20$. I strongly suspect that the limit is $0$ for large $N$. In fig. 19, where some analysis on the dependence of the results with $N$ is shown, the overlap for the output layer (the one which actually correlates with generalization) is not shown in panel c, as far as I can tell. I wonder why. In any case, the values of $N$ used are $3,4,5$ which makes it really hard to observe any effect.

If I'm right, I think that the whole teacher-student part should be either heavily revisited or left out entirely for a more in-depth study.

I have some additional issues that I'll leave for the "Requested changes" section.

Requested changes

The big ones, if I'm right, are to:

  1. Revert the description of the results as approximate;

  2. Revisit entirely the teacher-student generalization issue and corresponding simulations (possibly, removing the whole section and leaving it for a follow-up more in-depth study, ideally accompanied by convincing numerical evidence).

Apart from that, the following notes are mostly about typos or very minor issues, but there are a few more serious issues too.

  1. p.10 eq. 12: I think there has been a mix-up here, the $\eta$ integral doesn't belong here, it's introduced by the Fourier transform in eq. 38

  2. p.12: "Maybe one would not easily expect such asymmetry in the feed-forward network" I think the author meant "symmetry" here.

  3. p.27 eq. 34: The update move was chosen in such a way as to ensure that on average the spherical constraint on the weights is satisfied. However, due to the random-walk nature of the update, the averages may "diffuse" over time. This can have an effect on the autocorrelations, eq. 35, especially since $N=20$ is being used which is fairly small and might not allow to neglect effects of order $\sqrt N$. It might be necessary when computing the autocorrelations to rescale by a factor $\left\Vert J(t)\right\Vert$ (I'm assuming $J(0)$ is normalized, otherwise it should be too). A quick check should be sufficient to detect whether this is an issue or not.

  4. p.28: "Note that the behavior of the system is not symmetric to the exchange of input and output sides. We expect that this is a finite $N$ effect which disappears in $N \to \infty$ limit." I don't expect that, see comments in Report. Providing evidence with varying $N$ might be a good idea here.

  5. p.29: It seems like the same $M$ was used for training and testing, but there is no reason for limiting the amounts of testing patterns: the larger the $M$ used for testing, the more accurate is the measurement.

  6. p.39-43, figs. 15-19: The figures are very far from the portion of the text where the simulation results are presented and discussed, making it rather hard to follow. The caption of fig. 17 at p. 41 goes beyond the end of the page and it's partially unreadable (on the other hand reducing the figure size would make it hardly readable because of poor resolution, so the author should consider finding a way to improve the figure resolution).

  7. p.44 eq.38: There should be an $a$ index in the products of the $J$ and $S$ traces in the second line. Also is $\xi_{\mu \nu}$ a Kronecker delta symbol? Otherwise I can't make sense of this expression and the sudden appearance of another pattern index $\nu$. (And even if my interpretation is correct, the purpose of doing this is very unclear anyway.)

  8. p.44 (below eq. 39): "free-energy" repeated twice.

  9. p.44: "excplitely" -> "explicitly"

  10. p.46 eq.53: [Mentioned above] This equation is only valid for small $\lambda$. If then $\lambda$ is "put back to 1", the result is an approximation.

  11. p.47 eq. 55: The $\epsilon$ and $\varepsilon$ terms should multiply the second term inside the exponent too (the sums)

  12. p.47: "assuming $c \gg 1$" this is a leftover from the previous version, it should be $N$

  13. p.47 eq. 59: There are some superscripts $\mu$ which don't make sense there, I guess they should be $i$ (and to follow previous formulas they should go inside parentheses, with the replica indices outside).

  14. p.47 eq. 61: The replica and pattern indices are swapped compared to the previous formulas.

  15. p.47: The expressions in eqs. 56-61 are not clear, one has to guess a few things in order to follow. Mainly, that the dots on the l.h.s. of 56,57 seem to refer to some expression that can be factorized over the indices $i$ or $\mu$ (depending on the formula) and that the corresponding dots on the right hand sides are the factorized expressions (which however still depend on the index $i$/$\mu$?). Also since the $\blacksquare$ index has disappeared it seems like the assumption is that the dotted expressions do not depend on those. Moreover the index $i$ was dropped from the expressions in eqs. 58-59, but there is a spherical constraint on the $J$s (eq. 65) so that the trace over $J^c$ is actually not well defined here and the reader has to guess what's actually going on. At least defining the trace operators like in ref. [20] might help a bit. Additionally, the expression in the last line of 38 is not actually factorized over the $\blacksquare$, of course, so clarifying the assumptions used here (and how they relate to the $0$-th order expansion of the previous section) is quite important, I think. That expression becomes "tractable" after the expansion of eq. 71, but then only because they use the simplifications allowed by eqs. 63-64, which were derived in the interaction-free assumption...

  16. p.47 eq. 62: "where $\epsilon^*_{ab}$= [...] are determined by": again, it should be clarified here that these expressions come from the non-interacting assumption. In terms of the previous expression, it is like neglecting the dotted expressions, right? Otherwise the saddle points on the eplisons would involve those as well. This is basically said in the following paragraph when it's pointed out that they correspond to eq. 52, but I think that making it more explicit before the expressions would clarify the derivation considerably.

  17. p.48 eq. 65: $c$ -> $N$

  18. p.48 eq. 67: The $S$ should be lowercase.

  19. p.48 in the unnumbered equation between 68 and 69: The $c$ should be a superscript of $S$ instead of a subscript. Also $\epsilon_{aa}$ was never defined before, and it uses the wrong character ($\epsilon$ instead of $\varepsilon$); probably just writing that it is assumed to be $0$ beforehand would be enough.

  20. p.49: This section should probably refer to eq. 38 (the replicated one) rather then eq. 12

  21. p.50 eq.72: There's a factor $2$ missing in front of the $\cosh$ resulting in a missing constant term in the following two lines; it's irrelevant, but still...

  22. p.50: [This one is more important than the others] The difficulty arising when the $J$ are considered quenched is mentioned. Isn't this precisely the teacher-student situation (at least for what concerns the teacher)?

  23. p.51 and following: In all expressions with $S_{ent,...}"$ the $S$ should be lowercase for consistency with the previous sections.

  24. p.57 eq.115: a closing parenthesis is missing

  25. p.57 eq.123: the prime sign after the parenthesis in the numerator is a peculiar notation that should be defined.

  • validity: ok
  • significance: high
  • originality: top
  • clarity: high
  • formatting: good
  • grammar: good

Author:  Hajime Yoshino  on 2020-02-28  [id 748]

(in reply to Report 1 on 2020-02-26)
Category:
remark

I thank the referee for carefully reading the revised manuscript and providing useful comments. The following is a short note which hopefully helps to clarify some concerns raised by the referee. I will include the following in the 2nd revised manuscript.

In the appendix sec A.1.4 "Plefka expansion", I forgot to mention that higher-order terms $O(\lambda^{2})$ in the expansion eq (5) vanish in the present model. This is simply because the higher-order terms in the cumulant expansion reported in sec A.3 vanish (provided that we employ the annealed boundary condition discussed in sec A.3.2). Thus the expansion terminates exactly at order $\lambda$. The situation is similar to what happens in mean-field spinglass models (see for instance [77]). Indeed one can follow exactly the same steps to obtain the exact free-energy functional of the 'disorder-free spin models' [20], hard-spheres in large-d [11,29] and the family of p-spin (Ising/spherical) mean-field spinglass models (with or without quenched disorder) (see Chap 6 of [20]).

Concerning the simulation of teacher-student I agree with the referee that increasing N generalization ability decreases. Please find the pdf file attached below for the extended version of Fig. 19 c) which include $l=4$ layer. Indeed the correlation decreases also in the output. I agree with the referee that this should be shown and I will include this as a new panel d) of Fig 19 in the 2nd revised version. The purpose of the panel b) and c) is to show the small values of correlation in the bulk part so that the y-range is limited to [0:0.2]) . As $N$ increases, the 'bias field' (remaining signal in the center) becomes smaller so that it is not surprising that the simple gradient descent (greedy Monte Carlo) cannot recover the teacher machine well. Increasing $L$ the signal will certainly become weaker. How the symmetry breaking field works in various real algorithm is a very interesting question by itself. Probably more stochastic versions of the algorithm will find the teacher machine better. Anyway the increase of the correlation in the end despite the liquid like region in the center is remarkable. I hope the numerical results motivate further works in the future.

I will respond to the specific comments of the referee later.

Attachment:

plot_finite_N_spin_test_all.pdf

Author:  Hajime Yoshino  on 2020-03-01  [id 750]

(in reply to Hajime Yoshino on 2020-02-28 [id 748])
Category:
remark
reply to objection

I deeply thank the referee for the referee's efforts and the useful comments.

I completely agree that my sentence "Probably more stochastic versions of the algorithm will find the teacher machine better" is based on nothing. Let me delete it.

I also understand the concerns of the referee on the theoretical analysis of generalization presented in sec 3.5.2. In my view, the problem is that the 'fictitious' symmetry breaking field used in the theory (see sec. A.1.3) has no real counterpart in the 'test stage'. In the learning stage, I believe that the small bias field plays the role of the symmetry-breaking field. Importantly it is polarized toward the learning data but NOT toward test data. I will revise sec 3.5.2 substantially.

Indeed we can think of a simple thought experiment as follows: create a 'random' teacher machine and a student machine which is just a copy of the teacher machine. Then we completely reshuffle the bonds of the student machine in some layers in the center. So we are trying to mimic a situation that the student machine perfectly learns the teacher machine except for the center which is in the liquid phase. Then we can compare the spin configurations of the two machines subjected to some test inputs. The overlap of the two machines decreases significantly in the randomized layers. In the large $N$ limit this overlap should drop to $0$ so that it is non-zero only in finite $N$ systems. Approaching the output, the overlap increases amplifying the bias. But anyway the overlap in the last layer decreases with $N$.

Please find the pdf file attached below where I display a plot of a result of such a simple experiment (Binary perceptron $L=10$, $N=3,4,5$, with bonds on the two layers $l=5,6$ in the center are completely reshuffled.). I will put this in the revised manuscript.

Concerning some other important points raised in the referee's report:

1) [Report], 5th paragraph, "..but I don't see this supposed homogeneity; the expression is very clearly still asymmetric."

The replicated free-energy Eq. (76) turned out to be formally homogeneous. Here even the "local asymmetry" at the scale of each layer that we naturally expect is lost, which looks bizarre. This comes from the annealed boundary condition. I expect in 'strict' quenched boundary condition (sec.A.3.2), which I do not know how to treat properly, the input and output boundaries look different and this should also make differences in the bulk part to some extent. I will mention this in the revised text.

2) [Requested changes] 24. p.50: [This one is more important than the others] The difficulty arising when the J are considered quenched is mentioned. Isn't this precisely the teacher-student situation (at least for what concerns the teacher)?

I also understand this concern. Here I have to assume that both the teacher and student machines are subjected to the 'annealed boundary condition'. I assume the teacher machine is not quenched but evolving on very long time scales so that it looks quenched for the student machine. In the replica analysis, this is realized by taking $s\to 0$ limit using the $n=1+s$ replicas with the $s$ replicas for the students. This is the standard trick to obtain the Franz-Parisi potential (state following).

Attachment:

plot_simple_experiment.pdf

Anonymous on 2020-02-28  [id 749]

(in reply to Hajime Yoshino on 2020-02-28 [id 748])
Category:
remark
objection
suggestion for further work

Referee here. I'll only comment on the teacher-student part at this stage.

I find this sentence highly problematic: "Probably more stochastic versions of the algorithm will find the teacher machine better." There is no reason to think that, when the simulations point in precisely the opposite direction. The scenario in which that may happen is as follows: the teacher-student training problem is solved by an exponentially large number of configurations for the students, the overwhelming majority of which would behave like reparametrized versions of the teacher and generalize perfectly, while a vanishing minority would implement different transfer functions that only agree with the teacher on the training set. Yet, starting from a random configuration and seeking the closest solution to the training problem would systematically find the second type of students. This scenario is contrary to experience and intuition and I see no mechanism how it could be realized; thus I believe it's untenable without very good evidence. By the way, as a check I have added annealing in my code and while it slightly improves the solving capabilities of the algorithm nothing really changes as far as the generalization capabilities are concerned, no matter how fast or slow the annealing is performed.

About this: "the increase of the correlation in the end despite the liquid like region in the center is remarkable." On this I tend to agree (although I think it'll vanish in the large $N$ limit).

As for the theoretical side, I'll further summarize what I wrote in the review: if the generalization performance doesn't depend on $\alpha$ then to me it would be evidence that that part of the analysis cannot be correct.

---

## Round 2 · Referee Report · Anonymous (Referee 2) · 2020-3-5

Report

The author has made an effort to answer to my comments and the ones of the other referee. I still disagree with the author about the scaling of the limit of capacity of the network and though at this point I do not consider this as an obstacle to publication, I invite the author to double thinking and to look to recent and old literature about this point. I personally find the notation in which factor nodes are denoted by a black square (instead of a letter) very heavy but I leave to the author the choice to change or to keep it.

---

## Round 2 · Author Response

I would like to thank the referees for their efforts and patience to carefully read the manuscript and provide me constructive comments. The following are my responses. I also put the list of changes in the end.

(Note) In the following, [A].. is my response to the comments. In my responses, I refer to the equation numbers Eq. (...), Figure numbers, section numbers and page numbers of those in the revised version.

report 1

[Strengths]

  1. The theoretical computation is remarkable and, as far as I can tell, novel

  2. Potentially very interesting and useful results

  3. Well organized and written

[A] Thank you very much for the very positive response.

[Weaknesses]

  1. The approximation used to derive the results might prove too crude (it's hard to tell). I have some doubts in particular about the teacher-student case.

  2. Some numerical experiments (especially in the teacher-student scenario) are rather far from the theoretical analysis

[A]

Concerning 1, after re-checking, I realized that there is no 'Gaussian approximation' in the bulk part. In addition, I also realized that the 'quenched boundary condition' is replaced by a slightly annealed one as discussed below.

Concerning 2, I added additional results in the revised version. Still, I admit that more simulations are needed which I leave for further works.

[Report:]

The author leverages an approximation that allows him to perform an in-depth replica analysis, at arbitrary level of RSB, of a neural network with an arbitrary number of layers of constant size and with sign activation functions. This is technically quite remarkable. The results are also very interesting, and seem to capture some important features of the network, since they seem to roughly agree qualitatively with the results of Monte Carlo numerical experiments.

[A]

I thank the referee for the very encouraging responses.

My main doubt about the claims of the paper arises in the teacher-student scenario. In this case, the theory seems to produce a paradoxical situation in which the student "loses track" of the teacher in the middle of the network but then recovers it toward the end. The author proposes (p.22/23) a solution based on a vanishing symmetry breaking field. This is possible and would be quite interesting if true. However, one may well suspect that this is the result of the Gaussian approximation used, which in turn forces a bi-directional symmetry that is not present in the original system. Personally, I find this latter scenario more likely, unless persuaded otherwise, simply because it seems to be the most straightforward explanation to me. The following discussion on the generalization performance reinforces in me this opinion, since (if I understand it correctly) it seems to imply perfect or near-perfect generalization of a random function (the teacher) even in the over-parametrized regime. In turn, that would imply that the inductive bias imposed by the architecture is extremely strong, meaning that any such architecture would tend to realize, with overwhelming probability, a tiny subset of all the possible mappings. I don't think that's the case. So am I missing something?

[A]

First, let me start fixing my own confusion in the previous version:

1) Actually there is no 'feed-forwardness' (directionality) in the statistical mechanics of the solution space (see my reply on 2019-12-04). The problem at our hand is just the statistical mechanics of a spin model (constraint satisfaction problem) with certain boundary conditions. I did not appreciate this enough at the time of the 1st version of this paper. I revised the text adding new remarks in sec 2.1 below Eq.(9) in the new paragraph "Now our task is..".

2) By re-checking the computation, it turned out the higher-order terms in the cumulant expansion ( to evaluate the interaction part of the free-energy) vanish in the bulk part of the system (see sec A.3.1). So the treatment was exact in the bulk. On the other hand, our treatment to simply put $q_{ab}(0)=q_{ab}(L)=1$, $Q_{ab}(0)=Q_{ab}(L)=1$ do not exactly enforce 'quenched boundaries' but slightly annealed ones. This issue is explained in the revised version (see below Eq. (14) and appendix A.3.2).

There still remains an interesting problem of how to realize 'physically' (in computers) the symmetry breaking field which allows real algorithms to detect correct phases. I revised the text of sec 3.5.1 (3rd paragraph) to emphasize this issue.

The numerical experiments tend to agree with the theory in a qualitative way, but they are performed on very small systems, and furthermore with binary weights (p.26, sec. 6.2). The binary weights case can potentially be quite different from the continuous case (and much harder). So this makes me wonder about the reason for this choice. (In the random scenario the weights were continuous.) Also, the networks used in the experiments are really small, with L=6 layers of N=4 units at most (by the way the text says "N×L values of the bonds", shouldn't it be "N^2×L" as in eq. (3), i.e. 96 at the most? Minus 20% that's 77 weights). I suppose this is due to the difficulty of training the student with MC. Still, I think that this may be problematic for the comparison with the theory, as finite-size effects could be very strong....

[A]

The reason for the choice of the binary perceptron for the learning simulation was just because of its simplicity: the couplings $J_{\bs}^{k}$ only takes binary values. On the theoretical side, we could also do another version of the replica theory for the case of the binary perceptrons but we do not expect much qualitative difference in the case of the DNN (although they are rather different at the level of single perceptron). Of course, it is better to check it explicitly. Actually we analyzed also the replica theory in which spins take continuous values (spherical spins) but the results were very similar to those presented in this paper.

$N \times L$ values of the bonds", shouldn't it be "$N^2 \times L$" as in eq. (3) : Thank you. I corrected this.

Finite-size (finite width $N$ ) effect should be there. It is displayed in Fig. 19 and discussed in the last paragraph of sec 5.2. The qualitative feature agrees with the theoretical expectation. On the other hand, in theory, the depth $L$ does not need to be a large number.

On top of this, the teacher-student scenario has the problem of the permutation symmetry in the hidden units when measuring the overlaps. The text (p.27) reads "In order to remove this unwanted symmetry, we removed 20% of the bonds randomly in each layer, but exactly in the same way, from both the teacher and student machines." I don't understand this procedure unless the bonds were removed even before the training phase. If so, it should be clarified. Also, this is another (probably minor?) difference with the theoretical analysis, and it might further exacerbate finite-size effects. Furthermore, on such small networks, removing at random 20% of the weights might not be enough to disambiguate all the units: it means removing less than 1 link per unit on average; from the point of view of two units in a layer, if their input and output links are masked in the same way they have the same local structure and they are thus interchangeable. Indeed, a very quick test shows that in a random mask on a network with N=4, L=6 there is about a 65% chance that at least one such ambiguity arises in one of the layers 2 to 5 (with L=4 the chance reduces to about 41%). This could contribute to reduce the apparent overlap considerably on average (each ambiguous pair that the student learns in the wrong order reduces the overlap of a layer by half with N=4, so very roughly it's a reduction by a factor of 0.75*0.65+0.35=0.84 on average).[To me, the most natural (albeit definitely more expensive) approach without masks would be to find the permutation of the indices of the hidden units that maximizes the overlap with the teacher, starting from the first layer and moving up one layer at a time. This is still sub-optimal but computationally feasible (it amounts at solving a small bipartite matching problem at one layer; applying the permutation to that layer and the next; moving up and repeating).]

[A]

Thank you for the comments and suggestions.

  • Removal is done before the training phase. I indicated this in the revised text in sec 5.2 below Eq (37).

  • I added an additional simulation with 40% of bonds removed. See Fig. 17-18. The results qualitatively did not change.

I think that the author should comment on this issue. But just to clarify, in my opinion even without any teacher-student analysis at all the results on the random patterns case would still be extremely valuable and interesting, so I don't consider this as a major problem for publication.

[A]

Thank you for the useful comments and positive evaluation of our work.

Apart from this, there are still a few unclear points for me. One is the definition of "over-parametrized regime", which is said (p.6) to be when L>α. Yet, many experiments (e.g. fig. 7) use L=20 and α=4000, and the results suggest that the network is still "liquid" in the middle, thus that it is not at capacity. Either I'm not understanding what is meant with "over-parametrized regime" (and perhaps this is related to the teacher-student results discussed above), or this is an effect of the approximation. Relatedly: Is there a way to compute the critical capacity for the network in this framework? And how would it scale, and how would the order parameters look like? Etc. Or is there a reason not to compute it, technical or otherwise? I wish that the author clarified these points.

[A]

Thank you for these comments. The fact that the 'penetration depth' grows logarithmically with $\alpha$ means that the glass transition point $\alpha_{\rm g}(l)$ grows exponentially width the depth $l$. I added a new figure Fig. 6 which displays the exponential growth of the glass transition point. This strongly suggests that the storage capacity (jamming point) $\alpha_{\rm j}(l)$ should also grow exponentially with the depth. This is very far from what one would expect from the worst-case scenario which would say linear growth with the depth. We think this is a difference between the worst instances and typical instances that we analyze in the theory. Analyzing explicitly the capacity limit is an important problem by itself ( and easy in the sense that we just need to limit $L$ and increase $\alpha$ so that $\xi > L$ is achieved, to remove the liquid region) but we will do it in a separate work.

So the definition of over-parametrization as $\alpha > L$ is good for the worst instances and I decided to keep this terminology. But for the typical instances, the liquid phase disappears only for $\xi \propto \alpha > L$. I revised the text in sec. 4.1.4. (below Eq (24)) to highlight these issues. I also added a small remark below Eq. (5).

My only remaining issues are very minor points or typos, and I'll leave them for the "Requested changes" section.

[A]

Thank you very much for the very careful reading!

[Requested changes]

Apart from the points mentioned in the main report, here follows a list of minor issues:

  1. (p.4): "using a Gaussian ansatz" -> I don't think it can be called an ansatz, I suggest to use "approximation"

    [A]

    I removed "Gaussian ..." in the revised version because the theory was not a Gaussian approximation as mentioned above.

  2. (p.6): the notation for eq. 5 is a little ambiguous and confusing. As written, it looks like the expression is factorized on the nodes. Probably, the easiest fix would be to add parentheses that clarify that the first two products are meant to represent a multidimensional integral. Alternatively, a more common notation I've seen for this is to bring the products inside the integral (maybe with parentheses to group the integration variable and the spherical constraint).

    [A]

    Thank you. I fixed this.

  3. (p.8): "In Eq. (11) it is assumed that all replicas follow the same labels breaking the permutation symmetry. Second, the system is symmetric under permutations of perceptrons [ ] within the same layer and the permutations could be done differently on different replicas. In Eq. (11) this permutation symmetry is also broken." -> I think that it should be clarified how this second breaking (the one on the units) is achieved, as it is not clear at all.

    [A]

    Solutions with other permutations give exactly the same free-energy. It is similar to what happens when we study the ferromagnetic phase of the $O(N)$ spin model. THere we just need to consider the case that spins are pointing toward, say the 1st axis in the spin space. In learning dynamics, the choice of the initial condition will play the role of the symmetry-breaking field. (Of course in real networks the width $N$ is finite so there will be no real symmetry breaking like this.)

  4. (p.19, sec. 4.2): The second argument for the input overlap fluctuation structure reads "It is natural to consider the case that input data fluctuates during learning as it happens in the standard Stochastic Gradient Descent (SGD) algorithm." However, in SGD the fluctuations happen as a byproduct of the learning algorithm, whereas the optimization goal is still in principle to learn the whole, non-fluctuating training set. To me, what the author suggests seems more similar to adding noise in the inputs, or to some stochastic data augmentation techniques perhaps. Even then, assuming an ultrametric structure does not seem obvious or particularly "natural" for those scenarios to me. Unrelatedly, in the third argument, "Real data may not be totally random" is quite an understatement!

    [A]

    I understand these comments. I removed 2 and 3rd ones. I added a new one to mention a connection to unsupervised learnings in which the configuration on the boundary is forced to follow some externally imposed probability distribution.

  5. (p.21, and analogously for the beginning of p.29 where the argument is repeated): "We can regard this as a kind of ’renormalization’ of the input data [...] This means that a DNN works naturally as a machine for renormalization, i.e. classifications and feature detections." I think I understand what the author is saying here, and it is certainly a very interesting observation. However, I think that the link between the "renomarlization" operation as shown by the results of the analysis and its interpretation in the context of classification and feature detection is not as straightforward as is being put here. In my opinion, this sentence would require either a slight reformulation or to be expanded with additional discussions and arguments.

    [A]

    I understand the comments. I revised the text at the end of sec 3.4.2 adding a sentence "It will be very interesting to study further the implication of this result in the context of data clustering where the idea of ultrametricity is very useful.". I also changed the last sentence of item c) at the end of sec 5.1 as "Probably the spatial 'renormalization' of the hierarchical clustering and the presence of the liquid phase at the center stabilize the system against external perturbations or incompleteness of equilibration and contributes positively to the generalization ability of DNNs."

  6. (p.24-26, sec. 6.1): Maybe I missed it but do the MC simulations reach zero error? A plot of both the loss and the error as a function of time would be useful.

    [A]

    I included a plot of relaxation of energy (at each layer) in Fig. 16. The energy keeps relaxing (aging) within the time window. The relaxation is slower closer to the boundaries. The stationary value of the energy is not zero because the system is at finite temperature.

  7. (p.45, on the numerical solution of the k-RSB equations). There are several points where the procedure requires to compute functions involving a variable h (e.g. f, f', P..), which is then integrated over. Was this done by sampling h (and m) at regular points on a grid? If so, at what interval? If not, how? Please expand a little on the numerical procedure.

    [A]

    I added comments on these at the end of sec. B. 3.3 (appendix).

  8. Some figures (e.g. figs. 15, 16) are very hard to read even when zoomed, due to poor resolution. The author seems to be using gnuplot; I suggest using a vector graphics terminal such as svg (it can then be converted easily to pdf) or eps rather than a raster terminal such as pngcairo.

    [A]

    Unfortunately, the software which I'm using cannot handle svg files correctly. So I tried to enlarge the size of the figures as much as possible. I hope they are better now.

  9. Typos:

    [A]

    Thank you very much for the careful check! I corrected the following.

    • throughout the paper: "i. e." -> "i.e." p.4: "it can happen that solution space" -> missing "the" [fixed] p.4: "to understand the efficiently" -> "...the efficiency" [fixed] p.4: "a statistical inference problems" -> "...problem" [fixed] p.4: "creases" -> "increases" [fixed] p.5: "perceptrons A perceptron" -> missing full stop before "A" [fixed] p.6: "where we introduced 'gap'" -> "...the 'gap'" [fixed] p.7: "From physicist’s point of view" -> "...a physicist's..." [fixed] p.7: "As a complimentary approach" -> "...complementary..." [fixed] p.7: "by construction its own synaptic weights" -> something went wrong with this sentence, maybe "by adjusting its own synaptic weights" etc.? [fixed] p.11: "a delta functions" -> "a delta function" [fixed] p.15: "As the result it looks approximately like" -> "...as a result..." [fixed] p.15: "this amount to induce" -> "...amounts..." [fixed] p.16: "the 2nd glass transition induce" -> "...induces" [fixed] p.16: "an internal step like structure emerge continuously" -> "...emerges..." [fixed] p.16; "the emergence of the internal step amount" -> "...amounts" [fixed] p.16: "have interesting implications" -> "has..." [this sentence is removed] p.16: "the number of constraints increase, the allowed phase space become suppressed" -> "...increases...becomes..." [this sentence is removed] p.16: "the probability appear to decay" -> "...appears..." [fixed] p.17: "which leave behind river terrace like structure" -> "...behind a river-terrace-like structure..." [fixed] p.17: "finite glass order parameter emerge continuously" -> "a finite glass order parameter emerges..." (or "...parameters...") [fixed] p.17: "the layers included in the glass phase is" -> "...are" [fixed] p.17: "and then it implies" -> "and this implies" [fixed] p.17: "1−q(x,l) ,i. e." -> swapped comma/space [fixed] p.18: "To understand meaning" -> "...the meaning" [fixed] p.18: "The hierarchical organization of clusters imply" -> "...implies" [fixed] p.18: "small valleys are group" -> "...grouped" [this sentence is removed] p.18: "it progressively become" -> "...becomes" [fixed] p.18: "in deep enough network" -> "...networks" (or "in a deep enough...") [fixed] p.19: "suggests that basic hierarchical structure" -> "...that the basic..." [fixed] p.19: "spin configuration on the boundaries are allowed" -> "...configurations..." [fixed] p.19: "exhibit hierarchical, structure" -> "exhibits a hierarchical structure" [fixed] p.20: "the resultant glass order parameter" -> "...resulting..." [fixed] p.20: "the hierarchical structure put in the input propagate" -> "...propagates" [fixed] p.21: "Numerical solution suggests" -> "The numerical..." [fixed] p.21: "are progressively renormalized into $q=0$ sector" -> "...into the $q=0$ sector" [fixed] p.22: "which read" -> "which reads" [fixed] p.22: "grow from the boundary" -> "grows..." [fixed] p.23: "Most likely scenario is" -> "The most..." [fixed] p.23: "which do not contribute the order" -> "...does not contribute to..." [fixed] p.23: "remain in the central part" -> "remains..." [fixed] p.23: "naturally arise by" -> "...arises..." [fixed] p.24: "Each component of the spins only take...each of the bonds take" -> "...takes...takes" [fixed] p.25: "in the low a) ... in the low b) ... in the low c)" -> "in row a)..." etc. [fixed] p.27: "we used systems with L=4" -> it seems from fig. 16 that L=6 was also used, and that α=4 was only tested there. [fixed] p.28: "We argued that small the positive overlap can remain" -> "...that the small positive..." [fixed] p.28: "the opposite side of the system" -> "...sides..." [fixed] p.28: "which appear to be compatible" -> "...appears..." [fixed] p.28: "sclae" -> "scale" [fixed] p.28: "Weak bias which remain" -> "...remains" [fixed] p.30, caption of fig. 15: "In the 1st low" -> "...row"; same for 2nd and 3rd [fixed] p.32, fig. 17, panel a): The points are centered at half-integer values? [Yes. This is done on purpose. I put an explanation in the caption.] p.36 eq. 51: I suppose that c in the first equation (and in eq. 52 and in the sentence after eq. 53) should be N? [fixed] Also, shouldn't there be an (irrelevant) $1/2\pi$ factor? [fixed] p.38 eq. 66: There's an extra power of 2 in the 3rd order term, it should be $⟨A^{2}⟩⟨A⟩$ not $⟨A^{2}⟩^{2}⟨A⟩$. [fixed] p.38 eq. 67 (also eq. 68): I think that a property is used here which is a generalization of eq. 59 to the case of different nodes. What about writing the general form in eq. 59, with an extra delta? [fixed] p.45, point 2, step (7): should it be "compute Gi(l) ... using Eq. (112)"? [fixed] p.45, point 3, step (6): should "using eq. (18)" be "using eq. (87)"? [fixed] p.47: "One just keep in mind" -> I guess "...must..."? [fixed]

report 2

[Strengths]

1- The paper addresses a notoriously difficult problem

2- It proposes a new 'gaussian approximation' that allows analytic progress.

[A]

Thank you for the very positive response. Actually, after careful re-checking, I realized that the theory is not a 'gaussian approximation' but exact in the bulk part of the system. On the other hand, I also realized that 'quenched' boundary is replaced by a slightly annealed one. See the beginning of my response to << report 1>> [Report:].

[Weaknesses]

1- The choice of the regime M/N finite is not the most relevant one both for the storage problem and for generalization.

2- The paper concentrates on technical details much more than on physics.

3- The meaning, qualities and limitations of the main approximation are not discussed.

[A]

I do not agree with comment 1 (see below). I understand comments 2,3. I revised the text to improve concerning these points.

[Report:]

This paper propose an approximate analytical treatment to study the Gardner volume of multilayer neural networks. This is a notoriously difficult problem and any advances are welcome. The author considers networks of $L$ layers of $N$ formal neurons each, in the limit $N \to \infty$. The addressed problems are (1) the storage problem of random associations and (2) the generalization problem in the student-teacher setting. The author consider a number of examples $M$ given to the net which scales as $N$, namely $M=\alpha N$. This is far from the capacity in problem (1) and from the onset of generalization in problem (2) which both should scale as $M_{\rm c}\propto N^{2}L$.

[A]

$M_{\rm c}\propto N^{2}L$ is for the worst-case scenario. For typical instances, the present scaling is the relevant one. I put a remark below Eq. (5) and added discussions in sec 3.3.5.

The author introduces a 'gaussian approximation' that allows him an approximate analytic treatment through the replica method. Unfortunately, as a result of the approximation, the networks lose their feed-forward directionality and become symmetric under the exchange of input and output. For small $α$ the authors find that the system is in a 'liquid phase'. Increasing $α$ the author finds that freezing of couplings and internal representations propagates from the boundary towards the interior of the network, with a characteristic 'penetration length' scaling as $\log α$. A very detailed description of the various transitions is proposed. The propagation of freezing from the boundary towards the interior is confirmed by Montecarlo numerical simulations. Quite surprisingly the author claims that some form of generalization is possible in problem (2) for finite $α$. It is not clear to me if this is just an artifact of the gaussian approximation.

[A]

Actually 'feed-forwardness' disappears when we consider the statistical mechanics of the solution space. And the theory was not a 'gaussian approximation' but the 'quenched' boundary is replaced by a slightly annealed one as discussed above. See the beginning of my response to << report 1>> [Report:].

Though I am convinced that the results of the paper are potentially worth to be published, I found the paper very difficult to read, it concentrating on the explanation of the details of the replica techniques used in the paper, and much less on the physical motivations of the choice of the models and regimes under study, the implications of the various solutions for learning and so on. Above all, meaning, qualities and limitations of the gaussian approximation, that potentially is the main contribution of the paper are barely discussed at all.

I sincerely think that the results are potentially interesting, but the paper in the present form does not render justice to them. I would suggest the author to rewrite completely the paper concentrating much more on physically salient points rather then on replica details.

[A]

Thank you very much for the comments. I tried to revise the text adding more discussions and sending the detailed technical parts to appendices.

[Requested changes]

My suggestion is to rewrite completely the paper, with emphasis on physics rather than on technicalities.

[A] I added more discussions on physics in the main text and sent the technical details to appendices. I added a new section 3.6 to summarize the theory part.

---

## Round 2 · List of Changes

Abstract:

Slightly revised. The exponential growth of the capacity limit is mentioned explicitly. The word 'Gaussian approximation' is removed.

Main text:

  • sec 1. Introduction: A new 6th paragraph is added to emphasize that the system is near 'disorder-free' except for the boundaries. In the 7th paragraph, the exponential growth of the capacity is mentioned again.

  • sec 2. Model:

Below Eq (5), a new paragraph "The trajectories of such ..." is added. Here I mention the known results on the chaotic trajectories of random neural network which I believe important for discussions. I mention this often in this revised version.

Below Eq (9), a new paragraph "Now our task is ..." is added. There it is pointed out that 'feed-forwardness' can be forgotten in the statistical mechanics' problem. In the next new paragraph "The problem at our hand...", the analogy with assemblies of hard-spheres is used to provide some physical intuitions on our statistical mechanics problem.

  • sec 3. Replica theory: Extensively revised to make it less technical. I sent many of the technical details to appendices. More physical discussions are added. A new section, sec. 3.6 "Summary" is added to summarize the theoretical results.

  • sec 4. Simulations of learning: Slightly revised because of the new Figure, Fig. 16 and Fig. 18 (see below)

  • sec 5. Conclusion and outlook: In sec 5.1, item c) is revised. sec 5.2 "Outlook" is revised adding more comments.

References: some additional references are added.

Appendix:

  • sec A. Replicated free-energy : largely expanded including more details.
  • sec B. RSB solution for random inputs/outputs: no important changes.
  • sec C RSB solution for the teacher-student setting: slightly revised fixing some errors in (129),(130) and (154).

Figures:

  • A new figure Fig. 6 is added to show how the glass transition point $\alpha_{\rm g}(l)$ depends on the depth $l$.
  • In Fig. 7 (Fig 8 in the previous version), a new panel b) is added to show that $x(q)$ shows the same scaling as $x(Q)$.
  • Fig 13 is revised using the corrected saddle point equation (154).
  • A new figure Fig 14 is added to show a schematic picture to summarize the theoretical results.
  • A new figure 16 is added to display relaxation of energy during the simulation of the learning dynamics.
  • A new figure 18 is added to display the new data obtained by removing 40\% of bonds in the simulation of teacher-student setting.

---

## Round 3 · Author Response

I would like to thank the referees for their great efforts and patience to carefully read the 2nd version of the manuscript. The comments are very much useful. The following are my responses. I also put the list of changes in the end.

(Note) In the following, [A].. is my response to the comments. In my responses, I refer to the equation numbers Eq. (...), Figure numbers, section numbers and page numbers of those in the revised version.

report 1

[Strengths]

  1. The theoretical computation is remarkable and, as far as I can tell, novel
  2. Potentially very interesting and useful results
  3. Well organized and written

[A] Thank you very much for the very positive response.

[Weaknesses]

  1. I believe that the results are approximate (as was actually claimed in the previous version) rather than exact/quasi-annealed (as claimed in this version). Given that, I'll report my previous comment: The approximation used to derive the results might prove too crude (it's hard to tell). I have some strong doubts in particular about the teacher-student case.

    [A] I realized that the theory is a truly mean-field 'approximation' for the problem: a tree-approximation is made for the inter-layer fluctuations. Now it is clear that the 'left-right symmetry is an accidental symmetry due to the tree-approximation. Loop corrections will remove this. Although how other aspects of the results change by taking into account loop corrections is an open question, I believe other essential features remain.

  2. The setting of some of the numerical experiments (in the teacher-student scenario) are rather far from the theoretical analysis. Additionally, when the setting is pushed a little more in the direction of the theory (increasing N), the results go in an opposite direction than the theory suggests (generalization capability drops).

    [A] I agree with the referee. I decided to remove the presentation of the numerical simulation on teacher-student setting from the present paper leaving it for future works.

[Report]

The author has amended and expanded the paper and in doing so he has fixed most of the issues that I had raised in my previous review.

The new version however also includes now some stronger claims with respect to the previous one, as a result of a new computation (eq. 71) which shows that some terms of order higher than 2 become negligible in the bulk of the network. Based on this result, the author has removed the previous description of the computation as a "Gaussian approximation", claiming instead that the result is an "exact replica calculation" except for the boundaries where the calculation is not quenched but "slightly annealed". Since some features of the results would be surprising (mainly, the symmetry of the solution to an asymmetric model), the author presents some qualitative arguments and some analogies with other systems (e.g. hard spheres) in support to this claim.

I'm not convinced that this claim is correct. In brief, the procedure detailed in section A.1.4 called "Plefka expansion" is only exact in the limit of weak interactions (small $\lambda$), otherwise eq. 53 does not hold. But in that section it is written: "The parameter $\lambda$, which is introduced to organize a perturbation theory, is put back to $\lambda=1$ in the end." This is a weak-interactions approximation. It seems to me that this procedure is crucial to derive the results of the paper, in particular eq. 62, which allows to derive eqs. 63-64 which in turn allow to perform the necessary simplifications in eq. 71, which is the basis of the new, stronger claim. Indeed, the author writes that eq. 62 corresponds to eq. 52, which is valid in absence of interactions.

(I should say at this point that I have some additional comments about the clarity of the derivation in the appendix; it's possible that I misunderstood something, see below.)

[A] I understand the concern of the referee. As I noted above, the theory is actually a tree-approximation that neglects some loop-corrections needed to describe the one-dimensional inter-layer fluctuations. In the revised text, I explain this point in detail.

From the qualitative point of view, the author argues that the surprising symmetry of the result stems from the infinite $N$ limit, but I fail to see how that would happen here. For example, one argument is that "The system is locally asymmetric (at the scale of each layer) but globally homogeneous except at the boundaries." but I don't see this supposed homogeneity; the expression is very clearly still asymmetric. I don't dispute that it may happen that an asymmetry like this might become irrelevant in some limit for some models, I'm just not convinced that this is the case here, at least not exactly. The "entropic argument" given at p.14 is not unreasonable, but it's qualitative and an a-posteriori explanation; certainly it's not sufficient by itself to justify exact symmetry. (On the other hand, I still think that the results are relevant and probably qualitatively valid, and if so that the entropic argument gives a plausible mechanism).

[A] I fully understand the concern of the referee. As I noted above the left-right symmetry is just an accidental one at the level of tree-approximation. Apparently, loop-corrections should induce asymmetry especially close to the boundaries where more microscopic information is needed for accurate descriptions. This is explained in the new version.

The generalization result for the teacher-student scenario is still the most problematic in my opinion, as I still have a very hard time believing that near-perfect generalization is possible with a liquid phase in the middle of the network, just from "combinatorics-like" arguments (different networks should correspond to different N-bits-to-N-bits mappings, once permutation symmetry is accounted for). I had previously argued that, if one seeks an explicitly symmetric solution, imposing an overlap 1 at the initial end would obviously result in an overlap 1 at the other end. Now the author argues, from eq. 71, that the symmetry in the solution emerges spontaneously in the large $N$ limit rather than being enforced a priori from an approximation. Since as I wrote above I'm not convinced that that's the case, I maintain my previous opinion. One way to check if I'm right could be to test the equations at smaller values of $\alpha$: it the output overlap is always 1, implying perfect generalization for any size of the training set, which would be clearly absurd (one expects to see something like the single teacher-student perceptron with continuous weights, where the error decreases with $\alpha$), then the result is an artifact of the approximation.

[A] I fully understand this concern. In the revised version I keep the presentation of the replica analysis on training but revised extensively the discussions. The main problem is how to replace the fictitious symmetry breaking field used in the theory by some real ones. In the revised version I present a discussion on 'unlearning' (instead of learning).

Relatedly, the results of the simulations in the teacher-student case use extremely small values of $N$ and are thoroughly unconvincing. I have reimplemented the procedure described in the paper, and with the same parameters $(N=4, L=6, \alpha=1,2,4)$, I obtain precisely the same results for the generalization overlap that are reported in fig. 17, panel c, right plot. However, increasing $N$ makes all the overlaps drop drastically. As an example of parameter settings that I could test quickly, for $L=10$ and $\alpha=1$, I obtain the following values for $N=4,8,12: 0.65, 0.21,0.09$. I see similar results with $L=20$. I strongly suspect that the limit is $0$ for large $N$. In fig. 19, where some analysis on the dependence of the results with $N$ is shown, the overlap for the output layer (the one which actually correlates with generalization) is not shown in panel c, as far as I can tell. I wonder why. In any case, the values of $N$ used are 3,4,5 which makes it really hard to observe any effect.

If I'm right, I think that the whole teacher-student part should be either heavily revisited or left out entirely for a more in-depth study.

I have some additional issues that I'll leave for the "Requested changes" section.

[A] I truly thank the referee for doing independent simulations and also for the comment 2020-02-28. I agree that the effect becomes weaker everywhere including the output as I reported in the comments in my comments 2020-02-28 and 2020-03-01. I decided to remove this part from the present paper leaving it for future, more extensive numerical works.

[Requested changes]

The big ones, if I'm right, are to:

  1. Revert the description of the results as approximate;

    [A] Yes. Done.

  2. Revisit entirely the teacher-student generalization issue and corresponding simulations (possibly, removing the whole section and leaving it for a follow-up more in-depth study, ideally accompanied by convincing numerical evidence).

    [A] I revised extensively the theoretical part on the teacher-student setting and remove the numerical part.

    Apart from that, the following notes are mostly about typos or very minor issues, but there are a few more serious issues too.

  3. p.10 eq. 12: I think there has been a mix-up here, the $\eta$ integral doesn't belong here, it's introduced by the Fourier transform in eq. 38

    [A] Right. I fixed this.

  4. p.12: "Maybe one would not easily expect such asymmetry in the feed-forward network" I think the author meant "symmetry" here.

    [A] This sentence is removed.

  5. p.27 eq. 34: The update move was chosen in such a way as to ensure that on average the spherical constraint on the weights is satisfied. However, due to the random-walk nature of the update, the averages may "diffuse" over time. This can have an effect on the autocorrelations, eq. 35, especially since $N=20$ is being used which is fairly small and might not allow to neglect effects of order $\sqrt{N}$. It might be necessary when computing the autocorrelations to rescale by a factor $∥J(t)∥$ (I'm assuming $J(0)$ is normalized, otherwise it should be too). A quick check should be sufficient to detect whether this is an issue or not.

    [A] The normalization (rescaling) is always done. I also checked the effect of "diffusion" as suggested by referee and confirmed that it does not affect the results.

  6. p.28: "Note that the behavior of the system is not symmetric to the exchange of input and output sides. We expect that this is a finite $N$ effect which disappears in $N \to \infty$ limit." I don't expect that, see comments in Report. Providing evidence with varying $N$ might be a good idea here.

    [A] I changed this. The symmetry in the theory is an accidental one due to the tree-approximation as noted above.

  7. p.29: It seems like the same $M$ was used for training and testing, but there is no reason for limiting the amounts of testing patterns: the larger the $M$ used for testing, the more accurate is the measurement.

    [A] I agree. I will do so in the future. For the present paper I remove this simulation.

  8. p.39-43, figs. 15-19: The figures are very far from the portion of the text where the simulation results are presented and discussed, making it rather hard to follow. The caption of fig. 17 at p. 41 goes beyond the end of the page and it's partially unreadable (on the other hand reducing the figure size would make it hardly readable because of poor resolution, so the author should consider finding a way to improve the figure resolution).

    [A] I revised the figures improving the resolution. For Fig 15 (which is Fig 16 in the new version), I removed the three panels on the right hand side. I created a new figure Fig. 17 for the plots with logarithmic time scales where I show finite width $N$ effects.

  9. p.44 eq.38: There should be an $a$ index in the products of the $J$ and $S$ traces in the second line. Also is $\xi_{\mu \nu}$ a Kronecker delta symbol? Otherwise I can't make sense of this expression and the sudden appearance of another pattern index $\nu$. (And even if my interpretation is correct, the purpose of doing this is very unclear anyway.)

    [A] Fixed.

  10. p.44 (below eq. 39): "free-energy" repeated twice.

    [A] Fixed.

  11. p.44: "excplitely" -> "explicitly"

    [A] Done.

  12. p.46 eq.53: [Mentioned above] This equation is only valid for small $\lambda$. If then $\lambda$ is "put back to 1", the result is an approximation.

    [A] For some systems (mean-field models) the expansion stops exactly at $O(\lambda)$ and one obtain exact results. However, in the present paper, we have to invoke the tree-approximation to stop at $O(\lambda)$. I put these remarks below (55) in the new version.

  13. p.47 eq. 55: The $\epsilon$ and $\epsilon$ terms should multiply the second term inside the exponent too (the sums)

    [A] Fixed.

  14. p.47: "assuming $c \gg 1$" this is a leftover from the previous version, it should be $N$.

    [A] Done.

  15. p.47 eq. 59: There are some superscripts $\mu$ which don't make sense there, I guess they should be $i$ (and to follow previous formulas they should go inside parentheses, with the replica indices outside).

    [A] Fixed.

  16. p.47 eq. 61: The replica and pattern indices are swapped compared to the previous formulas.

    [A] Fixed.

  17. p.47: The expressions in eqs. 56-61 are not clear, one has to guess a few things in order to follow. Mainly, that the dots on the l.h.s. of 56,57 seem to refer to some expression that can be factorized over the indices $i$ or $\mu$ (depending on the formula) and that the corresponding dots on the right hand sides are the factorized expressions (which however still depend on the index $i/\mu$?).

    [A] Yes. I put an explanation below Eq. (59).

    Also since the $\blacksquare$ index has disappeared it seems like the assumption is that the dotted expressions do not depend on those.

    [A] I put $\blacksquare$ in Eq. (58),(59).

    Moreover the index $i$ was dropped from the expressions in eqs. 58-59, but there is a spherical constraint on the $J$s (eq. 65) so that the trace over $J^{c}$ is actually not well defined here and the reader has to guess what's actually going on. At least defining the trace operators like in ref. [20] might help a bit.

    [A] I put an integral expression of the spherical constraint in (10). Then below (60),(61) I explain that I included $\epsilon_{aa}$ in (60),(61) to include the integral to induce the spherical constraint.

    Additionally, the expression in the last line of 38 is not actually factorized over the $\blacksquare$, of course, so clarifying the assumptions used here (and how they relate to the $0$-th order expansion of the previous section) is quite important, I think. That expression becomes "tractable" after the expansion of eq. 71, but then only because they use the simplifications allowed by eqs. 63-64, which were derived in the interaction-free assumption...

    [A] I understand. I think it is now better.

  18. p.47 eq. 62: "where $\epsilon^{∗}_{ab}=[...]$ are determined by": again, it should be clarified here that these expressions come from the non-interacting assumption. In terms of the previous expression, it is like neglecting the dotted expressions, right? Otherwise the saddle points on the eplisons would involve those as well. This is basically said in the following paragraph when it's pointed out that they correspond to eq. 52, but I think that making it more explicit before the expressions would clarify the derivation considerably.

    [A] I understand. I put the explanation just below Eq. (64).

  19. p.48 eq. 65: $c \to N$

    [A] Right. I removed this and added a comment on the spherical constraint below Eq. ( 61).

  20. p.48 eq. 67: The $S$ should be lowercase.

    [A] Fixed.

  21. p.48 in the unnumbered equation between 68 and 69: The $c$ should be a superscript of $S$ instead of a subscript.

    [A] Fixed.

    Also $\epsilon_{aa}$ was never defined before, and it uses the wrong character ($\epsilon$ instead of $\varepsilon$); probably just writing that it is assumed to be 0 beforehand would be enough.

    [A] Fixed. $\epsilon_{aa}=0$ is put below Eq. (63).

  22. p.49: This section should probably refer to eq. 38 (the replicated one) rather then eq. 12

    [A] Done.

  23. p.50 eq.72: There's a factor 2 missing in front of the cosh resulting in a missing constant term in the following two lines; it's irrelevant, but still...

    [A] Right. Anyway this small section A.3.2.is not needed anymore so I rmoved it. Within the tree-approximation the boundary condition just amount to specify $q(0)_{ab}$ and $q(L)_{ab}$.

  24. p.50: [This one is more important than the others] The difficulty arising when the $J$ are considered quenched is mentioned. Isn't this precisely the teacher-student situation (at least for what concerns the teacher)?

    [A] Right. Anyway within the tree-approximation averaging over $J_{ij}$ is not a problem. The section A 3.2 is removed.

  25. p.51 and following: In all expressions with $S_{\rm ent}$, the $S$ should be lowercase for consistency with the previous sections.

    [A] Done.

  26. p.57 eq.115: a closing parenthesis is missing

    [A] Fixed.

  27. p.57 eq.123: the prime sign after the parenthesis in the numerator is a peculiar notation that should be defined.

    [A] I chaaged it to $d/dh$ operator.

referee 2

[Report]

The author has made an effort to answer to my comments and the ones of the other referee.

[A] I thank the referee for the positive evaluation on the revisions.

I still disagree with the author about the scaling of the limit of capacity of the network and though at this point I do not consider this as an obstacle to publication, I invite the author to double thinking and to look to recent and old literature about this point.

[A] I understand that there might be other possible scalings. One difficulty is that I cannot find relevant literatures on the capacity of DNNs. I put a remark in the text that there are some other possible scalings which involve $N$ differently in some two-layer problems in the revised version (below Eq. (5)). On the other hand, the numerical simulation on the learning dynamics suggests that taking $N \to \infty$ limit with fixed $\alpha=N/M$ is indeed reasonable (see Fig. 17, which is added in the new version).

I personally find the notation in which factor nodes are denoted by a black square (instead of a letter) very heavy but I leave to the author the choice to change or to keep it.

[A] I understand the notation is a bit unusual. But there are simply too many indiciess in this problem so I decided to use also the graphical symbol which looks like a label for a factor node.

---

## Round 3 · List of Changes

List of changes

  1. Introduction

2nd paragraph, 2nd sentence is introduced to mention that we consider the global coupling between layers. In the same paragraph, I mention that the present problem is a $1+\infty$ dimensional one.

7th paragraph, in the last sentence, I mention and briefly explain the tree-approximation.

  1. Model

In the paragraph below Eq. (11), I mention that in DNN homogeneity is expected in the bulk but inhomogeneity is expected close to the boundaries.

Fig 2 is added to explain the loop correction. In the paragraph below it, I explain that within the tree-approximation the inhomogeneity close to the boundaries cannot be taken into account accurately.

3.5 Teacher-student setting

The discussion is revised extensively. The last 3 paraphs are new.

  1. Simulation of learning

The presentation of the teacher-student setting is removed.

The text for the presentation on the random input/output setting is revised adding discussions on finite width $N$ effects and crossover from glassy dynamics to rapid decay.

Fig. 10 is revised removing some figures and increasing the resolution.

A new figure Fig. 17 is added to explain finite width $N$ effects.

Fig 18 is also revised adding data with various width $N$.

I noticed that the quantities presented here are the overlap between two (real) replicas instead of time auto-correlation functions (the behavior is very similar). I corrected the text concerning this.

  1. Conclusion and outlook

3rd paragraph, It is mentioned again that the tree-approximation is a weak point of our theory.

4th paragraph. It is emphasized that $\alpha=M/N$ scaling is supported by the numerical simulation.

Appendix

sec A.1.4. The last paragraph is added to mention some exact results and the limitation of the present work due to the tree approximation.

sec A.3. It is explained where the tree-approximation is needed.

---

## Editorial Decision

published